# Probing condensate microenvironments with a micropeptide killswitch

Yaotian Zhang[1,2], Ida Stöppelkamp[1,2,11], Pablo Fernandez-Pernas[3,11], Melanie Allram[3,11], Matthew Charman[4,5,6,11], Alexandre P. Magalhaes[1], Melanie Piedavent-Salomon[1], Gregor Sommer[1], Yu-Chieh Sung[1], Katrina Meyer[1], Nicholas Grams[5], Edwin Halko[4,6], Shivali Dongre[7], David Meierhofer[1], Michal Malszycki[1], Ibrahim A. Ilik[1], Tugce Aktas[1], Matthew L. Kraushar[1], Nadine Vastenhouw[7], Matthew D. Weitzman[4,5,6,8], Florian Grebien[3,9,10], Henri Niskanen[1✉] & Denes Hnisz[1✉]

Biomolecular condensates are thought to create subcellular microenvironments that have different physicochemical properties compared with their surrounding nucleoplasm or cytoplasm[1–5]. However, probing the microenvironments of condensates and their relationship to biological function is a major challenge because tools to selectively manipulate specific condensates in living cells are limited[6–9]. Here, we develop a non-natural micropeptide (that is, the killswitch) and a nanobody-based recruitment system as a universal approach to probe endogenous condensates, and demonstrate direct links between condensate microenvironments and function in cells. The killswitch is a hydrophobic, aromatic-rich sequence with the ability to self-associate, and has no homology to human proteins. When recruited to endogenous and disease-specific condensates in human cells, the killswitch immobilized condensate-forming proteins, leading to both predicted and unexpected effects. Targeting the killswitch to the nucleolar protein NPM1 altered nucleolar composition and reduced the mobility of a ribosomal protein in nucleoli. Targeting the killswitch to fusion oncoprotein condensates altered condensate compositions and inhibited the proliferation of condensate-driven leukaemia cells. In adenoviral nuclear condensates, the killswitch inhibited partitioning of capsid proteins into condensates and suppressed viral particle assembly. The results suggest that the microenvironment within cellular condensates has an essential contribution to non-stoichiometric enrichment and mobility of effector proteins. The killswitch is a widely applicable tool to alter the material properties of endogenous condensates and, as a consequence, to probe functions of condensates linked to diverse physiological and pathological processes.

Cells organize their biochemistry within localized membraneless compartments that are generally referred to as biomolecular condensates[1]. Condensates are concentrates of various biopolymers, including proteins and nucleic acids, and are thought to have emergent properties and functions beyond those of their individual constituent molecules[1–5]. For example, condensates may create microenvironments that facilitate the partitioning of small molecules into the condensates, in a manner that is not explained by stoichiometric interactions[8,10–13]. Moreover, recent studies suggest that the material and viscoelastic properties of condensates impact the activities of their constituent molecules[7,14–16]. These findings, together with substantial in vitro biochemistry experiments, support an emergent view that the material state of molecules

within condensates is inherently linked to biological functions that emerge from concentrating them in one location[17]. However, experimentally manipulating condensate material properties, condensate microenvironments and emergent condensate functions in vivo has been a major challenge.

Probing condensate properties and function in vivo is challenging owing to the limitations of currently existing tools. Several technologies have been developed to induce condensation of proteins in cells[15,18–25]. Manipulation of the material properties and functions of condensates in cells nevertheless has been more difficult. Genetic perturbation experiments that attempt to separate condensate formation from biological activity have been useful[26,27], but the readout of such experiments

[1]Max Planck Institute for Molecular Genetics, Berlin, Germany. [2]Institute of Chemistry and Biochemistry, Department of Biology, Chemistry and Pharmacy, Freie Universität Berlin, Berlin, Germany. [3]Centre of Biological Sciences, University of Veterinary Medicine, Vienna, Austria. [4]Division of Protective Immunity, The Children's Hospital of Philadelphia, Philadelphia, PA, USA. [5]Department of Pathology and Laboratory Medicine, University of Pennsylvania Perelman School of Medicine, Philadelphia, PA, USA. [6]Division of Cancer Pathobiology, The Children's Hospital of Philadelphia, Philadelphia, PA, USA. [7]Center for Integrative Genomics, University of Lausanne, Lausanne, Switzerland. [8]Penn Epigenetics Institute, University of Pennsylvania Perelman School of Medicine, Philadelphia, PA, USA. [9]St Anna Children's Cancer Research Institute (CCRI), Vienna, Austria. [10]CeMM Research Center for Molecular Medicine of the Austrian Academy of Sciences, Vienna, Austria. [11]These authors contributed equally: Ida Stöppelkamp, Pablo Fernandez-Pernas, Melanie Allram, Matthew Charman. ✉e-mail: niskanen@molgen.mpg.de; hnisz@molgen.mpg.de

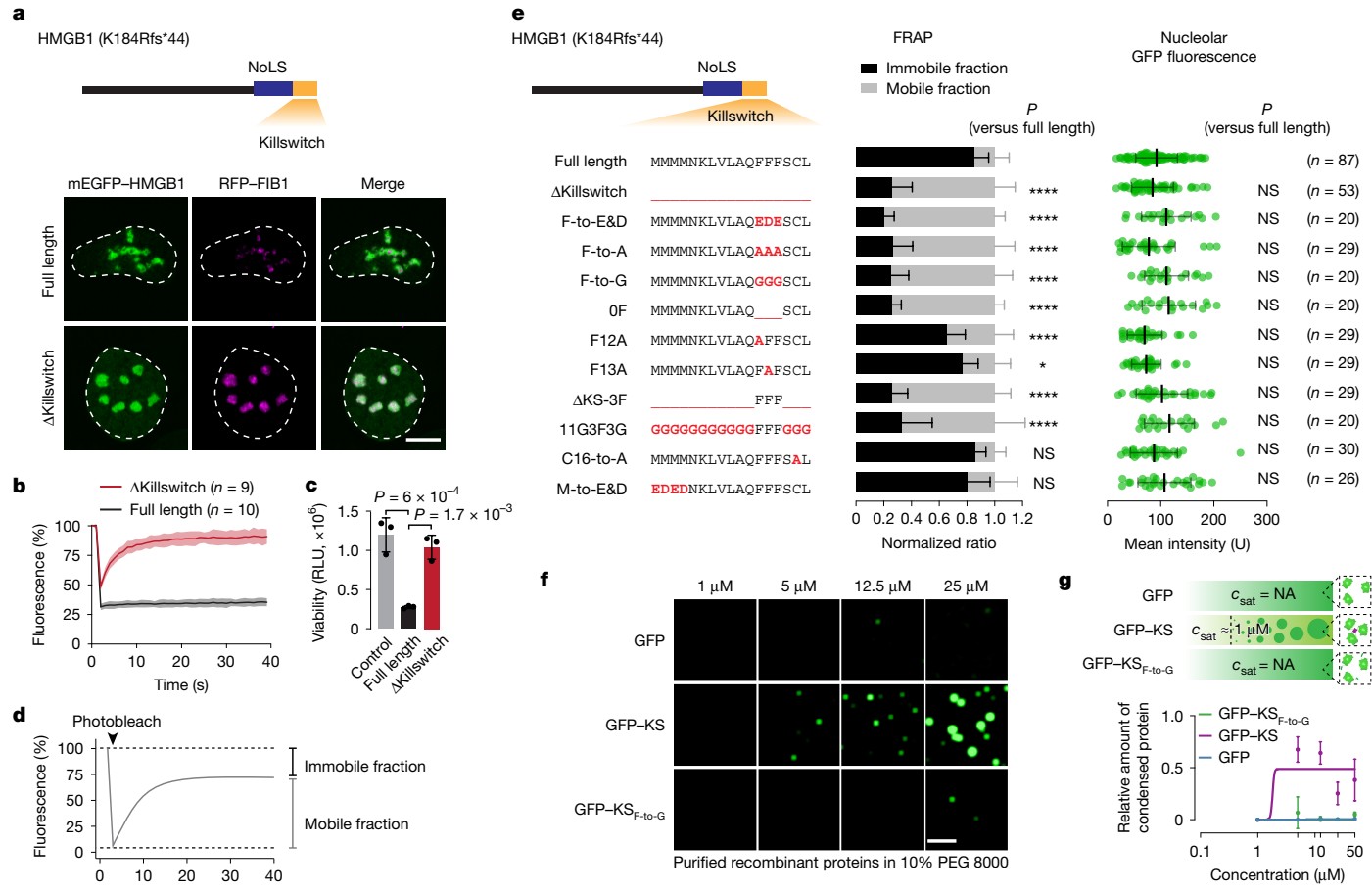

**Fig. 1 | The killswitch arrests nucleolar dynamics. a**, Schematic of fsHMGB1 (top). NoLS, nucleolar localization sequence; fs, frameshifted. Bottom, live-cell fluorescence microscopy images of U2OS cells expressing ectopic mEGFP–HMGB1 proteins and RFP–FIB1. The contour of the nucleus is highlighted with a dashed line. Scale bar, 10 μm. The experiment was repeated independently five times with similar results. **b**, FRAP analysis of mEGFP–HMGB1 in nucleoli. Data are mean ± s.d. **c**, Cell viability of U2OS cells expressing mEGFP–HMGB1 proteins. *n* = 3 biological replicates. Data are mean ± s.d. *P* values were calculated using two-way analysis of variance (ANOVA) followed by Dunnett's multiple-comparison test versus fsHMGB1(full length). RLU, relative light units. **d**, Schematic of the FRAP assay. The proportion of the signal that recovers in the experimental time window is a proxy for the mobile fraction of protein in the bleached condensate. The proportion of the signal that does not recover is a proxy for the immobile fraction of protein in the bleached condensate. **e**, The amino acid sequences of

the killswitch variants in the tested mEGFP–HMGB1 proteins (left). Middle, quantification of the mobile and immobile fractions of mEGFP–HMGB1 proteins in nucleoli. Data are mean ± s.d. Right, mean GFP fluorescence of bleached nucleoli. *n* is the number of nucleoli tested, from at least two independent experiment series. *P* values were calculated using two-way ANOVA (FRAP plots) or one-way ANOVA (GFP intensities) followed by Dunnett's multiple-comparison test versus fsHMGB1(full length). *$P < 0.05$, ***$P < 10^{-3}$, ****$P < 10^{-4}$. Exact *P* values are listed in the 'Statistics and reproducibility' section of the Methods. NS, not significant. **f**, In vitro droplet formation by purified, recombinant GFP-tagged killswitch peptides. The experiment was repeated independently twice with similar results. Scale bar, 5 μm. **g**, Quantification of the relative amount of protein in droplets. Data are mean ± s.d. *n* = 10 images from two independent experiments. $c_{sat}$, saturation concentration; NA, not available.

occurs dozens of cell generations after the genetic manipulation, by which time cells have adapted to the genetic changes. Small molecules that selectively target specific condensates are mostly lacking. Aliphatic alcohols, such as 1,6-hexanediol are commonly used, but such compounds disrupt weak hydrophobic interactions without any specificity, and therefore dissolve condensates in a non-specific manner[3]. For a few condensates, such as stress granules, selective molecules have been discovered[6–9,28]; however, for most condensates, such compounds remain to be identified. A few studies have reported protein-tagging approaches to dissolve ectopic condensates[21]. Deep light stimulation of proteins fused to the photo-oligomerizing CRY2 domain has been used to drive nucleoli into an arrested state[19,29], but such approaches rely on ectopic overexpression of condensate-forming proteins of which the behaviour is known to substantially differ from those of native proteins[3]. In summary, current approaches do not inform on the material properties of endogenous condensates, nor do they inform on the mechanistic underpinning of how material properties of molecules

in endogenous condensates contribute to condensate microenvironments and enable emergent functions. Condensate alterations have been linked to hundreds of human diseases[30–32]. Thus, universally applicable approaches to target endogenous condensates, and change their material properties in particular, could provide important insights into emergent functions of condensates implicated in a large variety of processes in health and disease.

## Features of the condensate killswitch

We previously identified a pathogenic frameshift variant of the nuclear protein HMGB1 that causes a rare complex malformation syndrome[31]. The frameshift generates a de novo C-terminal tail that leads to the mispartitioning of the mutant HMGB1 protein into the nucleolus, the largest biomolecular condensate in human cells[31]. The mutant HMGB1 arrests the dynamics of the nucleolus, ultimately leading to cell death[31] (Fig. 1a–c). The nucleolar arrest appears to be caused by the C-terminal

17 amino acids, as deleting these residues does not change nucleolar mispartitioning, but rescues nucleolar dynamics and cell viability[31] (Fig. 1a–c). We therefore hypothesized that the 17-amino-acid micropeptide sequence, hereafter referred to as the killswitch, could be engineered into a generally applicable tool to alter material properties of specific condensates, and as a consequence, to probe condensate microenvironments and functions in living cells.

We performed systematic mutagenesis of the killswitch to identify the sequence features and molecular mechanism that drive nucleolar arrest. GFP-tagged mutant HMGB1 transgenes were transiently expressed in human osteosarcoma (U2OS) cells, and the nucleolar dynamics were quantified as the ratio of mobile versus immobile fractions of GFP–HMGB1 in nucleoli using fluorescence recovery after photobleaching (FRAP) (Fig. 1d). We found that mutation of the cluster of three phenylalanines to negatively charged residues, alanines or glycines was sufficient to rescue nucleolar dynamics to the same degree as deleting the entire killswitch sequence (Fig. 1e). Mutation of individual phenylalanines did not lead to a rescue, suggesting that all three phenylalanines are necessary (Fig. 1e). However, the phenylalanines required the rest of the killswitch sequence, as mutation of the other 14 residues into glycine residues, or fusing three phenylalanines to a mutant HMGB1 lacking the killswitch sequence also rescued nucleolar dynamics (Fig. 1e). Shuffling the killswitch sequence further revealed that clustering of the phenylalanines was in itself not necessary, but enhanced nucleolar arrest (Extended Data Fig. 1a–d). The rescue of nucleolar dynamics correlated with rescue of the circularity of nucleoli and cell viability (Extended Data Fig. 2a–d). Similar results were observed when the killswitch was recruited to nucleoli through ectopic expression of the nucleolar protein NPM1 genetically fused to the killswitch sequence (Extended Data Fig. 2e–g). These results suggest that phenylalanines drive nucleolar arrest, but require the additional sequence context of the killswitch.

Droplet assays and size-exclusion chromatography (SEC) revealed phenylalanine-dependent self-association of the killswitch in vitro. The purified, GFP-tagged recombinant killswitch protein (GFP–KS), but not the F-to-G mutant (GFP–KS$_{F-to-G}$) formed droplets in a concentration-dependent manner in the presence of physiological salt (125 mM NaCl) and crowding agent (10% PEG 8000) (Fig. 1f,g). Consistent with the results of the droplet assays, around 18% of the GFP-tagged killswitch protein, but less than 1% of the F-to-G mutant, eluted as multimers after SEC (Supplementary Fig. 1a–c). Modelling the structure of the killswitch using AlphaFold 3[33] further confirmed that it is a helix-prone peptide that self-interacts and that the phenylalanines are necessary for the killswitch–killswitch interactions (Supplementary Fig. 1d,e). Taken together, we conclude that the killswitch has the ability to self-associate, and the phenylalanines have an important contribution to the valence of self-association.

We noted additional unique features of the killswitch. For example, the sequence itself does not appear in the human proteome (Supplementary Fig. 2a). A similar, short hydrophobic patch in TDP-43 was described previously, and oxidization of methionine residues in the patch is associated with the emergence of solid TDP43 aggregates[34]. In contrast to the killswitch, the TDP-43 hydrophobic patch failed to arrest nucleolar dynamics when tested with two different recruitment systems in cells (Supplementary Fig. 2b–h). Moreover, mutation of methionine residues in the killswitch did not affect its ability to arrest protein dynamics in nucleoli (Fig. 1e). The killswitch therefore appears to be non-natural in humans, and likely impacts condensate dynamic properties through a different mechanism than the TDP-43 hydrophobic patch.

## Targeting endogenous condensates

As a potentially universal approach to target the killswitch to any cellular condensate, we developed a nanobody-based recruitment system. We generated a DNA vector encoding the killswitch fused to an anti-GFP nanobody[35], with the idea that the anti-GFP nanobody (GFP-nb) would enable recruitment of the killswitch to any GFP-labelled condensate (Fig. 2a,b). The DNA vector also contained an mCherry reporter expressed bicistronically, because we found that fusion of the nanobody to mCherry was sufficient to perturb GFP–NPM1-labelled nucleoli, potentially due to the large size of the fusion protein (Extended Data Fig. 3a,b). Thus, the nanobody-expressing cells, but not the nanobody itself, are identified by mCherry in this system (Fig. 2b,c), which was corroborated using a nanobody variant attached to an HA tag (Extended Data Fig. 3c).

As an initial test of the nanobody system, the killswitch was recruited to several endogenous condensates including nucleoli, heterochromatin bodies and nuclear speckles in cultured cells. Nucleoli are multilayered condensates that consist of a granular component, a dense fibrillar component and a fibrillar centre, and are involved in rRNA transcription and ribosome biogenesis[36]. We generated HCT-116 human cell lines in which a GFP tag was knocked into the endogenous *NPM1* and *TCOF1* loci, to GFP label the granular component and fibrillar centre, respectively. The cell lines were transfected with nanobody–killswitch (nb–KS) expression vectors. FRAP experiments revealed substantially slower recovery after photobleaching of both NPM1 and TCOF1 in the cells that expressed nb–KS compared with untransfected cells, cells that expressed the nanobody alone or cells that expressed the nanobody fused to the Phe-to-Ala killswitch mutant (KS; Fig. 2e,f). As expected for specific targeting, the effect in the FRAP assay correlated with the amount of mCherry marker expressed in the corresponding cell (Extended Data Fig. 3d,e). The results were corroborated with three additional nanobodies (anti-ALFA, anti-V5, anti-VHH05) targeting ectopically tagged GFP–NPM1 protein (Extended Data Fig. 4a–d). Immobilization of condensate-forming proteins was also observed in multiple human cell lines in which nuclear speckles were labelled with a GFP tag on endogenous SRRM2, a key component of speckles[37] (Fig. 2d–f and Supplementary Fig. 3a–f); and in mouse embryonic stem (mES) cells, in which chromocentres were labelled with a GFP tag on endogenous HP1α–an important heterochromatin protein[38] (Fig. 2d–f and Supplementary Fig. 3g–i). In the case of chromocentres, two copies of the killswitch fused to the nanobody were necessary to elicit an effect (Fig. 2d–f and Supplementary Fig. 3g–i). In the case of nuclear speckles, an effect was observed with a single copy of the killswitch in HCT-116 cells, and the effect of two tandem killswitch copies was more prominent (Fig. 2d–f and Supplementary Fig. 3d–f). Over time, SRRM2–GFP collapsed into large, dense, spherical foci in cells expressing the killswitch (Fig. 2d and Supplementary Fig. 3a,d). Two tandem copies of the killswitch also led to a gradual demixing of GFP–NPM1 into dense foci (Fig. 2d). These results suggest that the killswitch reduces the dynamics of (that is, immobilizes) various proteins in nucleoli, nuclear speckles in human cells and chromocentres in mouse stem cells.

## The killswitch alters nucleolar composition

Investigating the consequences of immobilizing NPM1 in nucleoli with the killswitch revealed several insights. NPM1 is known as an essential scaffolding protein of the granular component of nucleoli, where ribosome assembly occurs[39,40]. To map changes in the composition of nucleoli targeted with the killswitch, we developed a fluorescence-activated cell sorting (FACS)-based method[41] to isolate nucleoli directly from cell lysates, termed nuclear fluorescence-activated non-membrane condensate isolation (NuFANCI) (Fig. 3a and Supplementary Fig. 4). Mass spectrometry (MS) was performed on nucleoli isolated from GFP–NPM1-encoding cells expressing nb–KS constructs. Comparison of the proteome of the isolated nucleoli with previous reference data[42–44] confirmed strong enrichment of known nucleolar proteins in our samples (Fig. 3b and Extended Data Fig. 5a–f). Comparative analysis revealed 20 proteins that were significantly depleted (≥2-fold, $P \leq 0.01$)

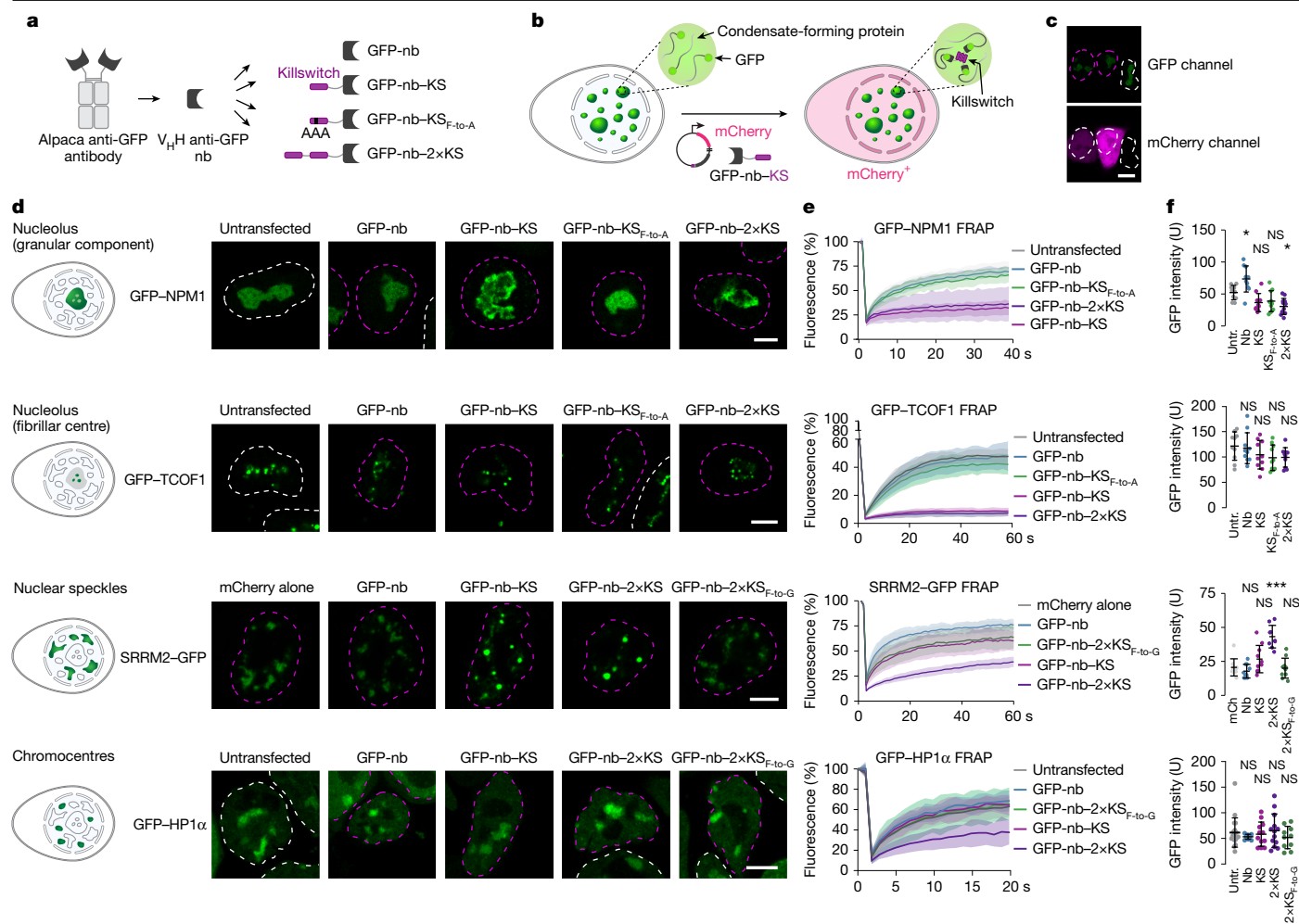

**Fig. 2 | The killswitch arrests the dynamics of endogenous cellular condensates. a**, Schematic of the nanobody–killswitch protein constructs. **b**, Schematic of the nanobody-based recruitment system in live cells. **c**, Example of the nucleus-highlighting strategy. The nuclear contour of mCherry (mCh)-expressing cells is highlighted with a magenta dashed line. The nuclear contour of untransfected (untr.) or mCherry⁻ cells is highlighted with a white dashed line throughout the figure. The experiment was repeated independently three times with similar results. Scale bar, 10 μm. **d**, Live-cell fluorescence microscopy images of cells expressing GFP–NPM1, GFP–TCOF1, SRRM2–GFP and GFP–HP1α transfected with the indicated GFP-nb or GFP-nb–KS constructs. The GFP in each cell line is knocked into the endogenous gene locus. Scale bars, 10 μm. **e**, FRAP analysis of the GFP in the condensates shown in **d**. Data are mean ± s.d.

*n* values are the same as described in **f**. **f**, Control quantification of GFP intensity in the bleached condensates. Data are mean ± s.d. One condensate for each cell was bleached, and one dot represents the value for one condensate. For the NPM1 plot, *n* = 10 cells for all cases, except for 2×KS (*n* = 12 cells), from one experiment. For the TCOF1 plot, *n* = 10 cells for all cases from two biologically independent experiments. For the SRRM2 plot, *n* = 10 cells for all cases, except for 2×KS (*n* = 9), from one experiment. For the HP1α plot, *n* = 19 (untransfected), 10 (Nb and 2×KS_F-to-G), 12 (KS) and 13 (2×KS) cells from one experiment. *P* values were calculated using one-way ANOVA followed by Tukey's post hoc test versus untransfected cells or mCherry alone; exact values are provided in the 'Statistics and reproducibility' section.

from nucleoli targeted with the killswitch over the control cells, and the proteins were enriched for RNA-binding proteins (*P* < 0.05, g:GOS test) (Fig. 3c, Extended Data Fig. 5g,h and Supplementary Table 2). Depletion of NEPRO, a known nucleolar interaction partner of NPM1, was confirmed with immunostaining (Fig. 3d–f). The compositional change of nucleoli was associated with changes in component mobility and function. For example, we observed a substantial decrease in the mobility of mCherry–RPL18 measured with FRAP in cells in which GFP–NPM1 was targeted with the killswitch, but no change in the mobility of mCherry–SURF6 was observed and the level of neither protein was affected (Fig. 3g and Supplementary Fig. 5a–f). Moreover, the NPM1-dense nucleolar regions demixed from 5.8S rRNA, suggesting a potential functional defect (Fig. 3h,i and Supplementary Fig. 6a,b). Finally, when added to the cell culture medium, killswitch that was conjugated to a ten-residue oligoarginine micropeptide (R10) was taken up by cells, localized to the nucleolus as expected[45] and killed cells under conditions in which the R10-alone or R10–KS_F-to-G controls

did not (Extended Data Fig. 6a–e). Cell death was associated with immobilization of GFP–NPM1 (Extended Data Fig. 6f–h). Collectively, these results revealed selective changes in nucleolar composition, component mobility and function in nucleoli when NPM1 is immobilized with the killswitch. To confirm that these changes are linked to changes in the material property of nucleoli, we reconstituted model nucleoli using purified, recombinant, GFP-tagged NPM1 in vitro[39,46]. The killswitch was targeted to GFP–NPM1 droplets with the R10 micropeptide system. We found that targeting the killswitch to GFP–NPM1 droplets significantly reduced the speed of droplet fusion (Extended Data Fig. 7a–f), suggesting that the killswitch alters the material property of in vitro-assembled nucleoli.

## Targeting fusion-oncoprotein condensates

Condensate alterations have been linked to hundreds of human diseases, notably, cancers driven by fusion oncoproteins and rare genetic

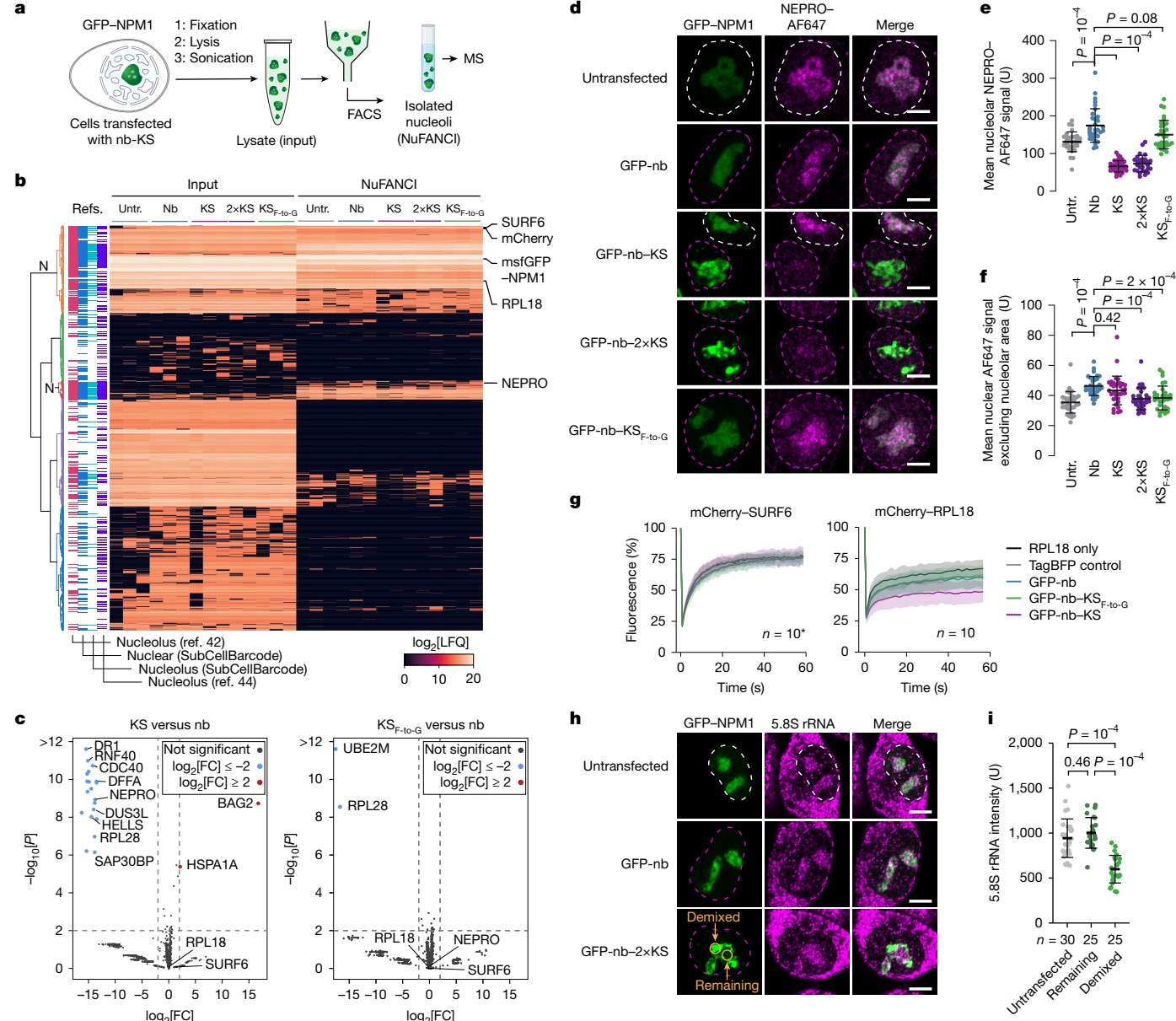

**Fig. 3 | The killswitch alters the composition, component dynamics and function of the nucleolus. a**, Schematic of the NuFANCI procedure. **b**, Protein expression profiles of NuFANCI-isolated nucleoli and their respective input samples. In the left four columns, nucleolar proteins are annotated based on previous proteomics data. N, two clusters of nucleolar proteins. LFQ, label-free quantification. **c**, The proteomes of GFP-nb–KS-targeted nucleoli and GFP-nb–KS$_{F-to-G}$-targeted nucleoli compared with the anti-GFP-nanobody control. *P* values were calculated using one-sided Student's *t*-tests. FC, fold change. **d**, Fixed-cell immunofluorescence images of GFP–NPM1-expressing cells that were untransfected or transfected with GFP-nb, GFP-nb–KS, GFP-nb–2×KS or GFP-nb–KS$_{F-to-G}$ constructs. Scale bars, 5 μm. **e**, NEPRO fluorescence intensity in nucleoli. Data are mean ± s.d. *P* values were calculated using one-way ANOVA followed by Dunnett's T3 multiple-comparison test versus the GFP-nb condition. *n* = 32 (untransfected), 33 (GFP-nb), 36 (GFP-nb–KS), 29 (GFP-nb–2×KS) and 31 (GFP-nb–KS$_{F-to-G}$) cells from two independent experiments. **f**, NEPRO fluorescence intensity in nuclei. Data are mean ± s.d. *P* values were calculated using one-way

ANOVA followed by Dunnett's T3 multiple-comparison test versus the GFP-nb condition. The numbers of cells are the same as described in **e** and are from two independent experiments. **g**, FRAP analysis of mCherry-signal in HCT-116 cells expressing GFP–NPM1 from the endogenous locus after transfection with the indicated nb constructs and co-transfection with mCherry–SURF6 or mCherry–RPL18. Data are mean ± s.d. *n* = 10 for all, except for the SURF6 experiment for the TagBFP-only sample (*n* = 8; asterisk). **h**, Fixed-cell immunofluorescence images of 5.8S rRNA in GFP–NPM1-expressing cells that were transfected with either the GFP-nb or GFP-nb–2×KS construct. Scale bars, 5 μm. **i**, Quantification of 5.8S rRNA mean fluorescence intensities in the demixed and remaining portions of nucleoli in GFP-nb–2×KS-expressing cells. Data are mean ± s.d. *n* represents the number of cells from one experiment. The experiment was repeated twice with similar results. *P* values were calculated using one-way ANOVA followed by Tukey's post hoc test. Some elements in the scheamtic in **a** were created in BioRender. Hnisz, D. (2025) https://BioRender.com/fa5zmne.

diseases[30–32]. We therefore tested the effect of recruiting the killswitch to multiple disease-specific condensates.

Around 2,000 fusion transcripts that arise from cancer-associated chromosomal rearrangements are predicted to produce cancer-specific proteins that may form condensates[32]. Recent studies have provided

insights into the cellular functions of a few fusion oncoprotein condensates; however, the functions and pathomechanism for most of them are yet to be identified[47]. We therefore tested the impact of recruiting the killswitch to well-characterized fusion oncoproteins known to form condensates. In these experiments, GFP-tagged fusion oncoproteins

were co-expressed with nb–KS constructs. For all eight fusion onco-protein condensates tested (EWS::FLI1, SS18::SSX2, CRTC1::MAML2, NUP98::HOXA9, NUP98::DDX10, NONO::TFE3, TAZ::CAMTA1 and PAX3::FOXO1), nb–KS had a negligible effect on the formation of condensates, but significantly reduced FRAP (Fig. 4a–c). Consistent results were observed when the killswitch was genetically fused to the fusion oncoproteins (Extended Data Fig. 8a–c). These results suggest that the killswitch altered the material properties of a diverse of condensate-forming fusion oncoproteins.

We selected two fusion-oncoprotein condensates for further analyses, and found specific changes in the condensate composition, component dynamics and cellular function elicited by the kills-witch. BRD4::NUT is a fusion between the bromodomains of BRD4 and the intrinsically disordered region of NUT, and is a key oncogenic driver in NUT-midline carcinoma[48]. The NUT portion of the fusion recruits p300 that catalyses the acetylation of histones that are the substrates of the bromodomains. This process results in the forma-tion of large BRD4::NUT condensates that can incorporate around 2 Mb of genomic DNA[49,50]. Consistent with this model, BRD4::NUT fusion-oncoprotein condensates were enriched for H3K27Ac in immunofluorescence experiments[50] (Extended Data Fig. 8d,e). RNA polymerase II (RNAPII) was enriched in BRD4::NUT condensates; how-ever, RNAPII enrichment was substantially reduced in condensates formed by the BRD4::NUT–KS protein (Fig. 4d), while co-localization with H3K27Ac was unaltered (Extended Data Fig. 8d,e). The deple-tion of RNAPII was confirmed in a cell-based artificial condensate model system[51] (Extended Data Fig. 8f–i). Fusion of the killswitch to BRD4::NUT also reduced the dynamics of ectopically expressed mCherry–p300 in the condensates (Extended Data Fig. 9a–c). The reduction in RNAPII enrichment in BRD4::NUT–KS condensates was associated with reduced transcription of target genes measured with RNA-sequencing (RNA-seq; Fig. 4e and Extended Data Fig. 9d–g). Treatment of the cells with 1,6-hexanediol dissolved BRD4::NUT but not BRD4::NUT–KS condensates, suggesting that the material proper-ties of the condensates may be affected by the killswitch (Extended Data Fig. 9h,i). These results indicate that the killswitch inhibits RNAPII partitioning and dynamics of p300 in BRD4::NUT condensates, which results in reduced transcriptional activity.

As a second case study of fusion-oncoprotein condensates, we focused on NUP98-fusion proteins that are important drivers in acute myeloid leukaemia (AML)[52]. In these leukaemias, the intrinsi-cally disordered region of the nuclear pore protein NUP98 is fused to domains of various nuclear proteins that are often involved in chromatin modification and transcriptional control, for example, HOXA9, DDX10, NSD1 and KDM5A[53,54]. In contrast to endogenous NUP98, oncogenic NUP98 fusion proteins are not associated with the nuclear pore complex and, instead, form nuclear condensates[53,54]. To measure dynamic changes in condensates, FRAP experiments were performed on mCherry-tagged CRM1 (XPO1), a known client pro-tein enriched in NUP98::DDX10 condensates[54]. We observed slower recovery of mCherry–CRM1 fluorescence in cells expressing GFP–NUP98::DDX10 fused to the killswitch (Extended Data Fig. 10a–c). To directly test whether reduced CRM1 mobility is caused by altered material properties of NUP98::DDX10, we designed mutants in which we clustered GLFG motifs, because clustering of aromatic residues has been linked to arrested condensate dynamics[55]. Similar to the effect of the killswitch, clustering of GLFG motifs in NUP98 indeed reduced the mobility of both GFP–NUP98::DDX10 and CRM1 in condensates (Mut2) (Extended Data Fig. 10d–g). Treatment of the cells with 1,6-hexanediol dissolved NUP98::DDX10 but not NUP98::DDX10–KS condensates, further confirming that the material properties of the condensates were affected by the killswitch (Extended Data Fig. 10h,i). These results suggest that the killswitch alters the microenvironment of oncoprotein condensates, which affects the composition and dynamics of client proteins within the condensates.

## The killswitch blocks mouse leukaemia

Finally, we tested whether the killswitch can reveal insights into cellular activities associated with the disease-specific condensates in vivo. We previously established a mouse AML model, in which fetal liver-derived haematopoietic stem cells (HPSCs) are transformed through expres-sion of the NUP98::KDM5A fusion oncoprotein and subsequent trans-plantation into recipient mice. NUP98::KDM5A-dependent AML cells are then isolated from the bone marrow[54,56] (Fig. 4f). To investigate the effect of the killswitch on oncogenic NUP98::KDM5A condensates, we introduced doxycycline-inducible GFP-nb–KS and GFP-nb–KS$_{F-to-A}$ constructs in a stable AML cell line in which NUP98::KDM5A is tagged with an N-terminal GFP tag (Fig. 4f). The vector also contained the bicis-tronic mCherry reporter (Fig. 4f). Growth curves of mCherry-sorted cells revealed that the doxycycline-induced expression of the nb–KS essentially blocked proliferation while the nb–KS$_{F-to-A}$ mutant did not have such an effect (Fig. 4g). The inhibitory effect of the killswitch was also observed in a competition-based proliferation assay (Sup-plementary Fig. 7a–c). In a second approach, genetic fusion of the kills-witch to GFP–NUP98::KDM5A impaired the transformation of mouse fetal liver-derived primary HPSCs into leukaemic cells as measured by reduced replating capacity, altered immunophenotype and reduced expression of NUP98::KDM5A-target genes (Supplementary Fig. 8a–e). These results suggest that the killswitch is sufficient to inhibit the growth of leukaemia cells of which the proliferation is dependent on NUP98::KDM5A condensates.

To investigate whether the proliferation defect is caused by altera-tions in NUP98::KDM5A condensates, we performed fluorescence microscopy. Notably, expression of the nb–KS—assessed by the appear-ance of mCherry in the cells—led to an almost instantaneous reduction in the number of NUP98::KDM5A condensates and the level of the fusion oncoprotein, whereas the KS$_{F-to-A}$ variant had no such effect (Fig. 4h,i and Supplementary Fig. 9a–c). Short-term treatment (3 h) of the cells with proteasome inhibitors partially rescued the expression level of the fusion protein in cells expressing nb–KS, although the NUP98::KDM5A protein formed large amorphous bodies (Fig. 4j and Supplementary Fig. 10a,b). Despite multiple attempts, assessment of the material properties of the oncofusion condensates with FRAP in the AML cells failed owing to the low-level expression of the protein. Thus, to gain further insights into the effect of the killswitch on NUP98::KDM5A con-densates, we transfected the fusion protein and the nb–KS constructs into HEK293T cells. FRAP experiments confirmed that the killswitch arrested the dynamics of NUP98::KDM5A condensates (Fig. 4k and Supplementary Fig. 10c–h). These results suggest that the antiprolif-erative effect of the killswitch in NUP98::KDM5A condensate-driven AML cells may be associated with altering the material properties of NUP98::KDM5A condensates. Furthermore, the unexpected finding of rapid proteasome-dependent degradation of the fusion oncoprotein after killswitch expression revealed that leukaemia cells do not toler-ate any perturbation of cancer-driving fusion oncoprotein-containing condensates.

## Targeting adenoviral nuclear condensates

As another example of disease-specific condensates, we targeted the killswitch to adenoviral nuclear bodies. Adenoviruses encode a packag-ing protein called 52K that forms nuclear condensates that are essential for the assembly of infectious viral particles from viral genomes and structural proteins[57]. We first tested whether the killswitch can arrest the dynamics of 52K condensates by performing FRAP after transient expression of GFP-tagged 52K protein fused to the killswitch. Wild-type 52K condensates recovered nearly all fluorescence within seconds after photobleaching (Fig. 5a–c and Supplementary Fig. 11a–c). By contrast, almost no recovery was observed when bleaching GFP–52K–KS con-densates, and fluorescence recovery was mostly rescued when using

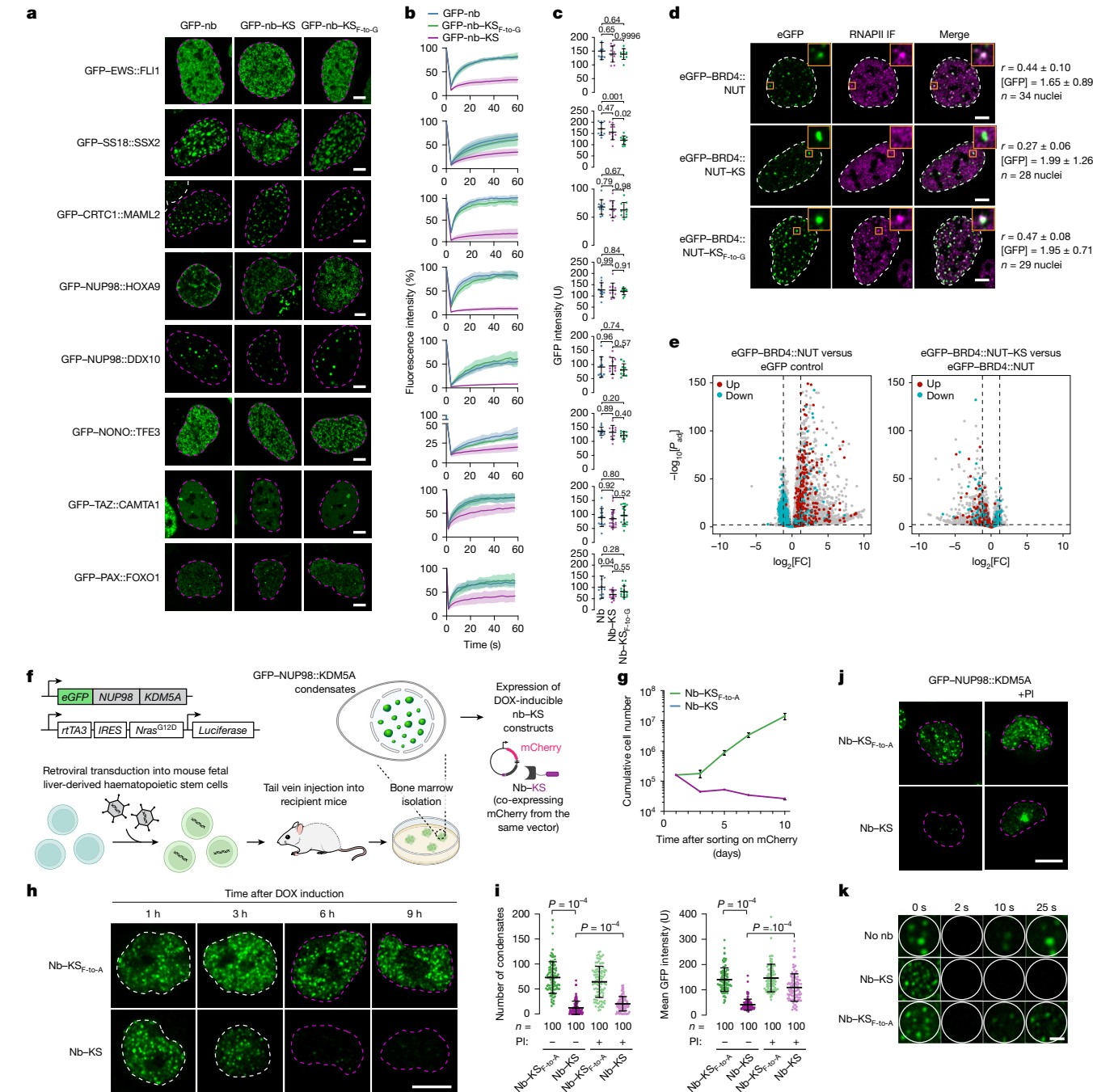

**Fig. 4 | The killswitch alters the compositions and functions of oncoprotein condensates. a**, Live-cell fluorescence microscopy images of U2OS cells expressing the indicated fusion oncoproteins tagged with GFP. GFP–EWS::FLI1 was imaged in MCF7 cells. One nucleus is shown in each image. The cells were co-transfected with a GFP–nb, GFP–nb–KS or GFP–nb–KS$_{F-to-G}$ construct. Scale bars, 5 µm. **b**, FRAP analysis of GFP-fusion-oncoprotein condensates. Data are mean ± s.d. *n* values are as described in **c**. **c**, Quantification of GFP intensity in the bleached condensates. *n* = 10 cells, except for TAZ::CAMTA1 (*n* = 14, 19 and 19 cells) and PAX::FOXO1 (*n* = 11, 14 and 15 cells) for GFP–nb, GFP–nb–KS and GFP–nb–KS$_{F-to-G}$, respectively, from two independent experiments. Data are mean ± s.d. *P* values were calculated using one-way ANOVA followed by Tukey's post hoc test. **d**, Fixed-cell immunofluorescence images of RNAPII in cells expressing GFP–BRD4::NUT, GFP–BRD4::NUT–KS, or GFP–BRD4::NUT–KS$_{F-to-G}$. Scale bars, 5 µm. IF, immunofluorescence; *r*, Pearson correlation coefficient. **e**, Differential gene expression analyses. Known BRD4::NUT targets are coloured. *P* values were determined using the Benjamini–Hochberg method. **f**, Schematic of the generation of the GFP–NUP98::KDM5A-expressing AML model. **g**, Growth curves of primary mouse AML cells that were transduced with nanobody constructs

co-expressing mCherry. The cumulative cell number was recorded after sorting mCherry-positive cells. Data are mean ± s.d. *n* = 3 biologically independent experiments. **h**, Fluorescence microscopy images of GFP–NUP98::KDM5A condensates in primary mouse AML cells expressing doxycycline (DOX)-inducible nb–KS$_{F-to-A}$ or nb–KS. The experiments were repeated independently three times with similar results. Scale bar, 5 µm. **i**, The number of condensates and the mean GFP intensity in the AML cells expressing the indicated nanobody constructs. Data are mean ± s.d. *P* values were calculated using unpaired two-tailed *t*-tests. *n* = 100 cells in all cases examined over three biologically independent experiments. PI, proteasome inhibitors. **j**, Representative images showing proteasome inhibition partially rescues the GFP–NUP98::KDM5A level in cells expressing nb–KS. The experiments were repeated independently three times with similar results. Scale bar, 5 µm. **k**, Representative FRAP images of GFP–NUP98::KDM5A nuclear condensates in HEK293T cells. The experiments were repeated independently three times with similar results. Scale bar, 1 µm. Some of the elements in **f** were created in BioRender. Hnisz, D. (2025) https://BioRender.com/fa5zmne.

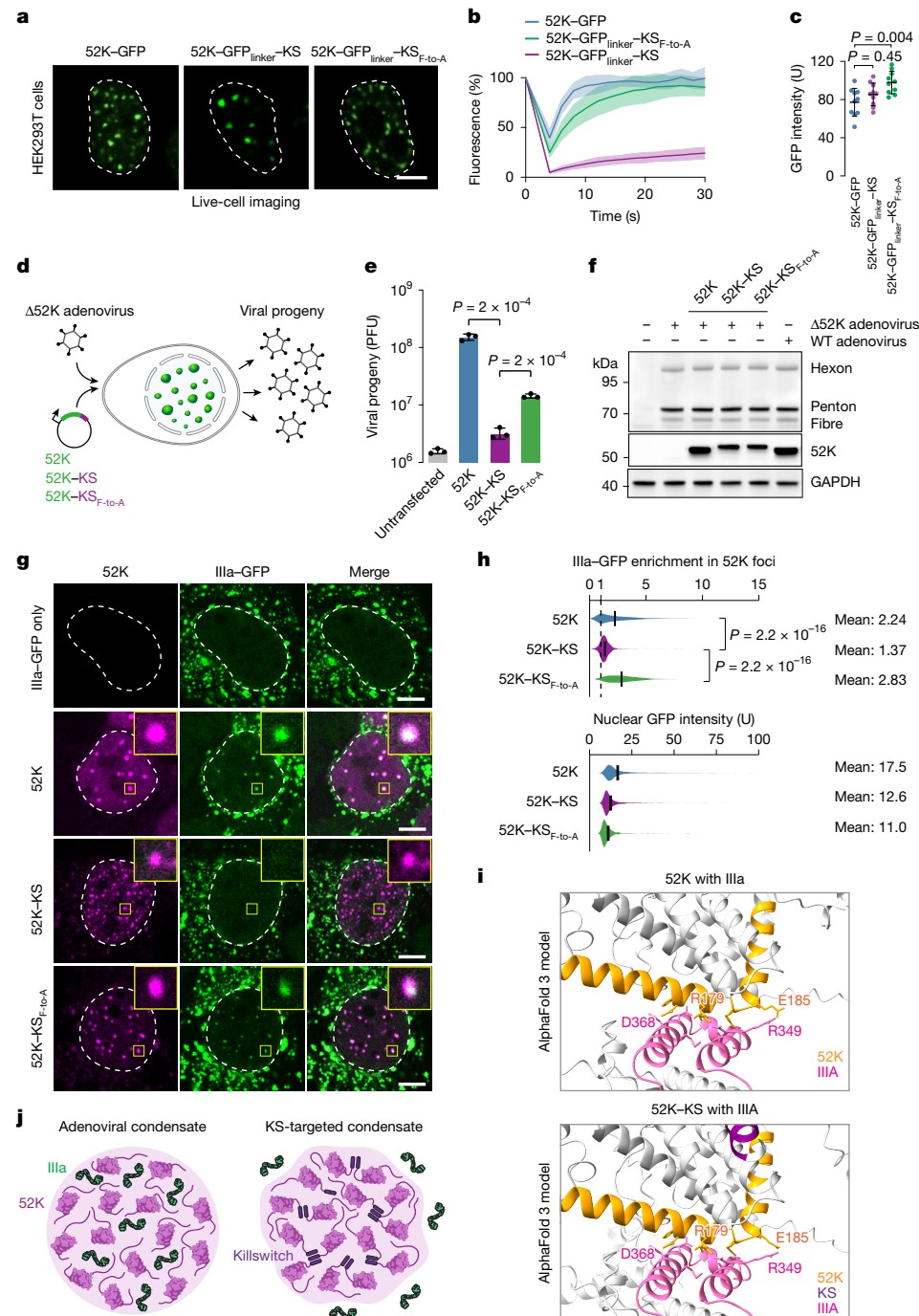

**Fig. 5 | The killswitch inhibits adenoviral condensate dynamics, partitioning of capsid proteins and viral particle assembly. a**, Live-cell fluorescence microscopy images of HEK293T cells expressing the indicated 52K proteins. The nucleus is highlighted with a dashed white line contour. Scale bar, 5 µm. **b**, FRAP analysis of GFP in the condensates shown in **a**. Data are mean ± s.d. $n = 10$ in all cases. **c**, Quantification of GFP intensity in the bleached condensates. Data are mean ± s.d. $n = 10$ cells in all cases from two biologically independent experiments. $P$ values were calculated using one-way ANOVA followed by Dunnett's multiple-comparison test versus 52K–GFP. **d**, Schematic of the 52K complementation experiment. HEK293T cells were infected with Δ52 adenovirus, and then transfected with plasmids encoding 52K proteins. **e**, Viral progeny production was measured as plaque-forming units (PFU). Data are mean ± s.d. $n = 3$ biological replicate experiments. $P$ values were calculated using unpaired two-tailed $t$-tests. **f**, Western blot showing similar levels of 52K proteins and viral structural proteins in the complementation experiment. The experiment was repeated independently twice with similar results. **g**, The killswitch inhibits

partitioning of IIIa into 52K condensates. Representative immunofluorescence images are shown of 52K and IIIa–GFP in HEK293T cells. Nuclei are highlighted with a dashed white line contour. Insets: magnified images of the highlighted condensates. The experiment was repeated independently twice with similar results. Scale bars, 5 µm. **h**, Quantification of the partitioning of IIIa–GFP into 52K condensates and the mean IIIa–GFP fluorescence. $n = 1,583$ (52K), $n = 2,518$ (52K–KS) and $n = 651$ (52K–KS_{F-to-A}). $P$ values were calculated using one-way ANOVA followed by Tukey's post hoc test versus 52K–KS. The solid black lines represent the mean. **i**, AlphaFold model of the interaction interface between 52K and IIIa. The interacting structural elements are coloured (yellow, 52K; magenta, IIIa). Contact probability matrices are shown in Supplementary Fig. 12. **j**, The condensate microenvironment model, and the effect of the killswitch on the condensate microenvironment. The partitioning of IIIa into 52K nuclear bodies is inhibited. Some of the elements in **j** were created in BioRender. Hnisz, D. (2025) https://BioRender.com/fa5zmne.

the $KS_{F-to-A}$ mutant (Fig. 5a–c and Supplementary Fig. 11a–c). These results suggest that the killswitch arrests the dynamics of adenoviral 52K condensates.

The impact of the killswitch on the function of adenoviral 52K condensates was tested in a complementation assay. HEK293T cells were infected with Δ52K mutant adenovirus and then were either left untransfected or were transfected with a vector encoding wild-type and killswitch-tagged 52K variants. Infected cells were collected for analysis at 48 h after infection, and infectious progeny production was measured in focus-forming units (Fig. 5d). We note that we used the genetic killswitch fusion in the complementation assay, because fixed-cell immunofluorescence suggested that the GFP tag had a slight effect on condensate formation by the 52K protein (Supplementary Fig. 11a,d). The killswitch had a considerable effect in the complementation assay, suppressing the production of viral particle production by more than 90% (Fig. 5e), a substantial portion of which was rescued by the $KS_{F-to-A}$ variant, suggesting that the suppression of viral particle production was at least in part due to the killswitch (Fig. 5e). As a key control, western blot analysis confirmed that viral structural proteins and the 52K variants were expressed at similar levels across all conditions (Fig. 5f). To investigate the mechanistic basis of the ability of the killswitch to suppress viral particle assembly, we assessed recruitment of adenovirus minor capsid protein IIIa into condensates, as recruitment of IIIa is known to be critical for progeny production[57]. Significantly less IIIa–GFP was detected in 52K condensates that were tagged with the killswitch, compared with the wild–type 52K condensates and 52K condensate tagged with the $KS_{F-to-A}$ variant (Fig. 5g,h and Supplementary Fig. 11e,f), while the mean nuclear concentration of IIIa–GFP was similar (Fig. 5h). These results were surprising, because the portion of 52K that is necessary for interaction with IIIa is known and present in the experiment[57], and the interaction interface between the two proteins is unaffected by the presence of the killswitch when modelled with AlphaFold 3 (Fig. 5i and Supplementary Fig. 12a–e). The data therefore suggest that IIIa partitions into 52K condensates facilitated by the microenvironment of 52K condensates rather than merely by stoichiometric interactions (Fig. 5j). Taken together these results revealed that the killswitch arrests the dynamics of 52K adenoviral condensates, and inhibits the production of viral particles, which is associated with reduced partitioning of the viral structural protein IIIa into 52K condensates.

## Targeting transcriptional bodies in vivo

Finally, we tested whether the killswitch can affect condensate function in a multicellular embryo in vivo. Cells in early zebrafish embryos contain transcriptional bodies that have key roles in transcribing two large clusters of microRNA loci. The bodies can be visualized by injecting RNA encoding mNeonGreen-tagged NANOG, and the microRNA native transcripts can be visualized with in vivo imaging[58,59]. Using this system, we found that targeting the killswitch to transcription bodies by fusing it to NANOG significantly reduced microRNA transcription in transcription bodies (Extended Data Fig. 11a–e).

## Discussion

Here, we present a micropeptide killswitch as a principal method that complements biochemical and genetic perturbation to investigate the functions of biomolecular condensates. The killswitch is unique in that it has a controllable and titratable effect on the material properties of a variety of endogenous condensates in living cells. We found that the killswitch immobilized constituent proteins of nucleoli, nuclear speckles, chromocentres, fusion oncoprotein condensates and adenoviral nuclear bodies. We note that we did not find evidence that the killswitch affects the mobility of the soluble fraction of the targeted protein in the systems we tested (Extended Data Fig. 12a–g), although this remains a possibility. For the condensates tested here, the

killswitch altered condensate composition, component dynamics and function. The changes appeared to be selective to certain condensate components, suggesting that various proteins probably partition into condensates through different underlying molecular features. The killswitch had greater effects on some condensates (such as NUP98-fusion condensates) and marginal effect on others (such as chromocentres), revealing differences in the nature of the interactions between the condensate-forming proteins in the various condensates.

One of the most intriguing predictions about biomolecular condensates is that they create microenvironments that facilitate partitioning of other biomolecules into the condensates in a manner not explained by stoichiometric interactions[4,11,12,17]. Current empirical evidence for this model includes measuring concentrations of small molecules that exceed the number of potential binding sites mostly within artificially assembled condensates[8,10–13] and, most recently, demonstrating a pH gradient within nucleolar condensates in cell lysates[60]. Using the killswitch, we provide evidence for unique microenvironments within multiple condensates in live cells. For example, the killswitch altered the composition and inhibited the dynamics of client proteins in nucleoli, oncoprotein condensates and viral nuclear bodies, revealing insights into their in vivo functions. We emphasize that the killswitch affected protein composition of condensates even though all of the interacting regions in the native proteins were present. We anticipate that the killswitch will facilitate future insights into the compositional biases of condensates in relation to their functions, enable the distinction of the set of proteins residing in condensates from their soluble pool, facilitate investigation of compositional and functional specificity of diverse condensates and, ultimately, accelerate manipulation of the material properties of condensate-forming proteins as potential therapeutics.

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

## Methods

### Generation of DNA constructs

A list of all of the oligonucleotides that were used to generate plasmids and used for quantitative PCR with reverse transcription (RT–qPCR) in this study is provided in Supplementary Table 1.

**Generation of mammalian expression vectors for expression of GFP-fused HMGB1 and NPM1.** Mammalian expression vectors pRK5-mEGFP-HMGB1-WT, -mutant and mutant-patchless were prepared in a previous study[31] (Addgene, 194548, 194550 and 194553, respectively). To generate fsHMGB1-full-length with KS variants C16-to-A, M-to-E&D and 3F-to-E&D (Fig. 1e), cDNA fragments were ordered from Twist Biosciences, PCR amplified and assembled into BsrGI + SalI-digested pRK5-mEGFP-HMGB1-mutant plasmid using the NEBuilder HiFi DNA assembly master mix (NEB, E2621). Other KS variants (F-to-A, F-to-G, 0F, F12A, F13A, ΔKS 3F, 11G3F3G, KS Shuffled01–10 and ΔKS-TDP43 HP) were constructed by amplifying pRK5-mEGFP-HMGB1-mutant vector with primers flanking the KS sequence, and KS variant inserts were generated as a single-stranded oligonucleotide that was used to bridge double-stranded, PCR-amplified vector using the NEBuilder HiFi DNA assembly master mix.

To generate PiggyBac carrier vectors, N-terminally eGFP-tagged coding sequence of human *NPM1* (Addgene, 131818)[61] was cloned into the backbone of the inducible Caspex expression vector (Addgene, 97421)[62] linearized by restriction digest with NcoI (NEB, R0193) and BsrGI (NEB, R3575). The eGFP sequence was cloned from same inducible Caspex expression vector. The KS variants were introduced into the C terminus of NPM1 through the overhang regions of the reverse PCR primers (Extended Data Fig. 2e).

pRK5-NPM1-WT and -KS plasmids were constructed by amplifying eGFP-NPM1-WT and -KS DNA fragments from PiggyBac carrier vectors described above and the fragments were assembled into AgeI + SalI-digested (NEB, R3552 and R0138, respectively) pRK5 backbone (mEGFP-HMGB1-WT) using the NEBuilder HiFi DNA assembly master mix.

pRK5-mEGFP-NPM1-TDP43-HP plasmid was constructed by amplifying the *NPM1* sequence from Addgene plasmid 131818 and cloned using the NEBuilder HiFi DNA assembly into a backbone of the pRK5-meGFP (Addgene, 18696)[63] linearized by restriction digest with BsrGI and AccI (NEB, R0161). The TDP43-HP sequence (amino acids 318–343) was introduced into the C terminus of *NPM1* through the overhang regions of the reverse PCR primer (Supplementary Fig. 2f).

**GFP-nb–KS expression vectors.** Initial design of GFP-nb expression constructs included mCherry fused to SV40-NLS-(GGGGS)×2-linker-GFP-nb-(GGGGS)×2 linker, followed by 0–2 repeats of killswitch peptide in the pRK5 vector backbone. To generate this, mCherry was amplified from the pET45-mCherry-NPM1 vector (Addgene, 194546)[31], cDNA sequences for SV40-NLS-(GGGGS)×2-linker-GFP-nb-(GGGGS)×2-(0–2 repeats of killswitch) were ordered from Twist Biosciences. The LaG-16 variant of GFP-nb[35] (Addgene, 128788)[64] was used in this study. DNA fragments were PCR-amplified and assembled into AgeI + SalI-digested pRK5-mEGFP-HMGB1 plasmid (Addgene, 194550) using the NEBuilder HiFi DNA assembly master mix.

As expression mCherry–GFP-nb fusion protein had unintended effects on nucleoli (Extended Data Fig. 3a), two cleavage sites (P2A and T2A) were introduced between mCherry and GFP-nb to ensure minimal levels of fusion protein while enabling the use of mCherry fluorescence as reporter for GFP-nb expression level. Unless stated otherwise, nanobody construct with cleavable mCherry was used in experiments using GFP-nb. Moreover, HA tag was introduced between T2A and SV40-NLS so that localization of GFP-nb into target condensate could be verified with immunofluorescence (Extended Data Fig. 3c). To generate these expression vectors, pRK5-mCherry-GFP-nb vector was PCR-amplified using primers that introduce HA tag before SV40-NLS, and an insert containing GSG-P2A-GSG-T2A sequence was generated as single-stranded oligonucleotide that was used to bridge double-stranded PCR-amplified pRK5-mCherry-GFP-nb vector using the NEBuilder HiFi DNA assembly master mix. pRK5-mCherry-P2A-T2A-HA-tag-SV40-NLS-GFP-nb-1×KS, -2×KS, -1×KS$_{F-to-A}$ and -1×KS$_{F-to-G}$ vectors were generated by amplifying mCherry-P2A-T2A-HA-tag sequence from vector described above and GFP-nb variant sequences from previously prepared pRK5-mCherry-GFP-nb vectors, and fragments were assembled into AgeI + SalI-digested pRK5 vector backbone using the NEBuilder HiFi DNA assembly master mix. Vectors with 2×KS$_{F-to-A}$ and 2×KS$_{F-to-G}$ variants were generated by amplifying the vector with primers flanking the KS sequence and 2×KS$_{F-to-A}$ and 2×KS$_{F-to-G}$ sequences were generated as single-stranded oligonucleotide that was used to bridge double-stranded, PCR-amplified vector using the NEBuilder HiFi DNA assembly master mix. For experiments involving mCherry–RPL18 and mCherry–SURF6 (Fig. 3g and Supplementary Fig. 5), mCherry reporter in the GFP-nb construct was replaced by TagBFP. To generate this vector, mCherry and GFP-nb sequences were cut out by digestion with AgeI + SalI, and the TagBFP and P2A-T2A-HA-tag-SV40NLS-GFP-nb sequences were amplified by PCR and assembled back into pRK5 backbone using the NEBuilder HiFi DNA assembly master mix. The TagBFP sequence was a gift from the E. Schulz laboratory.

**Killswitch expression vectors with other nanobodies.** Sequences for nanobodies against V5-tag (Addgene, 201475)[65], ALFA-tag (Addgene, 159986)[66] and VHH05-tag (Addgene, 171570)[67] were ordered as synthetic DNA from Twist Biosciences. mCherry-P2A-T2A-SV40NLS-linker sequences were cloned from previously generated GFP-nb vector and assembled with nanobody sequences with NEBuilder HiFi DNA assembly master mix. Linker and killswitch sequences were introduced to C termini of nanobodies in reverse PCR primers.

V5-tag, ALFA-tag or VHH05-tag was fused to the N terminus of *eGFP–NPM1* by introducing sequences into the forward cloning primers when cloning *eGFP–NPM1* from PiggyBac carrier vector into AgeI- and SalI-digested pRK5 backbone.

**pUC19 repair templates.** Repair templates with msfGFP flanked by 550–900 bp homology arms in the N termini of NPM1, TCOF1, HP1α, EWSR1 and FUS, and the C terminus of SRRM2 were generated into pUC19 vector backbone (NEB, N3041S). msfGFP was amplified from previously generated pRK5-mEGFP-HMGB1 plasmid (Addgene, 194550). DNA fragments for the 5′ and 3′ homology arms of NPM1, HP1α, SRRM2, EWSR1 and FUS were ordered from Twist Biosciences. The homology arms of TCOF1 were amplified from the gDNA of HCT-116. The repair template of each target gene was assembled into pUC19 vector digested with SalI + KpnI (NEB, R3142) or HindIII + BamHI (NEB, R0104 and R0136, respectively) using the NEBuilder HiFi DNA assembly master mix.

**gRNA–Cas9 expression vectors.** gRNAs targeting N or C termini of target genes were cloned into sgRNA-Cas9 expression vector pX458 (Addgene, 48138)[68], from which eGFP was replaced with mCherry. The pX458 backbone was amplified as three separate fragments, T2A-mCherry fragment was amplified from Addgene plasmid 161974[69] and fragments were assembled to generate pX458-mCherry vector using the NEBuilder HiFi DNA assembly master mix. Guide RNAs were cloned into pX458-mCherry vectors using DNA oligos (0.1 nmol) that were first phosphorylated for 30 min at 37 °C in T4 DNA ligase reaction buffer (NEB, M0202) with T4 polynucleotide kinase (NEB, M0201) in a total volume of 10 μl, annealed after 5 min of incubation in 95 °C by cooling down at room temperature for 20 min. The oligo duplex was diluted 1:200 and 1 μl of oligo duplex was ligated into 50 ng of BbsI-digested (NEB, R0539) pX458-mCh plasmid using T4 DNA ligase (NEB, M0202).

**Generation of GFP-fusion-oncoprotein expression vectors.** For expressing the N-terminally mEGFP-tagged fusion oncoprotein constructs (Fig. 4a), the fusion partner sequence of each fusion oncoprotein of *NUP98::HOXA9*, *NUP98::DDX10*, *SS18::SSX1*, *SS18::SSX2*, *NONO::TFE3*, *YAP::MAMLD1*, *TAZ::CAMTA1* and *PAX::FOXO1* were amplified from U2OS cDNA. The cDNA was first amplified using primers without overhang unless stated otherwise in Supplementary Table 1. The PCR products were then amplified as DNA template using primers containing overhang for Gibson assembly reaction. *EWS::FLI1* was cloned from Addgene plasmid 102813 (ref. 70). *CRTC1::MAML2* was cloned from Addgene plasmid 154265 (ref. 71). *NUP98::DDX10* Mut2 was cloned from synthesized NUP98 sequences from Twist Biosciences that were ligated to DDX10 sequence through Gibson ligation. The KS sequence was introduced through the overhang of the reverse PCR primers. The fusion oncoproteins were then cloned using NEBuilder HiFi DNA assembly into a backbone of the pRK5-meGFP (Addgene, 18696) linearized by restriction digest with BsrGI and AccI.

To generate expression plasmids with *eGFP−BRD4::NUT*, the *eGFP* sequence was amplified from pRK5-eGFP-NPM1 plasmid described above and cDNA for *BRD4::NUT* fusion protein was amplified from Addgene plasmid 171630 (ref. 49) as two separate fragments, while generating KS and KS$_{F-to-G}$ fusions in the C terminus with reverse primers. *BRD4::NUT* and *eGFP* fragments were assembled into AgeI + SalI-digested pRK5 vector backbone (pRK5-mEGFP-HMGB1) using the NEBuilder HiFi DNA assembly master mix.

For 1,6-hexanediol experiments, the mCherry-P2A-T2A sequence was fused N-terminally to eGFP−BRD4::NUT constructs by cloning it from previously prepared pRK5-mCh-P2A-T2A nanobody construct.

**pRK5-mCherry constructs.** To generate N-terminal mCherry fusions to SURF6, RPL18 and P300, mCherry sequence was cloned from previously prepared mCherry-P2A-T2A nanobody vectors and GGGGSGGGGS linker C-terminal to mCherry was introduced in the reverse primer. Coding sequences of SURF6, RPL18 and P300 were amplified from HEK293T cDNA and assembled with mCherry sequence into AgeI + SalI-digested pRK5 backbone using the NEBuilder HiFi assembly master mix.

To generate C-terminal mCherry fusion to CRM1, the mCherry sequence was cloned from previously prepared mCherry-P2A-T2A nanobody vectors and GGGGSGGGGS linker N-terminal to mCherry was introduced. The coding sequence of CRM1 was amplified from U2OS cDNA and assembled with mCherry sequence into AgeI + AccI-digested pRK5 backbone using the NEBuilder HiFi assembly master mix.

**Vectors for LacO-LacI tethering assay.** CFP-LacI-MCS plasmid[72] was digested with BamHI and XbaI, a stop codon was introduced to the end of the GAPGSAGSAAGGSAIA linker sequence after CFP-LacI with T4 PNK (NEB, M0201S) phosphorylated primers. CFP-LacI-NUT, -NUT-KS and NUT-KS$_{F-to-G}$ vectors were generated by digesting CFP-LacI-MCS plasmid with AsiSI and BsiWI restriction enzymes and NUT sequences were cloned from pRK5-eGFP-BRD4::NUT vectors and assembled with CFP-LacI backbone using NEBuilder HiFi DNA Assembly.

**Constructs for NUP98::KDM5A experiments.** The GFP−NUP98::KDM5A construct was generated by restriction-enzyme-guided removal of the IRES sequence from a plasmid for constitutive expression of *NUP98::KDM5A* as previously described[56]. For generation of GFP−IRES−NUP98::KDM5A−KS/KS$_{F-to-A}$/KS$_{F-to-G}$ constructs, the KS was amplified from OE_KS/F-to-A/F-to-G-mEGFP and introduced in frame into the GFP−IRES−NUP98::KDM5A vector using Gibson assembly. The TRE3G-mCherry-P2A-nb-KS was cloned into a lentiviral entry vector. The retroviral plasmid pCMV-gag/pol was acquired from Cell Biolabs.

**52K expression vectors.** To generate mammalian expression plasmids containing C-terminally tagged 52K fusion proteins (Fig. 5d−h and Supplementary Fig. 11d), plasmid backbones were linearized by restriction digest, and complete plasmids reassembled from linearized backbone and DNA fragments using Gibson assembly Master Mix (NEB, E2611) according to the manufacturer's guidelines. All restriction enzymes were purchased from New England Biolabs and were compatible with digest reactions in rCutsmart buffer (NEB, B6004S). Gene fragments corresponding to all variants of KS were purchased as double-stranded DNA from Azenta Life Sciences using the FragmentGENE service (Supplementary Table 1). For cloning of GFP-tagged fusion proteins (Fig. 5a and Supplementary Fig. 11a), plasmids were assembled from p52K-GFP[57] linearized by digestion with BsrGI (NEB, R3575) and the corresponding KS gene fragment. For cloning of fusion proteins without GFP tags, p52K[57] was digested with NheI (NEB, R3131) and SalI (NEB, R0138) restriction enzymes to excise the existing 52K open reading frame and linearize the plasmid backbone. An open reading frame encoding 52K, lacking the stop codon and containing complementary sequence required for Gibson assembly was PCR-amplified using specific primers. A three-fragment assembly was performed using digested backbone, PCR-amplified open reading frame and the corresponding KS gene fragment. The mammalian expression plasmid encoding GFP-tagged minor capsid protein IIIa was previously described[57].

**Generation of DNA constructs for protein purification.** For the purification of msfGFP labelled fusion proteins (Fig. 1f, Extended Data Fig. 7 and Supplementary Fig. 1) we amplified the msfGFP sequence from previously generated pRK5-mEGFP-HMGB1 plasmid (Addgene, 194550) for msfGFP, msfGFP−KS, msfGFP−KS$_{F-to-G}$ and msfGFP−NPM1. The *NPM1* sequence was amplified from pHcRed-NPM1wt-C1 plasmid (Addgene, 131818). The amplified gene fragments were cloned into a pET22b-backbone (Addgene, 166439) linearized by restriction digest with PmlI and BsrGI, using the NEBuilder HiFi DNA assembly master mix.

**Generation of DNA constructs for zebrafish experiments.** A plasmid containing full-length *Nanog* mNeonGreen was generated previously[58]. The KS and KS$_{F-to-G}$ control peptide sequences were cloned into the C-terminal end of Nanog-mNeonGreen. These plasmids were used as a template to synthesize RNA using the SP6 mMessage mMachine in vitro transcription kit (Invitrogen AM1340) according to manufacturer's instructions.

## Cell culture

Cells were cultured under standard conditions (37 °C and 5% CO$_2$) in sterile, TC-treated, non-pyrogenic, polystyrene tissue culture dishes (Corning). U2-OS (ATCC, HTB-96) HEK293T (ATCC, CRL-3216), HCT-116 (ATCC, CCL-247), MCF7 (ATCC, HTB-22), C2C12 (ATCC, CRL-1772), Lenti-X 293T (Takara Bio, 632180) and A673 (gifted by H. Kovar; CLS, 300454) cell lines were cultured in DMEM GlutaMAX (Gibco, 31966047). HAP1-SRRM2$^{tr0}$ (ref. 37) and TC71 (DMSZ, ACC516) cells were cultured in IMDM (Gibco, 12440053). H3122 (CLS, 300484) and 1765-92 (gifted by P. Åman) cells were cultured in RPMI 1640 Medium (Thermo Fisher Scientific, 61870036). All media included 10% FBS (Gibco, 10438-026) and 100 U ml$^{-1}$ penicillin−streptomycin (Gibco, 15140148).

For experiments involving expression of 52K and KS variants, HEK293 (ATCC, CRL-1573) and HEK293T (ATCC, CRL-3216) cells were grown in DMEM (Corning, 10-013-CV) supplemented with 10% FBS (VWR, 89510-186) and 1% penicillin−streptomycin (Gibco, 15140-122).

V6.5 mES cells were cultured on irradiated primary mouse embryonic fibroblasts, previously seeded on 0.2% gelatin-coated plates, in KO DMEM (Gibco, 1082901) containing 15% FBS (Gibco, 10438-026), 100 U ml$^{-1}$ penicillin−streptomycin, 1× non-essential amino acids (Gibco, 11140050), 0.05 mM β-mercaptoethanol (Gibco, 21985023) and laboratory-purified recombinant leukaemia inhibitory factor (LIF). The identity of all cell lines were verified using morphological characteristics, but lines have not been authenticated.

All of the cell lines tested negative for mycoplasma using the Look-Out Mycoplasma PCR Detection Kit (Sigma-Aldrich, MP0035) or the PCR Mycoplasma Test Kit II (Applichem, A8994). Mycoplasma testing was performed on 0.2–1 ml of cell culture medium taken from tissue culture dishes containing confluent monolayers of cells on a routine basis at least twice a year.

For experiments involving expression of NUP98::KDM5A and KS variants, mouse fetal liver cells were cultured in DMEM/IMDM (50:50%, v/v, Gibco, life technologies), supplemented with 10% heat-inactivated FBS (Sigma-Aldrich), 100 U ml$^{-1}$ penicillin, 100 µg ml$^{-1}$ streptomycin, 4 mM L-glutamine and 50 µM β-mercaptoethanol (all Gibco, Thermo Fisher Scientific) in the presence of 100 ng ml$^{-1}$ mSCF, 10 ng ml$^{-1}$ mIL-3 and 10 ng ml$^{-1}$ mIL-6 (all PeproTech). Ex vivo-isolated leukaemia cells were cultured in RPMI 1640 (Gibco, Life Technologies), supplemented with 10% FBS, 100 U ml$^{-1}$ penicillin, 100 µg ml$^{-1}$ streptomycin, 4 mM L-glutamine, 100 ng ml$^{-1}$ mSCF and 10 ng ml$^{-1}$ mIL-3. After 1 week, the medium of the ex vivo-isolated cells was switched to RPMI 1640 supplemented with 10% FBS, 100 U ml$^{-1}$ penicillin and 100 µg ml$^{-1}$ streptomycin, 4 mM L-glutamine, 1 mM sodium pyruvate (Sigma-Aldrich), 50 µM 2-mercaptoethanol (Gibco, Thermo Fisher Scientific) and 20 mM 4-(2-hydroxyethyl)−1-piperazineethanesulfonic acid (HEPES) (Sigma-Aldrich). Stable cell lines were established by continuous culture for over 4 weeks and the GFP–NUP98::KDM5A cell line was maintained using RPMI medium. Platinum-E (Cell Biolabs), HEK293T and Lenti-X 293T cells (Takara) were cultured in DMEM (Gibco, Thermo Fisher Scientific) supplemented with 10% FBS, 100 U ml$^{-1}$ penicillin, 100 µg ml$^{-1}$ streptomycin and 2 or 4 mM L-glutamine, respectively. Mouse leukaemia cells and HEK293T cells expressing nanobody constructs were incubated with doxycycline (Sigma-Aldrich, 24390-14-5) at 1 µg ml$^{-1}$ in growth medium to induce the expression. For proteasome inhibition (Fig. 4h–j and Supplementary Fig. 9a,b), murine leukaemia cells expressing the nanobody constructs were initially incubated with doxycycline as specified above for 3 h to induce expression. Cells were then treated with the UBA1 inhibitor TAK-243 (MLN7243) (MedChemExpress, HY-100487) at 0.25 µM, the NAE inhibitor pevonedistat (MLN4924) (MedChemExpress, HY-70062) at 1.25 µM and the proteasome inhibitor MG-132 (MedChemExpress, HY-13259) at 2.5 µM and incubated for 3 h before imaging. All cells were cultured at 37 °C under 5% $CO_2$ and 95% humidity.

## Cell-viability assay
A total of 150,000 cells per well were seeded onto a six-well plate, transfected the next day with 500 ng of pRK5-msfGFP-HMGB1-mutant KS variants using FuGENE HD (Promega, E2311) according to the manufacturer's instructions. GFP-expressing cells were sorted using the FACSAria II instrument (BD) the next day, and 10,000 cells per well were seeded onto 96-well plates and the viability was measured 48 h later using CellTiter-Glo 2.0 reagents (Promega, G9242) (Fig. 1c and Extended Data Fig. 2d).

## Generation of GFP knock-in cell lines
Generation of repair templates with msfGFP flanked by 550–900 bp homology arms in the N termini of NPM1, TCOF1, HP1α, EWSR1 and FUS, and the C terminus of SRRM2 into pUC19 vectors is described in the 'pUC19 repair templates' section. Linear repair template DNA fragments were generated by PCR (a list of the primers is provided in Supplementary Table 1), gel-extracted and purified using the QIAquick gel extraction kit (Qiagen, 28704).

A total of 350,000 HCT-116, U2OS, TC71 and 1765-92 cells per well was seeded onto six-well plates and transfected the next day with 2,400 ng of linearized repair template and 600 ng of pX458-mCherry-gRNA plasmid using Lipofectamine 3000 according to the manufacturer's instructions. Cells were first selected for mCherry expression using the FACSAria II instrument (BD) 48 h after transfection, cultured for 4–6 days, after which GFP-expressing cells were sorted into single-cell clones onto 96-well plates.

A total of 500,000 mES cells per well was seeded feeder-free. The medium was supplemented with 2× LIF and the cells were transfected the next day with 4,000 ng linearized repair template and 1,000 ng of pX-458-mCherry-gRNA plasmids using Lipofectamine 3000 according to the manufacturer's instructions. Cells were first selected for mCherry expression using the FACSAria II instrument (BD) 48 h after transfection, cultured for 4–6 days on 6 cm dish with feeder cells. mES cell colonies were hand-picked into 96-well plates with feeder cells.

Homozygous clones for NPM1, TCOF1, SRRM2 and HP1α were selected after verifying successful insertion into both alleles, and the knock-in of the WT allele of *EWSR1* was genotyped using the primers listed in Supplementary Table 1. Owing to the complex karyotype of U2OS cells, only heterozygous knock-in of *GFP–NPM1* was successful. Finally, the GFP-tagged proteins from the knock-in lines were verified by western blotting. Genotyping data for the cell lines generated in this study are included in Supplementary Figs. 13 and 15.

## Generation of doxycycline-inducible *eGFP–NPM1* overexpression systems in A673 cells
To generate a doxycycline-inducible overexpression system of *eGFP–NPM1*, we randomly integrated the coding sequences of *NPM1* wild type, KS, KS$_{F-to-G}$, KS$_{F-to-A}$ and KS$_{0F}$ into A673 cells using the PiggyBac transposon system.

Carrier plasmids (described above) and PiggyBac transposase expression vector (SBI, PB210PA-1) were co-transfected into A673 cells using Lipofectamine 3000 (Thermo Fisher Scientific) according to the manufacturer's instructions at a molar ratio of 5:1. The transfected bulk population was screened for integration by addition of 2 µg ml$^{-1}$ puromycin (Gibco) to the cell culture medium 24 h after transfection for a total of 5 days. The surviving cells were then used for experiments (Extended Data Fig. 2e–g).

## LacO–LacI tethering assay
U2OS 2-6-3 cells with LacO array[51] were seeded on eight-well chamber slides (Ibidi, 80826-90) at density of 30,000 cells per well and transfected the next day with CFP-LacI (empty control) or CFP-LacI-NUT plasmids using FuGENE HD and 175 ng plasmid per well according to the manufacturer's instructions. Then, 2 days after transfection, cells were fixed and stainings for RNAPII were performed as described in the 'Immunofluorescence' section.

Image analysis of the LacI–LacO tethering assay was performed using ZEN Blue v.3.9 software using the zone of influence method. LacI-NUT foci were detected using CFP signal (click thresholding; ValueLower: 16600; ValueUpper: 65535; Dilate: 1) and background regions were defined as rings surrounding the foci (ring distance: 5; thickness: 6). The mean CFP intensities and AlexaFluor 647 intensity for RNAPII were measured, and enrichment of the RNAPII signal was calculated by dividing the mean signal at foci by the mean signal at the background ring element.

## Transplantation-based models and *NUP98::KDM5A* cell line generation
*GFP–NUP98::KDM5A* mouse-model-derived cell lines were established as described previously[56]. In brief, *GFP–NUP98::KDM5A* cell lines were generated by retroviral co-transduction of MSCV−eGFP−NUP98::KDM5A with MSCV−rtTA3−IRES−Nras(G12D)−EF1a−Luc2 of mouse fetal liver-derived HPSCs (C57BL/6, Ly5.2). Then, 3.74 × 10$^6$ (6.1% GFP$^+$) cells were transplanted into sublethally irradiated (4,5 Gy) recipient mice (C57BL/6, Ly5.1) through tail-vein injection (Fig. 4f). Disease progression was monitored by whole-body luminescence imaging as previously described. Mice were euthanized after disease onset and bone marrow and spleen cells were collected. Stable cell lines were established by continuous culture of bone marrow cells for 4 weeks without supplemented cytokines. All animal studies were performed according to ethical animal license protocols and were

approved by the responsible authorities of the Austrian government (BMBWF-68.205/0199-V/3b/2018). For this, male and female C57BL/6J.SJL mice at the age of 10–12 weeks were used. Mice were kept in specific opportunistic pathogen free quality (SOPF) under stringent controlled standard conditions, in individually ventilated cages, fed with Sniff Haltungsfutter CHOW standard 10 mm pellets (catalog no. V1534-000), ad libitum. This study does not include any experiments in which animals were subjected to different treatment cohorts, for which sex-based analysis would be relevant.

## Live-cell imaging

All live-cell imaging experiments were performed using the LSM880 Airyscan microscope equipped with a Plan-Apochromat ×63/1.40 oil differential interference contrast objective, while incubating cells at 37 °C and 5% CO$_2$. Cells were seeded onto eight-well chamber slides (Ibidi, 80826-90) at 30,000 cells per well, transfected 24 h later and imaged 24 h after transfection. U2OS and HEK293T cells were transfected using FuGENE HD; and HCT-116, TC71, 1765-92, H3122 and V6.5 mES cells with Lipofectamine 3000 according to the manufacturer's instructions. Hoechst 33342 (0.2 µg ml$^{-1}$, Thermo Fisher Scientific, 62249) was added into cell culture medium for nuclear staining. To visualize nucleoli in living cells, RFP–fibrillarin fusion protein was expressed by transfecting cells with pTagRFP-C1-fibrillarin plasmid (Addgene, 70649)[73] together with plasmids for mEGFP-fsHMGB1-full-length variants.

For expressing *NUP98::KDM5A* and KS constructs, HEK293T cells were seeded onto eight-well chamber slides (Ibidi, 80826-90) and cultured until 70% confluency was reached. For transfection, 1 µg of plasmid DNA and 2.5% polyethylenimine (Polyscience, 26292) were mixed in 200 µl of opti-MEM I (Gibco, 31985062) and incubated for 20 min. The mixture was then added dropwise to each well. After overnight incubation, the medium was exchanged to fresh prewarmed growth medium before live-cell imaging. To visualize cell nuclei, cells were incubated with 5 µM DRAQ5 (NobusBio, NBP2-81125) 10 min before imaging.

In FRAP experiments, two regions of interest (ROIs) were determined: a rectangular ROI 1, and smaller, circular ROI 2 that covered the object to be bleached. GFP signal was bleached within ROI 2 using a 488 nm laser with 70–100% intensity, 5–20 iterations and GFP signal recovery was measured using 1–2 s intervals for 40–60 s. For co-FRAP assays with both GFP and mCherry signal, the mCherry signal was bleached using a 561 nm laser the same way as for the GFP signal. The laser intensity, number of iterations and size of ROIs varied between experiments, but were always identical within an experiment. Fluorescence intensities were acquired from 6–20 ROIs from separate condensates in each experiment, quantified using ZEN Black v.2.3 and reported as relative values to the pre-bleaching timepoint (Figs. 1b, 2e, 3g, 4b and 5b, Extended Data Figs. 1a, 2f, 3e, 4a, 6f, 8b, 9b, 10b,f, 12c,f and Supplementary Figs. 2d,g, 3b,e,h, 5b,e, 10d,g and 12b). Figures were generated using GraphPad PRISM9 and with R package ggplot2. In FRAP experiments with cells transfected with mCherry-P2A-T2A-GFP-nb vectors, an image was acquired from ROI 1 using Hoechst, mCherry and GFP channels before photobleaching, and the mCherry signal within nuclear area (defined by Hoechst signal) was used to quantify the mCherry expression level (Extended Data Figs. 3e and 4b and Supplementary Figs. 3c,f and 6b). To quantify the nuclear TagBFP intensity (Supplementary Fig. 5c,f), the mean intensity was measured from a square of 1.4 µm$^2$ manually placed at a nuclear region outside the nucleolus.

## NuFANCI

The NuFANCI method was adopted from the FANCI method[41]. Endogenously msfGFP–NPM1-tagged cells were used for the NuFANCI experiment. Six million cells were plated in 10 cm dishes, cultured for 24 h, the medium was then changed and cells were transfected using Lipofectamine 3000 with the GFP-nb constructs. The actinomycin-D-treated cells were treated with 400 nM actinomycin D (Sigma-Aldrich, A1410-2MG) for 1 h before collection. Then, 24 h after transfection, the untransfected, actinomycin-D-treated and GFP-nb-transfected cells were trypsinized, collected and then pelleted into 1.5 ml Eppendorf tubes. The cells were fixed with 1 ml 1% formaldehyde diluted in cell culture medium from 16% formaldehyde (Thermo Fisher Scientific, 28906) for 10 min at room temperature with rotation. The fixation was stopped by adding 1 M glycine (Jena Bioscience, CSS-510) to a final concentration of 200 mM for 5 min at room temperature with rotation. The cells were washed twice with cold PBS, each with spin of 1,000×*g* at 4 °C for 3 min, and the pellets were kept on ice for sorting as soon as possible without first freezing the cells. Next, the fixed cells were sorted on the BD FACSAria Fusion system to collect mCherry$^+$ cells from the transfected samples and mCherry$^-$ cells from the untransfected samples into 15 ml Protein LoBind Tubes (Eppendorf, 0030122216) coated with FACS buffer (2% FBS, 2 mM EDTA, in PBS). Around 850,000 and 500,000 events were collected from the transfected and untransfected samples, respectively. The sorted cells were pelleted in 1.5 ml Protein LoBind Tubes (Eppendorf, 0030108116). Next, the pellets were thoroughly resuspended in 1 ml lysis buffer B0 (50 mM HEPES pH 7.5, 150 mM KCl, 1% IGEPAL CA-630, cOmplete protease inhibitor (Sigma-Aldrich, 11873580001) and PhosSTOP (Merck 4906837001)) supplied with a final concentration of 1 mM DTT, 1:1,000 RNase inhibitor (NEB, M0314L) and 2 µg ml$^{-1}$ DAPI. The samples were incubated on ice for 5 min, transferred to Covaris milliTUBE (Covaris, 520130) and then sonicated using Covaris E220 (PIP: 140; duty factor: 5; duration: 120 s). Then, 30 µl of each sample was reserved as the input material for MS. The rest of each sample was transferred to 1.5 ml Protein LoBind Tubes and sorted on the BD FACSAria Fusion system with an SSC threshold of 1000 into 1.5 ml Protein LoBind Tubes (the sorting strategy and quality control are shown in Supplementary Fig. 4). For the gating strategy for the sorting of nucleoli, three gates were used: (1) DAPI (uv-450/50-A) versus GFP (b-530/30-A) was used to identify the population containing nucleoli (GFP$^+$DAPI$^{intermediate}$), determined by sorting different fractions outlined in Supplementary Fig. 4c and subsequent imaging (Supplementary Fig. 4d); (2) FSC-A versus SSC-A gate was used to exclude large events; (3) GFP (b-530/30) versus mCherry (yg-610/20) was used to sort for either mCherry$^-$ (for the samples untransfected and actinomycin D) or mCherry$^+$ (for the samples Nb, KS, KS$_{F-to-G}$ and 2×KS). Gates were determined by comparing to mCherry$^-$ samples (untransfected). Flow Cytometry data were collected and analysed using BD FACSDiva v.8.0.1; flow cytometry data visualization was performed using FlowJo. Around 400,000 events were collected from each sample. The collected nucleoli were centrifuged at 10,000×*g* for 10 min at 4 °C followed by one wash with cold PBS.

For MS sample preparation, the nucleolus samples were supplied to reach 1× buffer 4 (2× buffer 4: 100 mM Tris, pH 7.5, 50 mM NaCl and 4 mM MgCl$_2$) and incubated with 1,000 RPM shaking at 65 °C for 1 h and then 10 min at 95 °C. Each sample was then sonicated on the Qsonica Q700 sonicator equipped with microtip (Bioke, Q4417) with amplitude 5 for 10 s until there were no visible particles in the tube. Next, each sample was cooled on ice briefly before benzonase was added to a final concentration of 25 U µl$^{-1}$ (Thermo Fisher Scientific, 70-746-3) and the sample was incubated at 37 °C for 30 min with 1,000 rpm shaking. The samples were then supplied to reach 1× buffer 5 (2× buffer 5: 6 M GdmCl, 20 mM TCEP, 80 mM chloroacetamide) and incubated at 37 °C for 1 h with 1,000 rpm shaking. Next, to each sample 1 ml of ice-cold 100% acetone was added and precipitated at −20 °C overnight. The next day, the samples were centrifuged at 20,000×*g* for 10 min at 4 °C, the supernatant was discarded, and the sample was washed once with ice-cold 100% ethanol and centrifuged at 20,000×*g* for 10 min at 4 °C; the supernatant was discarded and the pellet was air-dried briefly to dry most of the ethanol but not to complete dryness. Next, the input and nucleolus samples were resuspended in 50 and 30 µl 100 mM (NH$_4$)HCO$_3$, respectively. The samples were then sonicated on the Qsonica Q700 sonicator equipped with microtip with amplitude 5 for 10 s until there were no visible particles in the

tube. The concentration of the samples was measured in duplicate on the Qubit 3 system using the Qubit protein assay kit (Thermo Fisher Scientific, Q33211). Around 2 mg of proteins was yielded from each nucleolus sample (the NuFANCI sample preparation quality control log is outlined in Supplementary Table 2). Next, 150 ng of each sample was digested with 5 ng trypsin and 5 ng Lys-C filled to final volume of 20 µl with 100 mM $(NH_4)HCO_3$ overnight at 37 °C with 800 rpm shaking. The peptides were acidified with formic acid to a final concentration of 2% and 150 ng of the digests was loaded onto Evotip Pure (Evosep) tips according to the manufacturer's protocol. Peptide separation was carried out by nanoflow reversed-phase liquid chromatography (Evosep One, Evosep) with the Aurora Elite column (15 cm × 75 µm inner diameter, C18 1.7 µm beads, IonOpticks) using the 20 samples a day method (Whisper Zoom 20 SPD). The LC system was coupled online to a timsTOF SCP mass spectrometer (Bruker Daltonics) using the data-dependent acquisition with parallel accumulation serial fragmentation method. The MS data were processed using MaxQuant (v.2.6.6.0; Max Planck Institute for Biochemistry) and searched against the human UniProtKB proteome (UP000005640; revision 2024 09 11). Additional modified sequences as outlined in Supplementary Table 1 were used accordingly. The match between run and label-free quantification features were used independently for the NuFANCI and input samples. The MS data have been deposited at the ProteomeXchange Consortium via the PRIDE partner repository[74] under dataset identifier PXD058854.

**Proteomics analysis.** MS data were acquired using TimsTOF SCP (Bruker). The raw peak files were processed using MaxQuant[75]. Label-free quantification (LFQ) values were calculated separately for NuFANCI and input, with a match between runs applied separately for NuFANCI and input. Alphastats (v.0.6.9)[76] was used to process the MaxQuant output. Protein group matrices were used as an input for Alphastats, and data were preprocessed to remove contaminants and reversed proteins. Principal component analysis (PCA) of the NuFANCI and input proteomes was performed with the 500 most variable proteins using ANOVA, and LFQ values were standardized (Extended Data Fig. 5a). For the NuFANCI subset, PCA was performed with a VST-transformed matrix (Extended Data Fig. 9b). Correlation plots were calculated using the SciPy package[77] in Python v.3.10 and plotted with Seaborn (Extended Data Fig. 5c). Heat maps were plotted using $log_2$-transformed LFQ values and clustered by Euclidean distance using the Ward method and plotted using Seaborn (Fig. 3b and Extended Data Fig. 9e). Protein expression plots were generated using Seaborn (Extended Data Fig. 9f). Volcano plot data were calculated using the Alphastats diff_expression_analysis function set to t-test and then plotted with Seaborn (Fig. 3c and Extended Data Fig. 5g). A list of the differentially detected protein groups is provided in Supplementary Table 2.

### Cell-penetrating-peptide experiments
Peptides ($R_{10}$, $R_{10}$MMMMNKLVLAQFFFSCL and $R_{10}$MMMMNKLVLAQGGGSCL) with N-terminal TAMRA labels were synthesized by Peptide Specialty Laboratories and reconstituted in DMSO into 2 mM stocks. GFP–NPM1 U2OS cells were seeded onto eight-well chamber slides (Ibidi, 80826-90) at a density of 60,000 cells per well. The next day, cells were washed twice with PBS and exposed to 3 µM peptides in PBS for 30 min in 37 °C. Wells were washed once with PBS and the cells were then kept in an 37 °C incubator for 3 h in the presence of 0.2 µg ml$^{-1}$ Hoechst 33342 (Thermo Fisher Scientific, 62249) in cell culture medium, followed by imaging on the LSM880 microscope. We noted that R10 control peptide was often present in cytoplasmic foci and only rarely in the nucleolus. To facilitate cellular distribution of R10 control peptide, cells were exposed to Texas Red filter light using the HXP lamp for 30 s, followed by a 30 s waiting period before image acquisition. Imaging of R10–KS and R10–KS$_{F-to-G}$ was done without an additional illumination step.

To analyse nuclear and nucleolar TAMRA signals in cells treated with cell-penetrating peptides, nuclei were first segmented using Otsu thresholding (click thresholding: 12, 255) and nucleoli were detected on the basis of the GFP–NPM1 signal (click thresholding: 6, 255). Mean nuclear, nucleoplasmic (outside nucleoli) and nucleolar TAMRA signal intensities were measured.

For the images of R10–KS-peptide-treated cells, $z$-positions for dying cells were processed separately, as the aggregated cytoplasmic and extracellular TAMRA signal biased the use of max intensity projections. We note that, for the TAMRA intensity calculations, in some cases a nucleus could be detected at different $z$-positions and may be counted twice. For this reason, the number of dying cells, indicated by the presence of small, condensed nuclei and increased Hoechst staining intensity, was counted by visual inspection. Images were acquired from two biological replicates and included combined 356, 460 and 511 cells for peptides R10, R10–KS and R10–KS$_{F-to-G}$, respectively. Replicates were pooled for TAMRA intensity measurements (Extended Data Fig. 6d).

### NPM1–R10 in vitro droplet fusion assay
For in vitro NPM1 droplet fusion assays, 30 µM of purified recombinant msfGFP–NPM1 was mixed with 2 µM of either TAMRA–R10, TAMRA–R10–KS, or TAMRA–R10–KS$_{F-to-G}$ and 5% PEG 8000 in a PCR tube and pre-assembled in the tube for 5 min. The volume for each droplet was 5 µl, consisting of 0.75 µl msfGFP–NPM1 in storage buffer (50 mM Tris pH 7.5, 125 mM NaCl, 10% glycerol), 0.5 µl of TAMRA–peptide in DMSO (only DMSO for the DMSO control condition), 1.25 µl 20% PEG 8000 in water and 2.5 µl storage buffer. The resulting 5 µl was pipetted onto a chambered coverslip (Ibidi, 80800). Images were acquired after 3 min equilibration of the drop on the slide, with an LSM880 confocal microscope equipped with a Plan Apochromat ×63/1.40 NA oil DIC objective with a ×1 zoom. For each field of view, time-series imaging captures 300 consecutive images with a 0.26 s time interval. Quantification of droplet fusion events was based on three independent image series per condition.

**In vitro droplet fusion analysis.** The droplet fusion events were sub-tracked in Fiji (v.2.3.0/1.53f) to contain the 40 slices that capture the one slice before the droplet fusion and 39 slices during the droplet fusion. Then, the fusing droplets were converted to binary mask and the ROI of the fusion droplets are measured for $\ell_{major}$ and $\ell_{minor}$ for each slice using Fiji's built-in measurement function. The aspect ratio, relaxation time ($\tau$), length scale ($\ell$) and inverse capillary velocity ($\eta/\gamma$) were calculated as previously described[36,46,78,79]. In brief, the aspect ratio of the fusing droplets was calculated by AR = $\ell_{major}/\ell_{minor}$. The time evolution of the aspect ratio was fit to function AR = $1 + (AR_{t0} - 1) \times \exp(-t/\tau)$, where $t$ is time, $\tau$ is relaxation time and $AR_{t0}$ is the aspect ratio at the first timepoint. The length scale of the fusion events was calculated by $\ell = (\ell_{minor,t0} \times (\ell_{major,t0} - \ell_{minor,t0}))^{0.5}$, where $t0$ indicates the value at the first timepoint. Plots of $\tau$ versus $\ell$ were fit to a line of the form $\tau = (\eta/\gamma) \times \ell$ to determine the inverse capillary velocity $\eta/\gamma$, which is the ratio of viscosity ($\eta$) to surface tension ($\gamma$) (Extended Data Fig. 7).

### Image analysis for live-cell imaging
**Generation of FRAP curves.** The recorded fluorescence intensity from each timepoint of each FRAP ROI was normalized to the signal intensity of the first timepoint. The replicates of each timepoint in the same FRAP series were plotted in GraphPad and RStudio with the error bar representing the s.d.

**Calculation of the mobile and immobile fraction.** The mobile fraction (Fig. 1e and Extended Data Figs. 1a, 2f and 3e) of the FRAP ROI was calculated using the signal intensity normalized as described above with the following equation:

$$\text{mobile fraction}$$
$$= \frac{\text{last timepoint signal intensity} - \text{after bleach signal intensity}}{1 - \text{after bleach signal intensity}}$$
$$\text{immobile fraction} = 1 - \text{mobile fraction}$$

**Circularity.** Live-cell images were acquired using a ×63 oil objective on a LSM880-airyscan under ZEN Black v.2.3 (Zeiss). For each condition, 8–31 regions were imaged, with a minimum of 26 nuclei captured. The resulting images were quantified in the image analysis module ZEN v.3.4 (Zeiss). In brief, within images, nuclei were identified by nuclear counterstaining using auto-intensity thresholds after smoothing (Gauss, 3.0). Nucleoli were segmented within nuclei by applying a fixed intensity threshold on GFP signal after faint smoothing (Gauss, 1.3) and using the rolling-ball algorithm. The maximum GFP area was set to 1,000, and the circularity score was extracted for each GFP object (Extended Data Fig. 2b).

**Experiments related to NUP98::KDM5A.** Raw files were imported into Arivis Vision 4D (Arivis) and an automated segmentation pipeline was designed manually. This pipeline consisted of median-based denoising, background correction, Cellpose deep-learning segmentation for cells and nuclei, intensity threshold segmentation for condensates, particle finder and size and sphericity filters. The segmented objects (cells, nuclei and condensates) were manually proofread, and settings were adjusted if necessary. For the data shown in Fig. 4h, ROIs were set based on the mCherry signal, and the number of condensates and mean GFP intensity was calculated for at least 100 individual cells per condition. The same approach was used for the data shown in Supplementary Fig. 9c, but at least 60 cells for each condition were analysed.

## Immunofluorescence

For immunofluorescence experiments performed in *GFP–NPM1* knock-in HCT116 cells (Fig. 3h and Supplementary Fig. 6a) and *eGFP–BRD4::NUT*-expressing cells (Fig. 3g,h and Supplementary Fig. 6g,h), cells were seeded on 8-well or 18-well chamber slides (Ibidi, 80826-90 and 81816) with 30,000 or 12,000 cells per well, and transfected 24 h later and fixed 24 h after transfection with 4% PFA in PBS for 10–15 min. Cells were permeabilized with 0.5% Triton X-100 (Thermo Fisher Scientific, 85111) in PBS for 10–15 min, incubated in blocking buffer containing 1% BSA (BSA Fraction V, Gibco, 15260037) and 0.1% Triton X-100 in PBS followed by overnight staining with primary antibodies in +4 °C with gentle agitation. Slides were washed five times with blocking buffer, incubated with secondary antibodies (AlexaFluor 647 donkey anti-mouse or anti-rabbit antibodies, Jackson Immuno-Research, 715-605-150 and 711-605-152, 1:1,000) in blocking buffer for 1 h in room temperature, washed twice with blocking buffer, stained with 0.5 µg ml$^{-1}$ DAPI in PBS (Invitrogen, D1306) and washed three times with PBS. The following primary antibodies were used: 5.8S rRNA (Novus, NB100-662SS, 1:500), HA-tag (Cell Signaling, C29F4, 1:1,000), NEPRO (Santa Cruz, sc-376579, 1:100), RNAPII (Abcam, ab26721, 1:500) and H3K27Ac (Abcam, ab4729, 1:1,000). Imaging was performed using the LSM880 Airyscan microscope equipped with a Plan-Apochromat ×63/1.40 oil differential interference contrast objective.

For the immunofluorescence experiment of NEPRO (Fig. 3d), all of the procedure steps were identical to as described above except for the sample fixation. The cells used for NEPRO immunofluorescence (IF) were fixed on the slide with 1% formaldehyde diluted in culture medium at room temperature for 10 min, quenched with a final concentration of 200 mM glycine for 5 min, and then washed once with PBS and followed by permeabilization.

For IF experiments performed in 52K, 52K–KS and 52K–KS$_{F\text{-to-A}}$ expressing cells (Fig. 5g and Supplementary Fig. 11d), cells were grown on 12 mm glass coverslips (Electron Microscopy Sciences, 72196-12) in 24-well, non-pyrogenic, polystyrene plates. For transient expression of transgenes, mammalian expression plasmids were transfected into HEK293T or HEK293 cells using X-tremeGENE HP (Roche, 6366236001) according to the manufacturer's instructions for a 3:1 reagent:plasmid ratio. Plasmids were transfected at a ratio of 1 µg DNA:3 µl X-tremeGene HP:4 × 10$^5$ cells and scaled up or down accordingly for all experiments. Then, 24 h after transfection, cells were fixed in 4% PFA in PBS at 37 °C for 10 min and washed once in PBS, followed by permeabilization with 0.5% Triton X-100 in PBS at room temperature for 10 min. The samples were blocked in 3% BSA in PBS (+0.05% sodium azide) for 1 h at room temperature, incubated with primary antibodies in 3% BSA in PBS (+0.05% sodium azide) for 1 h at room temperature, washed three times in 3% BSA in PBS (+0.05% sodium azide), followed by incubation with secondary antibodies and DAPI for 1 h at room temperature. The coverslips were then washed twice in PBS and mounted onto glass slides using ProLong Gold Antifade Reagent (Cell Signaling Technologies, 9071). The following primary antibodies were used: 52K (gift from P. Hearing[80]; rabbit, polyclonal, 1:500) and DBP (gift from A. Levine[81], mouse, B6-8, 1:400). AlexaFluor goat anti-rabbit 488 fluorophore-conjugated secondary antibody (Life Technologies, A-11008) or goat anti-mouse 488 fluorophore-conjugated secondary antibody (Life Technologies, A-11001) was used at a concentration of 1:1,000. Coverslips were imaged using a Leica DMi8 Thunder Imager and LAS X acquisition software. Images were processed in FIJI (v.1.53f51) using equivalent settings. Image analysis was performed using FIJI (v.1.53f51). Analysis of enrichment of IIIa in 52K condensates (Fig. 5h) is described below in Image Analysis.

## Image analysis for IF

**Pearson's correlation coefficients from IF images.** Pearson's correlation coefficients between GFP and IF staining intensities within nuclei were analysed using ImageJ v.2.14.0/1.54f (Fig. 4d, Extended Data Fig. 8e and Supplementary Fig. 6b). First, nuclei were segmented into ROIs using thresholding on the DAPI channel and the AnalyzeParticles tool, and Pearson's correlation coefficients between GFP and AF647 reported by Coloc2 plugin were collected and reported for each nucleus with mCherry or GFP expression. Thresholds for DAPI, mCherry and GFP channels varied between experiments, but were kept constant when thresholding images within each experiment.

**Quantifying 5.8S rRNA intensity in nucleoli.** Analysis of nucleolar 5.8S rRNA intensity in de-mixed and remaining nucleolar regions of GFP-nb–2×KS-expressing *GFP–NPM1* HCT-116 cells (Fig. 3i) was performed manually with ImageJ by selecting mCherry$^+$ cells where de-mixing of GFP–NPM1 was clearly visible. Measurements of GFP–NPM1 and AF647 intensities were performed within circular areas of 0.8 µm$^2$ that were positioned in de-mixed and remaining regions of nucleoli using the GFP–NPM1 intensity as illustrated in Fig. 3h.

**Enrichment of GFP–IIIa in 52K condensates.** Enrichment of GFP–IIIa in 52K condensates was analysed using ZEN software and the Zones of Influence analysis tool to identify 52K condensates with AlexaFluor 647 channel using click thresholding (value lower: 113; value upper: 246), and background as ring element (Segmentation Zoi Ring Distance: 3; Ring Thickness: 3) surrounding 52K objects. Enrichment of the mean GFP signal in 52K condensates over the mean background signal is displayed for individual 52K foci (Fig. 5h). Background GFP intensities are calculated as mean GFP signal at ring elements surrounding 52K foci, and are displayed in Fig. 5h.

## Protein purification

Overexpression of recombinant protein in BL21 (DE3) (NEB M0491S) was performed as described previously[72]. In brief, *Escherichia coli*

pellets were resuspended in 50 ml of ice-cold buffer A (50 mM Tris pH 7.5, 500 mM NaCl) supplemented with cOmplete protease inhibitors (Sigma-Aldrich, 11697498001), 0.2% Triton X-100 (Thermo Fisher Scientific, 851110) and 5% DMSO (Sigma-Aldrich, D2650-100ml), and sonicated for 360 cycles (5 s on, 5 s off) on the Branson SFX150 sonicator. The bacterial lysate was kept under stirring on a stirrer during sonication on an ice bucket at 4 °C in a cold room. Bacterial lysates were cleared by centrifugation at 15,000×g for 15 min at 4 °C. For protein purification, we used the Äkta avant 25 chromatography system. All 50 ml of the cleared lysate was loaded onto the cOmplete His-Tag purification column (Merck, 6781535001) pre-equilibrated in buffer A. The loaded column was washed with 15 column volumes (CV) of buffer A. Fusion protein was eluted in 10 CV of elution buffer (50 mM Tris pH 7.5, 500 mM NaCl, 250 mM imidazole) and diluted 1:1 in storage buffer (50 mM Tris pH 7.5, 125 mM NaCl, 1 mM DTT, 5% DMSO, 10% glycerol). The fractions enriched for GFP were pooled after His-affinity purification and manually loaded through an injection valve connected to a 500 µl capillary tube onto an equilibrated Superdex 200 increase 10/300 GL column (Cytiva, 28-9909-44). The loaded column was equilibrated with 0.15 CV of ice-cold SEC buffer (50 mM Tris pH 7.5, 125 mM NaCl, 5% DMSO, 1 mM DTT) supplemented with cOmplete protease inhibitors. Fusion proteins were eluted into 300 µl fractions with 1.1 CV of ice-cold SEC buffer supplemented with cOmplete protease inhibitors. Elution fractions were pooled. Eluates were further concentrated by centrifugation at 4,000×g for 30 min at 4 °C using 30 kDa MWCO Amicon Ultra centrifugal filters (Merck, UFC903024). The concentrated fraction was diluted 1:100 in storage buffer, reconcentrated and stored at −80 °C.

### In vitro msfGFP droplet formation assay
For in vitro droplet-formation experiments (Fig. 1f), we measured the concentration of purified msfGFP-tagged proteins using the NanoDrop 2000 system (Thermo Fisher Scientific) and subsequently diluted protein to the required concentration in storage buffer (50 mM Tris pH 7.5, 125 mM NaCl, 1 mM DTT, 5% DMSO, 10% glycerol). The in vitro droplet-formation assay was performed as previously described[82,83]. Protein preparations were mixed 1:1 with 2.5 µl 20% PEG 8000 in deionized water (w/v). The resulting 5 µl was pipetted onto a chambered coverslip (Ibidi, 80826-90). Images were acquired after 3 min equilibration of the drop on the slide, with an LSM880 confocal microscope equipped with a Plan Apochromat ×63/1.40 NA oil DIC objective with a ×1 zoom. Quantification of condensate formation was based on at least ten images acquired in at least two independent image series per condition.

### Image analysis of in vitro droplet formation
Protein droplets were detected using the ZEN blue v.3.4 Image Analysis and Intellesis software packages. Using a previously trained Intellesis model in spectral mode, we achieved image segmentation of individual pixels into objects (droplet area) or background (image background). Relative amounts of condensed protein were calculated by dividing the sum of the GFP signal in objects defined as droplet area by the overall sum of the GFP signal in the field of view. All values were calculated using RStudio. Plots were generated using GraphPad PRISM9. To fit data to a sigmoidal curve, we applied the in-built nonlinear regression function (Sigmoidal; x is concentration) (Fig. 1g).

### RNA-seq
For experiments involving eGFP–BRD4::NUT (Fig. 4e), 450,000 HEK293T cells were seeded onto six-well plates and transfected the next day with 3 µg pRK5-eGFP–BRD4::NUT plasmids using 9 µl PEI STAR transfection reagent (Tocris, 7854). Total RNA was isolated from cells 24 h after transfection using the RNEasy mini kit with in-column DNase digestion (Qiagen). RNA-seq libraries were prepared using the KAPA HyperPrep Kit with RiboErase (Roche, KK8562) and sequenced in 100 bp

paired-end mode on the Illumina NovaSeq2 system for 55–65 million fragments per sample.

**Bulk RNA-seq analysis.** Raw RNA-seq data were filtered and trimmed using TrimGalore v.0.6.10 (https://doi.org/10.5281/zenodo.7598955) with the default settings. Filtered data from HEK293K cells were mapped to a custom human genome hg38, including the eGFP sequence cloned using the STAR aligner[84] to hg38 human genome. Count-read tables were generated by using the same program. Differential expression analysis was performed using the DEseq2 package[85] in R (v.4.4)[86]. Differentially expressed genes were defined as having a fold change ≥ 1.2, Benjamini–Hochberg $P \leq 0.01$ and a minimum mean read count across the experimental samples of 50 reads. The differentially expressed genes are listed in the Supplementary Table 3.

PCA was performed using the PCAPlot function from the DEseq2 package on the normalized read matrix that was transformed using the variance-stabilizing transformation function from the DEseq2 package and plotted using ggplot2 (Extended Data Fig. 9d). The distance matrix was calculated using the dist function in R using the Euclidean distance and visualized with a pheatmap in R (Extended Data Fig. 9e). Volcano plots were created using ggplot2 (Fig. 4e and Extended Data Fig. 9g).

For the reference data on BRD4::NUT targets from a previous study[87], raw data were downloaded from the Gene Expression Omnibus (GSE233302) and processed as described above. Differentially expressed genes were defined with a fold change cut-off of 1.2 and an adjusted $P < 0.01$. Only BRD::NUT and control samples were used.

### RT–qPCR
For experiments involving expression of NUP98::KDM5A and KS variants, total RNA was extracted using the Monarch Total RNA Miniprep Kit. Reverse transcription was performed using the RevertAID RT Kit (Thermo Fisher Scientific) and qPCR was done using the SsoAdvanced Universal SYBR Green Supermix (Bio-Rad) or AceQ Universal SYBR qPCR Master Mix (Vazyme, Red Maple Hi-tech Industry Park) on the Bio-Rad CFX-Connect Real-Time PCR Detection System. The results were normalized to GAPDH and analysed using the $2^{-\Delta\Delta C_t}$ method (Supplementary Fig. 8e).

### 1,6-hexanediol treatments
Cleavable mCherry was included in eGFP–BRD4::NUT and GFP–NUP98::DDX10 expression vectors to better distinguish and compare transfected cells. U2OS cells were seeded on eight-well chamber slides (Ibidi, 80826-90) at density of 30,000 cells per well and transfected the next day with pRK5-mCherry-P2A-T2A-eGFP–BRD4::NUT plasmids using FuGENE HD and 150 ng plasmid per well according to the manufacturer's instructions. The next day, cells were treated with 5% 1,6-hexanediol (Sigma-Aldrich, 240117) in cell culture medium for 5 or 15 min, fixed with 4% PFA for 10 min, washed and stored in PBS. Nuclei were stained using 0.5 µg ml⁻¹ DAPI in PBS (Invitrogen, D1306).

For the eGFP–BRD4::NUT 1,6-hexanediol experiments, images were analysed using ZEN Blue v.3.9 software. Nuclei were segmented using automatic Otsu thresholding and the mean intensity, s.d. and maximum intensities from the GFP and mCherry channels were measured for each nucleus. Nuclei with no expression (mean mCherry intensity < 1.5) abnormally high expression (mCherry expression > 40) and cells with saturated GFP signals were excluded. Images were acquired from three biological replicate experiments and, in the end, measurements were pooled and combined into single plots (Extended Data Fig. 9i).

To analyse 1,6-hexanediol treated GFP–NUP98::DDX10-expressing cells, nuclei were first segmented using Otsu thresholding (click thresholding: 2, 255) and GFP–NUP98::DDX10 foci in nuclei were detected on the basis of the GFP signal (click thresholding: 9, 255). Nuclear background areas outside the foci were determined with inverted thresholds and the area was shrunk with Erode set to 2. To reliably measure diffuse GFP in the nucleus, GFP acquisition settings were

set accordingly. However, due to the high contrast of GFP intensity in NUP98::DDX10 foci versus nucleoplasm, the GFP signal was often saturated at NUP98::DDX10 foci already in cells with low GFP expression. As a compromise, we excluded all nuclei with saturated foci detected in areas greater than 8 pixels. Nuclei with no expression, defined as mean mCherry intensity < 1.5, and nuclei with very high expression, defined as mCherry expression > 40, were excluded. Images were acquired from two biological replicate experiments. Measurements were pooled and combined into single plots (Extended Data Fig. 10i). Owing to the small regions of saturated GFP that were permitted, mean GFP intensities for NUP98::DDX10 samples were not plotted, s.d. values of GFP intensity should be interpreted with caution and, instead, the background GFP is considered most reliable.

### Colony-formation assay for NUP98::KDM5A

Each experiment was performed in triplicates (Supplementary Fig. 8b). In brief, $25 \times 10^3$ mouse leukaemia cells were seeded in methylcellulose (MethoCult M3434, Stem Cell Technologies) and colonies were scored and replated ($10 \times 10^3$ cells or $5 \times 10^3$ cells) every 7 days.

### Flow cytometry for NUP98::KDM5A

For characterization of cells derived from the NUP98::KDM5A colony-formation assay, cells were washed with PBS and resuspended in PBS with 0.5% FCS, and then stained for 30 min with 1:200 dilutions of the following antibodies (all from BioLegend): anti-mouse Gr-1/Ly-6C BV421 (RB6-8C5) and anti-mouse KIT APC (2B8). For gating strategy, live cells were discriminated on the basis of forward scatter height (FSC-H) and side scatter height (SSC-H). Single cells were gated on the basis of forward scatter height (FSC-H) and forward scatter area (FSC-A). mCherry$^+$ cells were identified by their signal intensity in the ECD channel. Cellular staining for anti-mouse Gr-1/Ly-6C was assessed on the basis of the signal intensity in the Pacific Blue channel (BV421), while anti-mouse KIT staining was evaluated on the basis of the signal intensity in the APC channel. The samples were analysed on the BD FACSCanto II. Data analysis was performed using the FlowJo (FlowJo) software package (Supplementary Fig. 8c,d).

### Competitive cell proliferation assay for NUP98::KDM5A

To investigate the effect of the KS on proliferation, a competition-based proliferation assay was performed using GFP–NUP98::KDM5A AML cells transfected with doxycycline-inducible nb–KS constructs at an infection rate of approximately 25–30% as described previously[56]. In total, $5 \times 10^5$ cells were seeded in 24-well plates in triplicates. Doxycycline was added to the medium and fluorescence expression of mCherry (bicistronically expressed with nb–KS constructs) was measured every 2–3 days using the Cytoflex S instrument (Beckman Coulter). The effect of the nb–KS constructs on cell fitness was monitored by time-resolved measurements of mixed populations of competing cells expressing nb–KS constructs (mCherry$^+$) versus cells not expressing nb–KS constructs (mCherry$^-$). Data analysis was performed using the FlowJo (FlowJo) software package and values were normalized to day 3 after doxycycline induction (Supplementary Fig. 7b,c).

### Growth curves for NUP98::KDM5A

Cells were sorted for mCherry and seeded in biological triplicates and treated with doxycycline every 48 h or 72 h. Cell numbers were determined at regular intervals using the Intellicyt iQue Screener (Essen BioScience, Sartorius Group) and integrated with ForeCyte Software (Essen Bioscience; Standard Edition 10.0 (R1) v.10.0.8272; build date, 25 August 2022; Fig. 4g).

### Viruses and infections

For retrovirus production related to NUP98::KDM5A experiments, Platinum-E cells were co-transfected with 20 µg transfer vector and 5 µg pCMV-gag/pol using polyethyleneimine (branched, molecular mass, 25,000 Da, Sigma-Aldrich). The viral supernatant was collected 48 h and 56 h after transfection, filtered (0.45 µm) and supplemented with recombinant mIL-3 (10 ng ml$^{-1}$), mIL-6 (10 ng ml$^{-1}$) (both PeproTech) and mSCF (100 ng ml$^{-1}$). Mouse progenitor cells were spinoculated with viral supernatants (1:2 diluted) for 45 min at 37 °C in 500×$g$ in the presence of polybrene (4 µg ml$^{-1}$) (Merck Chemicals and Life Science). For lentivirus production, Lenti-X 293T cells were transfected with 4 µg transfer vector, 2 µg psPAX2 and 1 µg pMD2.G using PEI. Lentivirus was collected 48 h and 72 h after transfection and filtered (0.45 µm). Target cells were spinoculated after addition of virus supernatant (1:3 diluted) followed by centrifugation for 90 min at 37 °C at 1,000×$g$ in the presence of polybrene (5 µg ml$^{-1}$).

For experiments related to 52K constructs, human adenovirus type C5 (Ad5) wild type was purchased from ATCC (VR-5), propagated on HEK293 cells, purified through caesium chloride density ultracentrifugation and stored in 40% glycerol at −20 °C for infections. The Ad5 Δ52K mutant pm8001 (Δ52K)[88] (gift from P. Hearing) was propagated on a transgenic cell line (A549) expressing WT 52K[57], purified by caesium chloride density ultracentrifugation and stored in 40% glycerol at −20 °C for infections. Viruses were purified using two sequential rounds of ultracentrifugation in caesium chloride density gradients. To achieve a cryoprotective solution for storage, virus was diluted as followed: two parts virus in caesium chloride, one part 5× viral dilution solution (40 mM Tris pH 8, 400 mM NaCl, 0.4% BSA in $H_2O$), two parts 100% glycerol. Viral titres were determined by infectious focus-forming assay as described in the 'Adenoviral progeny production' section. All infections were carried out using a multiplicity of infection of 10 unless stated otherwise and collected at the indicated hours after infection. To infect cells, virus was diluted in cell culture medium without FBS. After 2 h at 37 °C, culture medium containing 10% FBS was added. For virus-yield assays, the virus infection medium was removed after 2 h, and cells were washed once in PBS before addition of culture medium to remove excess virus.

**Adenoviral progeny production.** Infected cells were collected by scraping and were then lysed by four cycles of freeze–thawing in liquid nitrogen and a 37 °C water bath. Cells were collected at 48 h after infection unless stated otherwise. Cell debris was removed from the lysates by centrifugation at maximum speed at 4 °C for 5 min. For analysis of virus yield, lysates were diluted serially in DMEM supplemented with 2% FBS and 1% penicillin–streptomycin and used to infect A549 cells. The infection medium was removed 2 h after infection, cells were washed once in PBS to remove excess virus, and cells were overlaid with growth medium. Cells were incubated for 24 h before fixation in 4% paraformaldehyde and analysed by immunofluorescence confocal microscopy using immunostaining of the viral DNA-binding protein (DBP) as an indicator of infection. For each of the three independent replicates, three fields of view were captured and the percentage of DBP-positive cells was determined. The serial dilution resulting in the closest to 50% DBP-positive cells was selected for calculation of progeny production. The total cell number was determined by counting cells grown in parallel under equivalent conditions. The number of focus-forming units was calculated as the product of total cell number and the percentage of antigen-positive cells, adjusting for the Poisson distribution. Virus input was determined by collecting infected cells at 4 h after infection, the number of infectious units was determined and the mean of replicates was calculated (Fig. 5e). For complementation of Δ52K mutant virus infection, cells were infected and then subsequently transfected at 2 h after infection.

### Zebrafish embryo manipulations

Zebrafish were maintained and raised under standard conditions, and according to Swiss regulations (canton Vaud, license number VD-H28). Wild-type (TLAB) fish were used for this study. Embryos were collected immediately after fertilization. The embryos were grown at 28 °C and

the developmental stage was determined as described by previously[89]. In total, 180 pg of each RNA was injected in the yolk of one-cell-stage embryos. The embryos were allowed to develop in a 28 °C incubator and then manually dechorionated using forceps. Embryos were transferred to mounting medium (0.8% low-melting-point agarose containing 15% (v/v) OptiPrep (Sigma-Aldrich, D1156)) at 37 °C and mounted onto an Ibidi glass-bottom μ-dish (Ibidi, 81158-400).

**Microscopy and image analysis for zebrafish embryos.** Embryos were imaged on a Nikon spinning disc, using the Nikon SR HP Plan Apo ×100/1.35 Sil WD 0.3 objective, in a temperature-controlled chamber set at 28 °C. 4D imaging, $x$, $y$, $z$ and time, was performed by keeping a $z$-stack of 27 μm and $z$-step of 0.3 μm with a time resolution of 2 min. Microscopy images were further analysed and processed using ImageJ. Whole nuclei at the 512-cell stage were first segmented in 3D and further analysis was performed using the 3D objects Counter plugin[90]. The generated data frames in .csv format were exported to RStudio software for plotting and visualization. Statistical analysis was performed using GraphPad prism. Kruskal–Wallis tests with Dunn's multiple-comparison correction were performed to calculate the significance between different groups.

### SDS–PAGE and immunoblot analysis
**For the generation of msfGFP knock-in cell lines.** Cultured cells were washed twice in PBS and lysed in RIPA buffer for 30 min at 4 °C on an orbital shaker. Subsequently, the cell lysates were centrifuged for 10 min at 20,000×$g$. The cleared lysates were transferred to a new tube and quantified using a BCA assay (Thermo Fisher Scientific). Then, 20 μg of extracted protein was supplemented with lithium dodecyl sulfate (LDS) loading buffer (Thermo Fisher Scientific, NP0007) supplemented with 50 mM DTT and boiled at 98 °C for 5 min. The samples were then run on a 4–12% NuPAGE SDS gel (Thermo Fisher Scientific, NP0322BOX) and transferred onto a polyvinylidene fluoride membrane (Thermo Fisher Scientific, IB24002x3) using an iBlot2 Dry Gel Transfer Device (Invitrogen) according to the manufacturer's instructions. Membranes were blocked with 5% skimmed milk in TBST for 1 h then incubated with primary antibodies diluted in 2% milk and TBST overnight at 4 °C. Primary antibodies used in this study include TCOF1 (Santa Cruz, sc-374536, 1:750), GFP (Invitrogen, A11122, 1:2,000), NPM1 (Invitrogen, 32-5200, 1:2,000), HP1α (CST, 2616, 1:1,000), histone H3 (Abcam, ab1719, 1:10,000), GAPDH (CST, 14C10, 1:4,000) and HSP90 (BD, 610419, 1:2,000). HRP-conjugated secondary antibodies were used against the host species at 1:1,000 dilution. The proteins were visualized with HRP substrate SuperSignal West Dura (Thermo Fisher Scientific) and detected using Bio-Rad Universal Hood III and Image Lab 6.1 software (Supplementary Fig. 13).

**For the experiments performed relating to the adenovirus-52K experiments.** SDS–PAGE and immunoblot analysis was performed using standard methods. In brief, protein samples were prepared using LDS loading buffer (Thermo Fisher Scientific, NP0007) supplemented with 25 mM DTT and boiled at 95 °C for 10 min. Equal amounts of protein lysates were separated by SDS–PAGE. Proteins were transferred onto methanol-activated polyvinylidene fluoride membrane (Millipore-Sigma, IPFL00010) at 30 V for 60–120 min and blocked in blocking buffer (5% milk in TBST supplemented with 0.05% sodium azide) for 1 h at room temperature. Membranes were incubated overnight at 4 °C with primary antibodies diluted in blocking buffer, washed for 30 min in TBST, incubated for 1 h at room temperature with HRP-conjugated secondary antibody diluted in blocking buffer and washed again for 30 min in TBST.

The following primary antibodies were used: anti-adenovirus type 5 antibody (raised against whole adenovirus capsids; recognizing late proteins Hexon, Penton, Fiber (Abcam, ab6982); rabbit, polyclonal,

western blot, 1:10,000), antibody to 52K (gift from P. Hearing[80], rabbit, polyclonal, western blot, 1:10,000), IIIa (gift from P. Hearing[91], rabbit, polyclonal, western blot, 1:10,000), DBP (gift from A. Levine[81], mouse, B6-8, western blot, 1:1,000) and GAPDH (GeneTex, GTX100118, 43712, rabbit, polyclonal, western blot, 1:5,000). For immunoblot analysis, horseradish-peroxidase-conjugated goat anti-rabbit secondary antibody (Jackson Laboratories, 111-035-045) was used at a concentration of 1:10,000.

Proteins were visualized using the Pierce ECL Western Blotting Substrate (Thermo Fisher Scientific, 34577) and detected using a Syngene G-Box using GeneSys acquisition software. Images were processed and assembled in Adobe Illustrator CS6 (Fig. 5f and Supplementary Figs. 11f and 14).

### Statistics and reproducibility
All experimental observations were confirmed by independent repeat experiments. No blinding or sample randomization was used during data capture or analysis. No data were excluded from data analysis. No statistical tests were performed to predetermine the sample size. Unless stated otherwise, the mean of numerical data is shown, with error bars representing the s.d. of the sample. Statistics were performed using GraphPad Prism v.9 and RStudio v.2024.04.2 and the multicomp package. All $t$-tests were two-sided. A normal distribution was assumed when determining statistical tests. Equal variance was not assumed, except for Tukey's multiple-comparison tests. Exact $P$ values were as follows: Fig. 1e: FRAP plots, $P \leq 0.0001$ (Δkillswitch), $P \leq 0.0001$ (F-to-E&D), $P \leq 0.0001$ (F-to-A), $P \leq 0.0001$ (F-to-G), $P \leq 0.0001$ (0F), $P \leq 0.0001$ (F12A), $P = 0.02$ (F13A), $P \leq 0.0001$ (ΔKS-3F), $P \leq 0.0001$ (11G3F3G), $P \geq 0.9999$ (C16-to-A), $P = 0.57$ (M-to-E&D); GFP intensity plot, $P = 0.97$ (Δkillswitch), $P = 0.56$ (F-to-E&D), $P = 0.68$ (F-to-A), $P = 0.53$ (F-to-G), $P = 0.27$ (0F), $P = 0.13$ (F12A), $P = 0.28$ (F13A), $P = 0.95$ (ΔKS-3F), $P = 0.19$ (11G3F3G), $P = 0.9999$ (C16-to-A), $P = 0.71$ (M-to-E&D). Fig. 2f: NPM1 plot, $P = 0.02$ (nb), $P = 0.16$ (KS), $P = 0.29$ (KS$_{F-to-A}$), $P = 0.01$ (2×KS); TCOF1 plot, $P = 0.99$ (Nb), $P = 0.41$ (KS), $P = 0.18$ (KS$_{F-to-A}$), $P = 0.19$ (2×KS); SRRM2 plot, $P = 0.92$ (Nb), $P = 0.39$ (KS), $P \leq 0.0001$ (2×KS), $P = 0.9998$ (2×KS$_{F-to-G}$); HP1α plot, $P = 0.85$ (Nb), $P = 0.99$ (KS), $P = 0.99$ (2×KS), $P = 0.76$ (2×KS$_{F-to-G}$).

### AlphaFold models
AlphaFold models were predicted using AlphaFold (v.3)[33] in multimeric mode using the default parameters. Models were visualized with ChimeraX (v.1.6)[92]. Contact plots were generated using custom scripts.

### Reporting summary
Further information on research design is available in the Nature Portfolio Reporting Summary linked to this article.

## Data availability
Sequencing data were deposited at the Gene Expression Omnibus under accession code GSE284494. MS data were deposited at the ProteomeXchange Consortium via the PRIDE partner repository under dataset identifier PXD058854. The NGS experiments of human samples used the human genome hg38 and annotation from GENCODE GRCh38.p13. Plasmids were deposited at Addgene (237619–237693 and 238231–238298). All raw and processed data were deposited at Zenodo, and are publicly available[93] (https://doi.org/10.5281/zenodo.15322636). All raw and processed data are also available at https://hdl.handle.net/21.11101/0000-0007-FE67-8. Source data are provided with this paper.

## Code availability
All custom code used for data analyses are publicly available at Zenodo[93] (https://doi.org/10.5281/zenodo.15322636).

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

**Acknowledgements** We thank the staff at the microscopy facility of the MPIMG, in particular, T. Mielke, R. Buschow and B. Fauler; the members of the FACS facility at the MPIMG, in particular, C. Giesecke-Thiel and U. Marchfelder; the staff at the MS and sequencing facilities at the MPIMG; and J. Nijssen, D. Kuster, X. Yan and A. Hyman for advice on the TDP-43 patch. The research was also supported by resources of the VetCore Facility (Imaging) of the University of Veterinary Medicine Vienna, in particular, U. Reichart and S. Handschuh. The schematics in Figs. 3a and 4f, and Supplementary Fig. 8a were partially created using BioRender (www.BioRender.com). This work was funded by the Max Planck Society (T.A., M.L.K., D.H.), and was partially supported by the following sources: Deutsche Forschungsgemeinschaft (DFG) Priority Program Grant HN 4/3-1 (D.H.), Klaus Tschira Boost Fund (H.N.), Behrens-Weise-Foundation (D.H.), HFSP long-term fellowship LT000086/2020-L (K.M.), Austrian Science Fund (grant https://doi.org/10.55776/P35298 and https://doi.org/10.55776/TAI490) (F.G.), DOC Fellowship of the Austrian Academy of Sciences (M.A.), DFG Priority Program Grant SPP2191 (M.L.K., project number 506373047) supporting K.M. and the National Institute of Allergy and Infectious Diseases (NIAID) of the National Institutes of Health (NIH) by grants R01-AI145266 (M.D.W.), R01-AI121321 (M.D.W.) and R01-AI118891 (M.D.W.). M.M. is paid from project no. 507866755 of DFG Priority Program 2202 (given to T.A.). Work in the Hnisz lab is also supported by the Mark Foundation for Cancer Research and the Worldwide Cancer Research foundations.

**Author contributions** HMGB1 mutagenesis experiments: H.N., I.S. and G.S. msfGFP in vitro experiments: Y.Z. Design and generation of GFP-nb–KS expression systems: H.N. GFP knock-in and experiments with GFP–NPM1 (H.N.), GFP–TCOF1 (Y.Z.), SRRM2–GFP (H.N.), GFP–HP1α (I.S.). NuFANCI: Y.Z. Fusion-oncoprotein experiments: Y.Z., H.N. and Y.-C.S. AlphaFold3 modelling: A.P.M. and Y.Z. RNA-seq and MS data analysis: A.P.M. R10 peptide experiments: H.N. and Y.Z. *NUP98::KDM5A* experiments: P.F.-P. and M.A. 52K experiments: M.C., N.G. and E.H. Zebrafish experiments: S.D. Flow cytometry operation: M.P.-S. MS operation: D.M. Assistance in NPM1 and SRRM2 experiments: K.M., M.M. and I.A.I. Resources: T.A., M.L.K., N.V., M.D.W., F.G. and D.H. Writing—original draft: Y.Z., I.S., H.N. and D.H. Writing—review and editing: all of the authors. Funding acquisition: T.A., M.L.K., N.V., M.D.W., F.G., H.N. and D.H.

**Funding** Open access funding provided by Max Planck Society.

**Competing interests** Y.Z., H.N. and D.H are listed as inventors on a patent application (223664PEP) on micropeptide tools to manipulate condensates filed by the Max Planck Society based on results in the study. D.H. is a founder and scientific advisor of Nuage Therapeutics. The other authors declare no competing interests.

**Additional information**
**Correspondence and requests for materials** should be addressed to Henri Niskanen or Denes Hnisz.

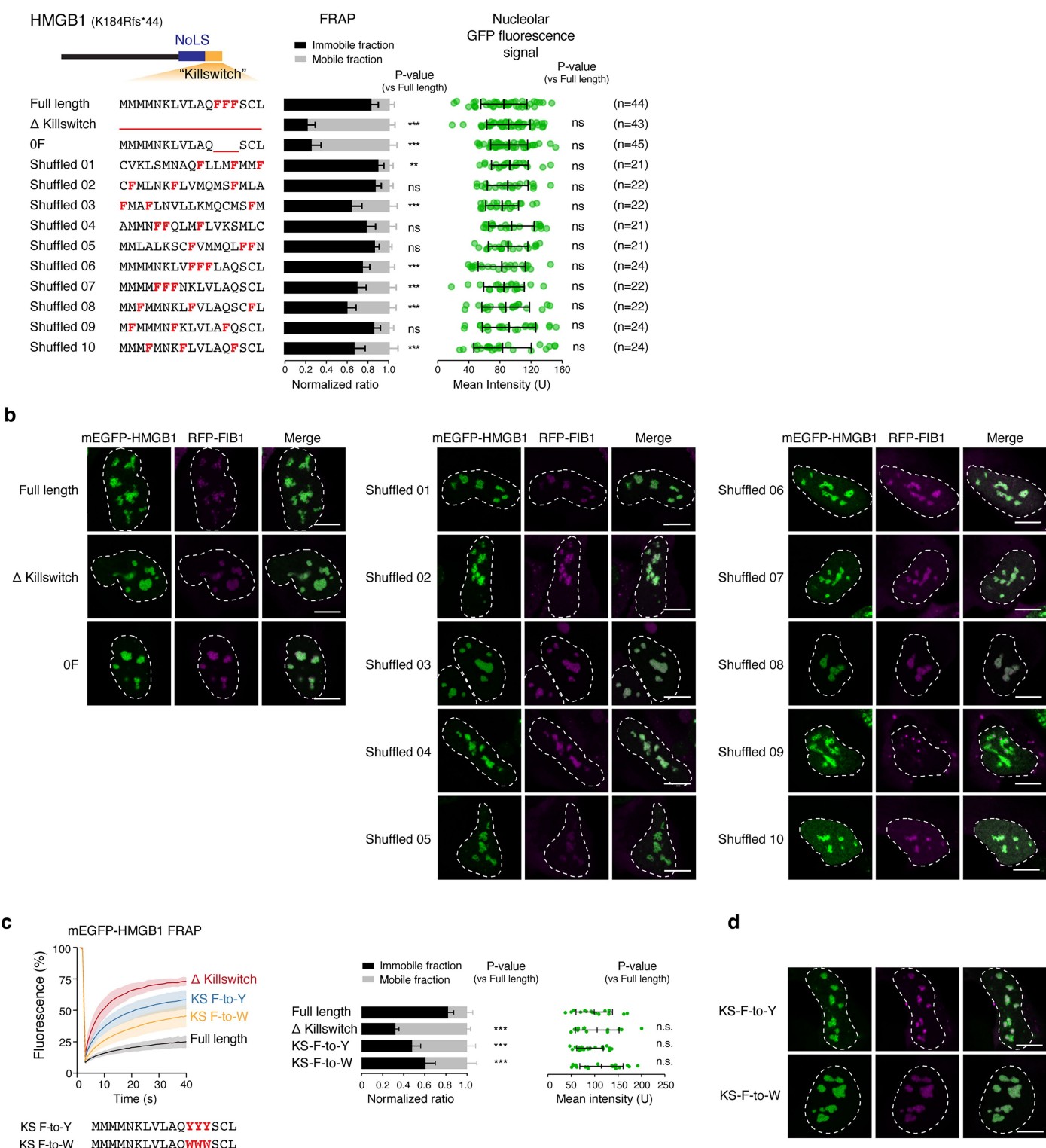

**a**

HMGB1 (K184Rfs*44)

NoLS / "Killswitch"

FRAP | Nucleolar GFP fluorescence signal

| | Sequence | | P-value (vs Full length) | | P-value (vs Full length) | |
|---|---|---|---|---|---|---|
| Full length | MMMMNKLVLAQ**FFF**SCL | | | | | (n=44) |
| Δ Killswitch | ——————— | | *** | | ns | (n=43) |
| 0F | MMMMNKLVLAQ___SCL | | *** | | ns | (n=45) |
| Shuffled 01 | CVKLSMNAQ**F**LLM**F**MM**F** | | ** | | ns | (n=21) |
| Shuffled 02 | C**F**MLNK**F**LVMQMS**F**MLA | | ns | | ns | (n=22) |
| Shuffled 03 | **F**MA**F**LNVLLKMQCMS**F**M | | *** | | ns | (n=22) |
| Shuffled 04 | AMMN**FF**QLM**F**LVKSMLC | | ns | | ns | (n=21) |
| Shuffled 05 | MMLALKSC**F**VMMQL**FF**N | | ns | | ns | (n=21) |
| Shuffled 06 | MMMMNKLV**FFF**LAQSCL | | *** | | ns | (n=24) |
| Shuffled 07 | MMMM**FFF**NKLVLAQSCL | | *** | | ns | (n=22) |
| Shuffled 08 | MM**F**MMNKL**F**VLAQSC**F**L | | *** | | ns | (n=22) |
| Shuffled 09 | M**F**MMMN**F**KLVLA**F**QSCL | | ns | | ns | (n=24) |
| Shuffled 10 | MMM**F**MNK**F**LVLAQ**F**SCL | | *** | | ns | (n=24) |

■ Immobile fraction  ▨ Mobile fraction

0  0.2  0.4  0.6  0.8  1.0
Normalized ratio

0  40  80  120  160
Mean Intensity (U)

**b**

mEGFP-HMGB1 | RFP-FIB1 | Merge

Full length
Δ Killswitch
0F

Shuffled 01
Shuffled 02
Shuffled 03
Shuffled 04
Shuffled 05

Shuffled 06
Shuffled 07
Shuffled 08
Shuffled 09
Shuffled 10

**c**

mEGFP-HMGB1 FRAP

Δ Killswitch
KS F-to-Y
KS F-to-W
Full length

Fluorescence (%)
100
75
50
25
0
0  10  20  30  40
Time (s)

■ Immobile fraction  ▨ Mobile fraction

P-value (vs Full length) | P-value (vs Full length)

| | | |
|---|---|---|
| Full length | | |
| Δ Killswitch | *** | n.s. |
| KS-F-to-Y | *** | n.s. |
| KS-F-to-W | *** | n.s. |

0  0.2  0.4  0.6  0.8  1.0
Normalized ratio

0  50  100  150  200  250
Mean intensity (U)

KS F-to-Y  MMMMNKLVLAQ**YYY**SCL
KS F-to-W  MMMMNKLVLAQ**WWW**SCL

**d**

KS-F-to-Y
KS-F-to-W

**Extended Data Fig. 1** | See next page for caption.

**Extended Data Fig. 1 | Shuffled killswitch sequences arrest the dynamics of nucleoli. a**. (left) Model of fsHMGB1 and shuffled sequences of the killswitch (KS) within the tested mEGFP-fsHMGB1 proteins. NoLS: nucleolar localization sequence. (middle) Quantification of the mobile and immobile fractions of mEGFP-HMGB1 proteins in nucleoli. (right) Mean GFP fluorescence of bleached nucleoli. Data are mean ± s.d. *P*-values are from Dunnett's post-hoc testing versus fsHMGB1-full length after one-way ANOVA. For FRAP plots, $P_{(\Delta killswitch)} = <2e-16$, $P_{(0F)} = <2e-16$, $P_{(Shuffled1)} = 0.002$, $P_{(Shuffled2)} = 0.18$, $P_{(Shuffled3)} = <2e-16$, $P_{(Shuffled4)} = 0.08$, $P_{(Shuffled5)} = 0.46$, $P_{(Shuffled6)} = 2.5e-5$, $P_{(Shuffled7)} = 3.3e-12$, $P_{(Shuffled8)} = <2e-16$, $P_{(Shuffled9)} = 0.77$, $P_{(Shuffled10)} = <2e-16$. For GFP intensity plots, $P_{(\Delta killswitch)} = 0.97$, $P_{(0F)} = 0.93$, $P_{(Shuffled1)} = 0.97$, $P_{(Shuffled2)} = 0.999$, $P_{(Shuffled3)} = > 0.9999$, $P_{(Shuffled4)} = 0.85$, $P_{(Shuffled5)} = 0.998$, $P_{(Shuffled6)} = > 0.9999$, $P_{(Shuffled7)} = > 0.9999$, $P_{(Shuffled8)} = >0.9999$, $P_{(Shuffled9)} = 0.997$, $P_{(Shuffled10)} = 0.9995$. ***:$P < 10^{-3}$, **:$P < 10^{-2}$. n = cells from four biologically independent experiments for Full length, Δkillswitch, 0 F, and two biologically independent experiments for Shuffled 1-10. **b**. Live cell fluorescence microscopy images of U2OS cells expressing ectopic mEGFP-HMGB1 and RFP-FIB1. The cell nucleus is highlighted with a dashed white line contour. Scale bar: 10 μm. **c**. (left) FRAP of mEGFP-HMGB1 and F-to-Y and F-to-W variants. Data are mean ± s.d. (middle) Quantification of the mobile and immobile fractions of mEGFP-HMGB1 proteins in nucleoli. Data are mean ± s.d. *P*-values are from Tukey's post-hoc test versus fsHMGB1-full length after one-way ANOVA. (right) Mean GFP fluorescence of bleached nucleoli from two independent experiment series. Data are mean ± s.d. For the middle plot, $P_{(\Delta killswitch)} = <1e-6$, $P_{(KS\_F-to-Y)} = <1e-6$, $P_{(KS\_F-to-W)} = <1e-6$; for the right plot, $P_{(\Delta killswitch)} = 0.97$, $P_{(KS\_F-to-Y)} = 0.91$, $P_{(KS\_F-to-W)} = 0.69$. n = 20 for all samples, except n = 15 for "Full length" and "ΔKillswitch", and n = 19 for F-to-W. ***:$P < 10$-3. **d**. Live cell fluorescence microscopy images of U2OS cells expressing ectopic mEGFP-HMGB1 proteins and RFP-FIB1. The contour of the cell nucleus is highlighted with a dashed white line. Scalebar: 10 μm.

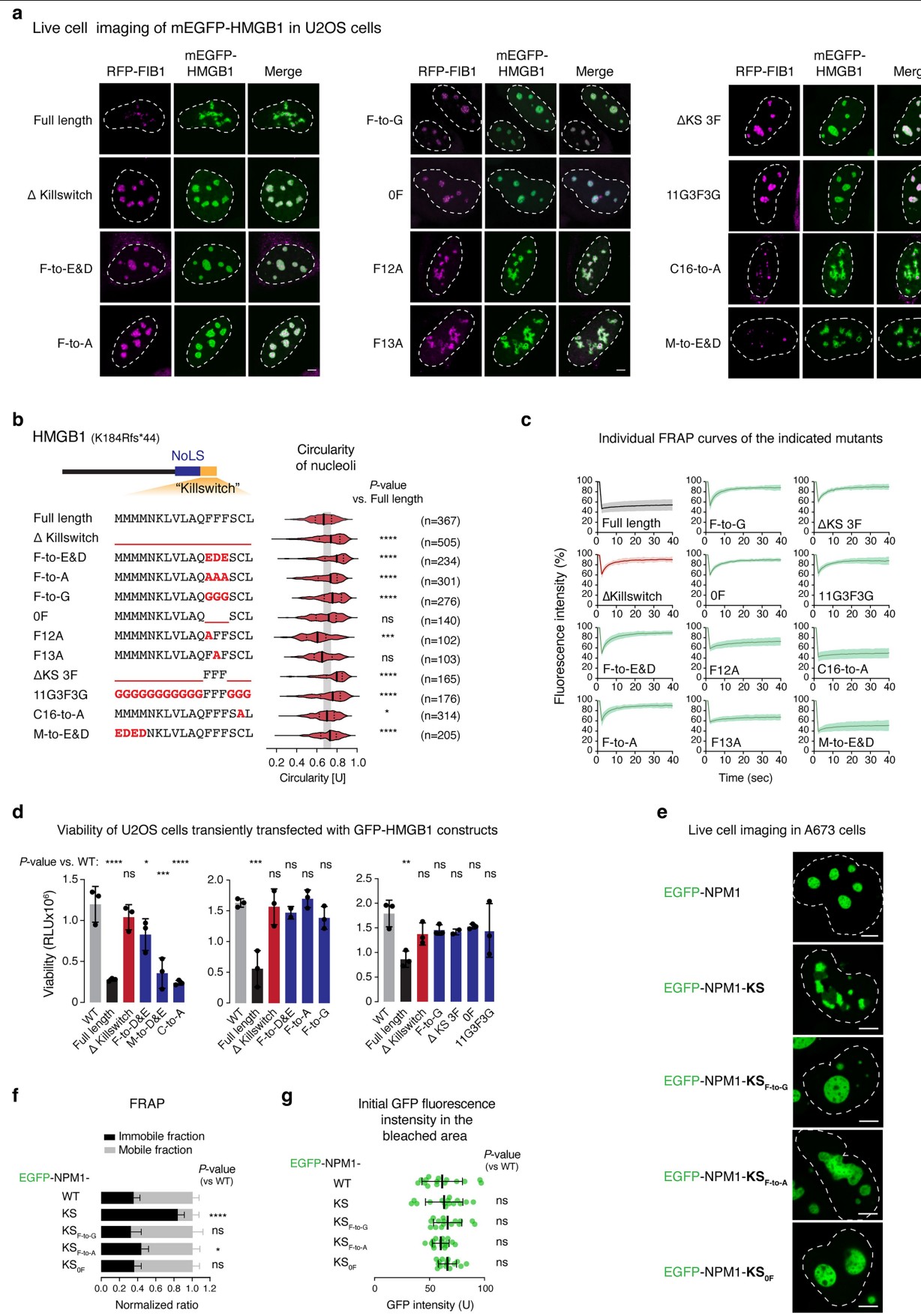

**a** Live cell imaging of mEGFP-HMGB1 in U2OS cells

**b** HMGB1 (K184Rfs*44)

NoLS
"Killswitch"

Circularity of nucleoli

|  | | *P*-value vs. Full length | |
|---|---|---|---|
| Full length | MMMMNKLVLAQFFFSCL | | (n=367) |
| Δ Killswitch | | **** | (n=505) |
| F-to-E&D | MMMMNKLVLAQ**EDE**SCL | **** | (n=234) |
| F-to-A | MMMMNKLVLAQ**AAA**SCL | **** | (n=301) |
| F-to-G | MMMMNKLVLAQ**GGG**SCL | **** | (n=276) |
| 0F | MMMMNKLVLAQ___SCL | ns | (n=140) |
| F12A | MMMMNKLVLAQ**A**FFSCL | *** | (n=102) |
| F13A | MMMMNKLVLAQF**A**FSCL | ns | (n=103) |
| ΔKS 3F | ___FFF___ | **** | (n=165) |
| 11G3F3G | **GGGGGGGGGGG**FFF**GGG** | **** | (n=176) |
| C16-to-A | MMMMNKLVLAQFFFS**A**L | * | (n=314) |
| M-to-E&D | **EDED**NKLVLAQFFFSCL | **** | (n=205) |

Circularity [U]

**c** Individual FRAP curves of the indicated mutants

Fluorescence intensity (%) — Time (sec)

Full length, F-to-G, ΔKS 3F
ΔKillswitch, 0F, 11G3F3G
F-to-E&D, F12A, C16-to-A
F-to-A, F13A, M-to-E&D

**d** Viability of U2OS cells transiently transfected with GFP-HMGB1 constructs

*P*-value vs. WT: **** ns * ****
ns ***

Viability (RLUx10⁶)

WT, Full length, Δ Killswitch, F-to-D&E, M-to-D&E, C-to-A
*** ns ns ns
WT, Full length, Δ Killswitch, F-to-D&E, F-to-A, F-to-G
** ns ns ns ns ns
WT, Full length, Δ Killswitch, F-to-G, Δ KS 3F, 0F, 11G3F3G

**e** Live cell imaging in A673 cells

EGFP-NPM1
EGFP-NPM1-**KS**
EGFP-NPM1-**KS**F-to-G
EGFP-NPM1-**KS**F-to-A
EGFP-NPM1-**KS**0F

**f** FRAP

■ Immobile fraction
▨ Mobile fraction

EGFP-NPM1-

*P*-value (vs WT)

| WT | |
| KS | **** |
| KS F-to-G | ns |
| KS F-to-A | * |
| KS 0F | ns |

Normalized ratio

**g** Initial GFP fluorescence instensity in the bleached area

EGFP-NPM1-

*P*-value (vs WT)

| WT | |
| KS | ns |
| KS F-to-G | ns |
| KS F-to-A | ns |
| KS 0F | ns |

GFP intensity (U)

**Extended Data Fig. 2 |** See next page for caption.

**Extended Data Fig. 2 | The killswitch arrests the dynamics of nucleolar condensates, alters their morphology and reduces cell viability. a**. Live cell fluorescence microscopy images of U2OS cells expressing ectopic mEGFP-fsHMGB1 variants and RFP-FIB1. The cell nucleus is highlighted with a dashed white line contour. Scale bar: 5 μm. **b**. (left) Model of fsHMGB1 and sequences of the KS variants within the tested mEGFP-HMGB1 proteins. (right) Quantification of circularity of nucleoli in cells expressing ectopic mEGFP-HMGB1 KS variants. *P*-values from Dunnett's multiple comparison test versus fsHMGB1-full length after one-way ANOVA. $P_{(\Delta killswitch)} = {<}0.0001$, $P_{(F\text{-to-}E\&D)} = {<}0.0001$, $P_{(F\text{-to-}A)} = 0.0009$, $P_{(F\text{-to-}G)} = {<}0.0001$, $P_{(0F)} = 0.0972$, $P_{(F12A)} = 0.0003$, $P_{(F13A)} = {>}0.9999$, $P_{(\Delta KS\text{-}3F)} = {<}0.0001$, $P_{(11G3F3G)} = {<}0.0001$, $P_{(C16\text{-to-}A)} = 0.0346$, $P_{(M\text{-to-}E\&D)} = {<}0.0001$. ****:$P < 10^{-4}$, ***:$P < 10^{-3}$, *:$P < 0.05$. **c**. FRAP of the indicated mEGFP-HMGB1 variants. The quantification of the corresponding mobile/immobile fraction ratios are shown in Fig. 1e. Data are mean ± s.d. **d**. Cell viability of U2OS cell expressing mEGFP-HMGB1 proteins. Data are mean ± s.d. *P*-values from one-way ANOVA Dunnett's multiple comparison test versus fsHMGB1-full length. (Left): $P_{(Full length)} = {<}0.0001$, $P_{(\Delta killswitch)} = 0.5987$, $P_{(F\text{-to-}D\&E)} = 0.0320$, $P_{(M\text{-to-}D\&E)} = {<}0.0001$, $P_{(C\text{-to-}A)} = {<}0.0001$;

(Middle): $P_{(Full length)} = {<}0.0001$, $P_{(\Delta killswitch)} = 0.9975$, $P_{(F\text{-to-}D\&E)} = 0.8919$, $P_{(F\text{-to-}A)} = 0.9967$, $P_{(F\text{-to-}G)} = {<}0.5244$. (Right): $P_{(Full length)} = 0.0046$, $P_{(\Delta killswitch)} = 0.2873$, $P_{(F\text{-to-}G)} = 0.4834$, $P_{(\Delta KS\text{-}3F)} = 0.4818$, $P_{(0F)} = 0.6900$, $P_{(11G3F3G)} = 0.4578$. n = three (except two for F-to-D&E from the middle plot and ΔKS-3F from the right plot) biologically independent experiments. **e**. Representative live cell fluorescence microscopy images of A673 cells expressing EGFP-NPM1-WT and -KS variants 24 h after doxycycline induction. The cell nucleus is highlighted with a dashed white line contour. The experiments were repeated independently twice with similar results. Scale bar: 5 μm. **f**. Quantification of the mobile and immobile fractions of EGFP-NPM1 proteins. Data are mean ± s.d. n = 15 cells for all samples from two biologically independent experiments. *P*-values from Dunnett's multiple comparisons test versus EGFP-NPM1-WT after one-way ANOVA. $P_{(KS)} = {<}0.0001$, $P_{(KS\_F\text{-to-}G)} = 0.76$, $P_{(KS\_F\text{-to-}A)} = 0.03$, $P_{(KS\_0F)} = 1.00$. **g**. Mean GFP fluorescence of the bleached area. Data are mean ± s.d. n = 15 cells for all samples from two biologically independent experiments. *P*-values from Dunnett's multiple comparisons test versus EGFP-NPM1-WT after one-way ANOVA.

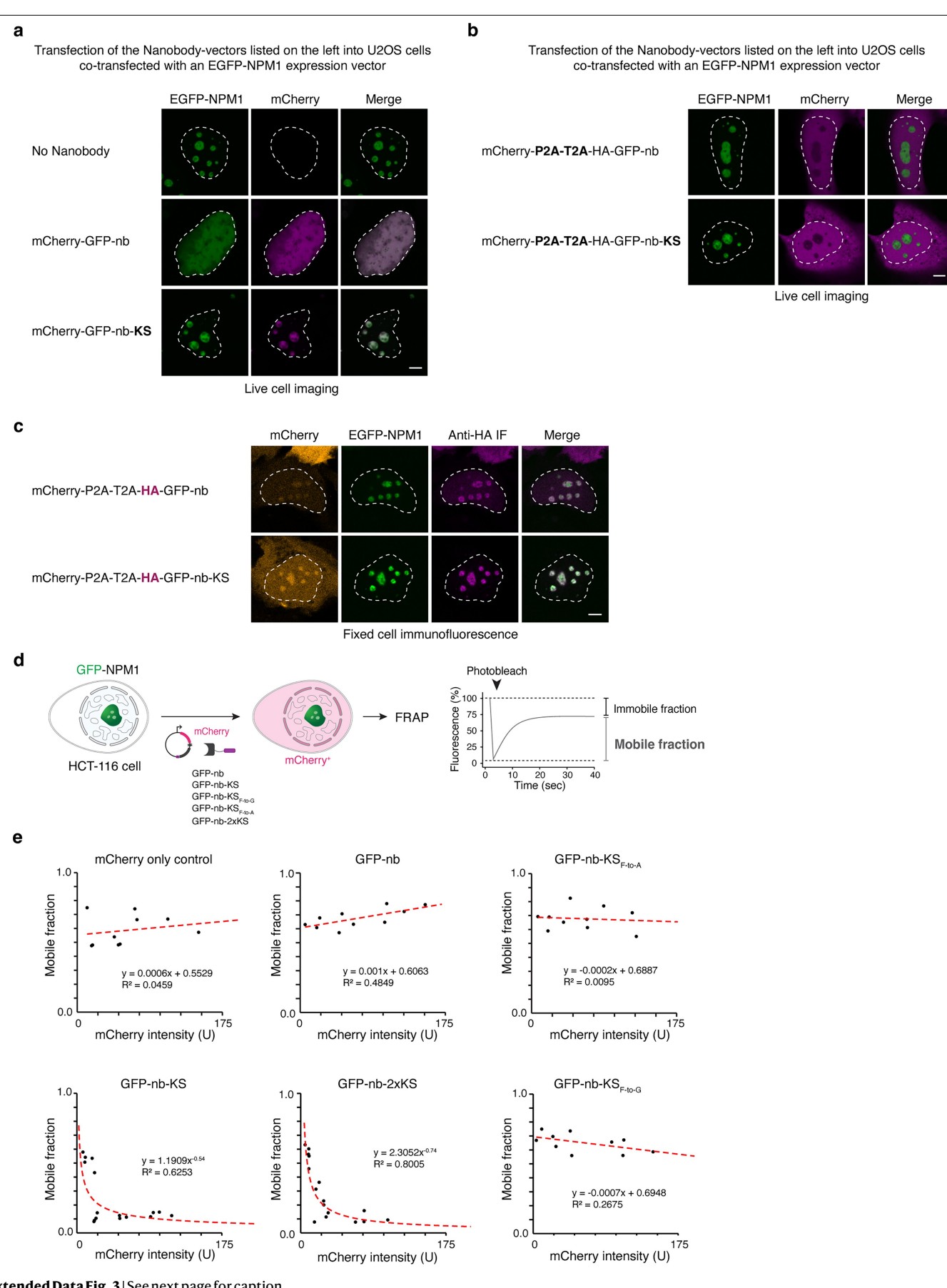

**Extended Data Fig. 3** | See next page for caption.

**Extended Data Fig. 3 | Effects of nanobody-killswitch on NPM1 mobility in nucleoli. a**. Live cell fluorescence microscopy images of U2OS cells expressing ectopic EGFP-NPM1 alone or co-transfected with mCherry-GFP-nanobody fusion proteins. The cell nucleus is highlighted with a dashed white line contour. Scale bar: 5 μm. The experiment was repeated independently twice with similar results. **b**. Live cell fluorescence microscopy images of U2OS cells expressing ectopic EGFP-NPM1 and GFP-nanobody (GFP-nb) constructs with cleaved mCherry reporter. The cell nucleus is highlighted with a dashed white line contour. Scale bar: 5 μm. The experiment was repeated independently twice with similar results. **c**. Fixed-cell immunofluorescence of cells expressing ectopic EGFP-NPM1 and GFP-nb constructs with cleaved mCherry reporter. GFP-nb is recognized through HA-tag using anti-HA-tag antibody. The cell nucleus is highlighted with a dashed white line contour. Scale bar: 5 μm. The experiment was repeated independently twice with similar results. **d**. Schematic model of the nanobody-based recruitment system and procedure to measure immobile and mobile fractions of GFP-NPM1 molecules. **e**. Quantification of mobile fraction of GFP-NPM1 as a function fluorescence intensity of the mCherry reporter (as a proxy for nanobody concentration). Curves from linear and power regression models are shown as dashed red lines. n = 10 for all samples, except n = 16 for GFP-nb-KS and −2xKS.

**Extended Data Fig. 4** | See next page for caption.

**Extended Data Fig. 4 | The killswitch can be recruited via different epitope tags to immobilize NPM1 in nucleoli. a**. Schematic of the experiment with four different Nanobodies. Four different GFP-NPM1 transgenes were created: one with an ALFA-tag, one with a V5-tag, one with a VHH05-tag and one with no tag at the N-terminus. The constructs were co-transfected with the corresponding nanobody from a vector that also encodes an mCherry reporter. **b**. FRAP of EGFP-NPM1 tagged with the indicated epitope tags when targeted by their corresponding nanobodies. Data are mean ± s.d. n = 20 for all samples. **c**. (left) Mean GFP fluorescence of bleached area. (right) Mean mCherry reporter intensity in nuclei of investigated cells. Data are mean ± s.d. $P$-values are from Tukey's multiple comparison test against Nb-only control after one-way ANOVA. For GFP intensity plots, in EGFP-NPM1 condition: $P_{(Neg\_ctrl)} = 0.996$,

$P_{(KS)} = 0.64$, $P_{(KS\_F-to-G)} = 0.91$; in ALFA-EGFP-NPM1 condition: $P_{(Neg\_ctrl)} = 0.45$, $P_{(KS)} = 0.47$; in V5-EGFP-NPM1 condition: $P_{(Neg\_ctrl)} = 0.99$, $P_{(KS)} = 0.64$, $P_{(KS\_F-to-G)} = 1.0$; in VHH05-EGFP-NPM1 condition: $P_{(Neg\_ctrl)} = 1.0$, $P_{(KS)} = 0.78$, $P_{(KS\_F-to-G)} = 0.73$. For mCherry intensity plots, in GFP-nb condition: $P_{(Neg\_ctrl)} = <0.001$, $P_{(KS)} = 0.05$, $P_{(KS\_F-to-G)} = 0.999$; in ALFA-EGFP-NPM1 condition: $P_{(Neg\_ctrl)} = <0.0001$, $P_{(KS)} = 0.77$; in V5-EGFP-NPM1 condition: $P_{(Neg\_ctrl)} = <0.0001$, $P_{(KS)} = 0.97$, $P_{(KS\_F-to-G)} = 0.75$; in VHH05-EGFP-NPM1 condition: $P_{(Neg\_ctrl)} = <0.0001$, $P_{(KS)} = 0.32$, $P_{(KS\_F-to-G)} = 0.48$. n = 20 cells for all samples from two biologically independent experiments. **d**. (left) Quantification of mobile fraction of GFP-NPM1 as a function fluorescence intensity of GFP-NPM1. (right) Quantification of mobile fraction of GFP-NPM1 as a function fluorescence intensity of mCherry reporter.

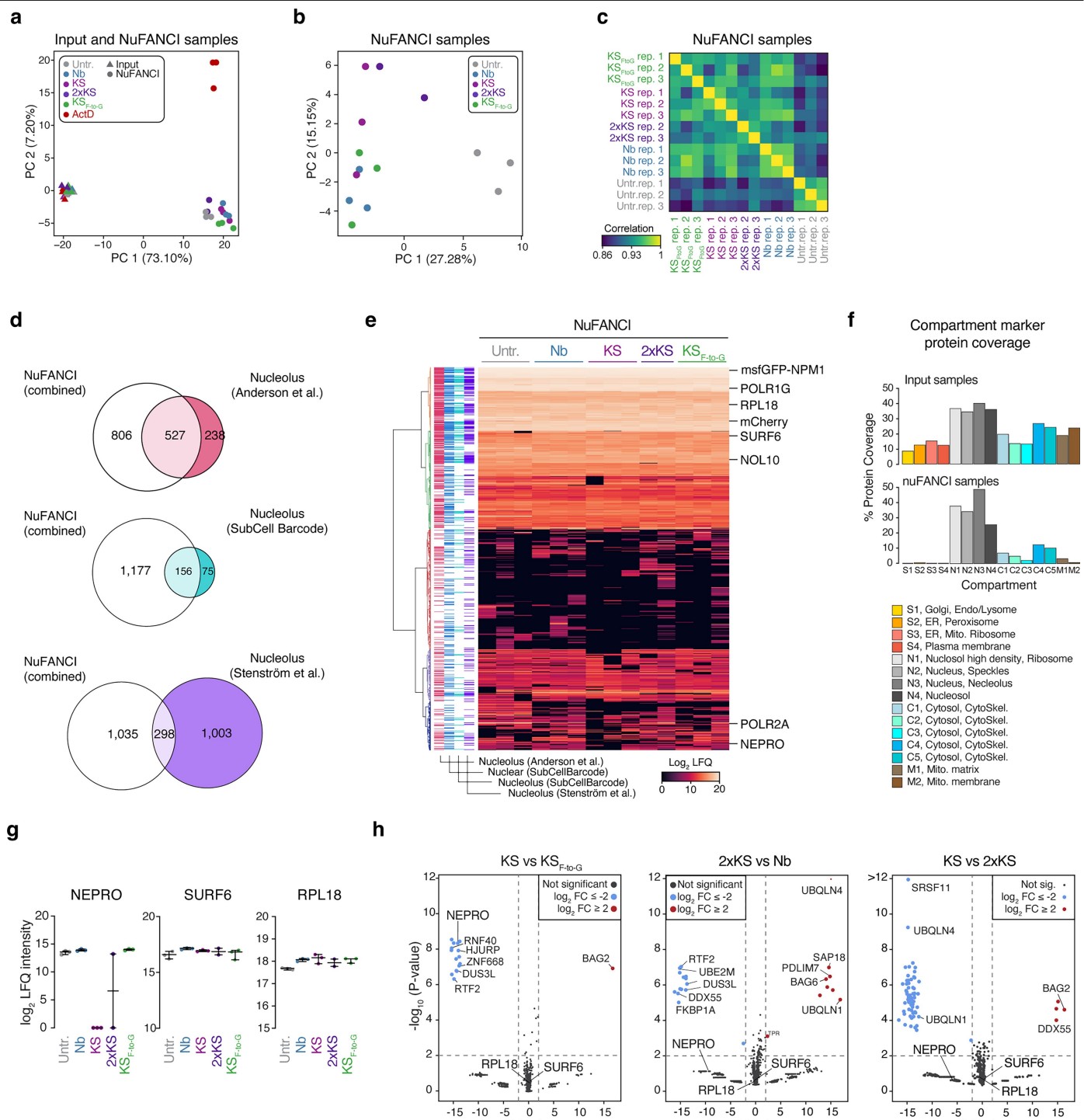

**Extended Data Fig. 5 | NuFANCI proteome analysis of nucleoli targeted with killswitch. a**. Principal Component (PC) analysis of NuFANCI-isolated nucleoli from untransfected cells, ActD-treated, GFP-nb, GFP-nb-KS, GFP-nb-2xKS, GFP-nb-KS_F-to-G, and their respective input samples. The expression values were then standardized. **b**. Principal component analysis of NuFANCI-isolated nucleoli without input and ActD-treated samples. **c**. Correlation plot of NuFANCI samples using Pearson's correlation coefficient. **d**. Euler diagrams showing the overlap of NuFANCI-detected proteins in nucleolus reference proteomes. **e**. Heatmap showing the protein expression profiles of NuFANCI-isolated nucleoli. Protein markers used in nucleolus isolation, mCherry and msfGFP-NPM1, some common nucleolar proteins, and protein marker (NEPRO) tested by IF are annotated on

the right. Proteins have also been annotated based on the proteome of isolated nucleoli from previous proteomics data on the left. **f**. Barplots showing the percentage of proteins in the pooled input (top) or pooled nuFANCI samples (bottom) mapping to individual compartments annotated in the SubCellBarcode database. **g**. Quantification of NEPRO, SURF6, and RPL18 expression in NuFANCI samples shown in **b**. Data are mean ± s.d. n = 3 biologically independent experiments. **h**. Volcano plots of NuFANCI pairwise-comparisons. Significance values are from one-sided Student's t-tests. Non-significant proteins are shaded in grey, whereas enriched and depleted are shaded in red and blue, respectively. Protein marker NEPRO was tested by IF.

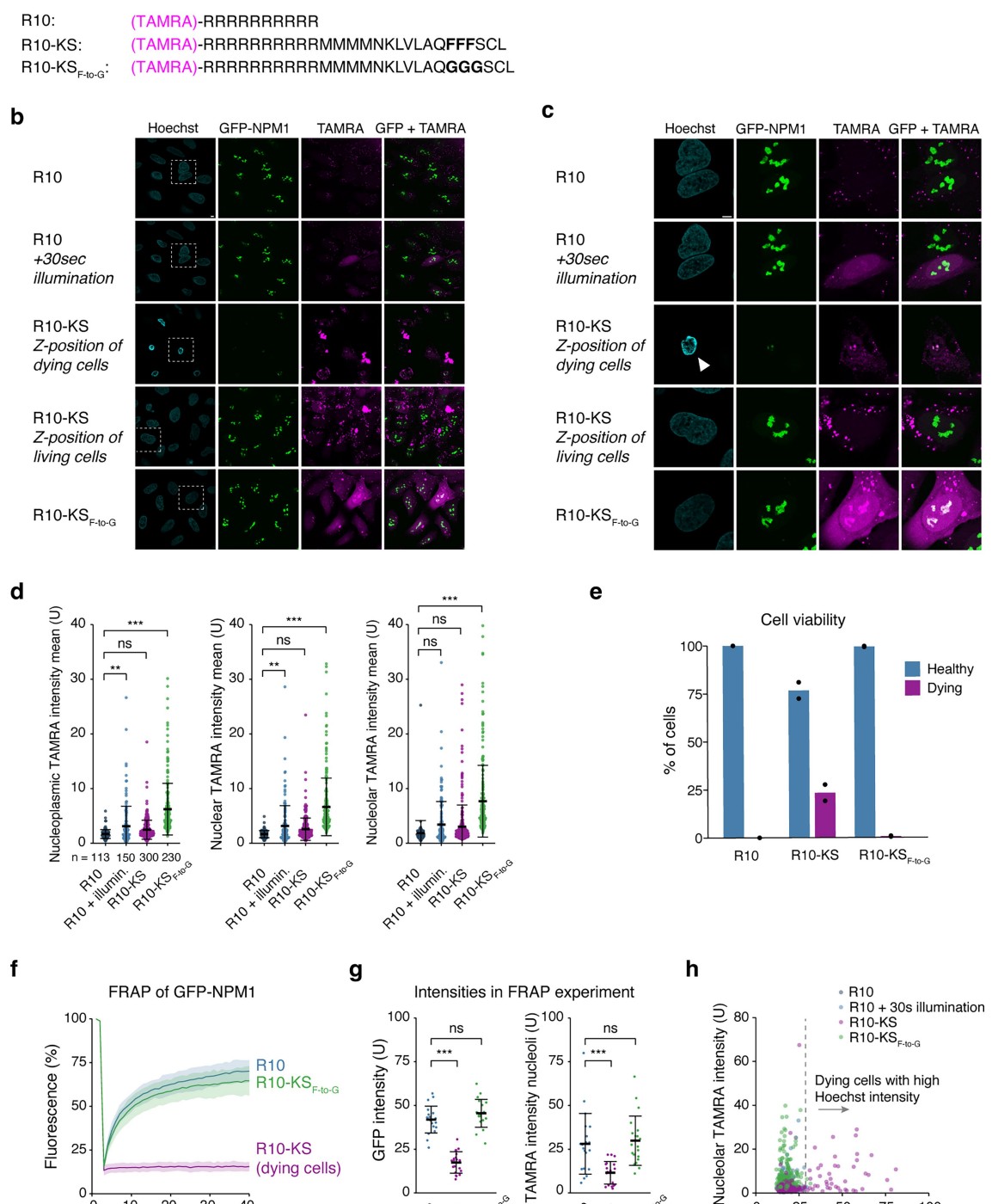

**a**

R10:            (TAMRA)-RRRRRRRRRR
R10-KS:         (TAMRA)-RRRRRRRRRRRMMMMNKLVLAQ**FFF**SCL
R10-KS_{F-to-G}:  (TAMRA)-RRRRRRRRRRRMMMMNKLVLAQ**GGG**SCL

**Extended Data Fig. 6 | Exposing cells to nucleoli-targeting cell penetrating peptides conjugated with the killswitch is cytotoxic. a**. Sequences of cell penetrating peptides used. All peptides had N-terminal TAMRA-dye for visualization **b**. Live cell fluorescence microscopy images of U2OS cells expressing GFP-NPM1 from endogenous locus. Cells were exposed to 3 µM peptides for 30 min and imaged 3 h after peptide addition. Scale bar: 5 µm. **c**. Zoom-in to regions highlighted with dashed square on **b**. Dying cell with nucleolar R10-KS peptide, condensed nuclei and increased Hoechst intensity highlighted with white arrow. Scale bar: 5 µm. **d**. Quantification of mean TAMRA signal intensities in nucleoplasmic areas (left), nuclear areas (middle) and nucleolar areas (right). Data are mean ± s.d. $P$-values are from Tukey's multiple comparison test after one-way ANOVA. (Left): $P_{(R10\,vs\,R10+illumin)} = 0.002$, $P_{(R10\,vs\,R10-KS)} = 0.12$, $P_{(R10\,vs\,R10-KS\_F-to-G)} = <0.001$; (middle): $P_{(R10\,vs\,R10+illumin)} = 0.003$,

$P_{(R10\,vs\,R10-KS)} = 0.08$, $P_{(R10\,vs\,R10-KS\_F-to-G)} = <0.001$; (right): $P_{(R10\,vs\,R10+illumin)} = 0.08$, $P_{(R10\,vs\,R10-KS)} = 0.08$, $P_{(R10\,vs\,R10-KS\_F-to-G)} = <0.001$. n represents number of cells examined from two biologically independent experiments. **e**. Percentage of healthy and dying cells from cells exposed to 3 µM peptides for 30 min, imaged 3 h after peptide addition. Measurements are from two biological replicates. **f**. FRAP of GFP-NPM1 in cells with nucleolar TAMRA signal after exposure to cell penetrating peptides. Data are mean ± s.d. n = 20 for all samples. **g**. Mean GFP fluorescence (left) and TAMRA fluorescence (right) of bleached area. Data are mean ± s.d. $P$-values are from Tukey's multiple comparison test against R10 control after one-way ANOVA. (Left): $P_{(R10-KS)} = <0.0001$, $P_{(R10-KS\_F-to-G)} = 0.27$; (right): $P_{(R10-KS)} = 0.0007$, $P_{(R10-KS\_F-to-G)} = 0.91$. n = 20 cells for all samples from two biologically independent experiments. **h**. Scatterplot depicting nucleolar TAMRA intensity as function mean nucleolar Hoechst intensity.

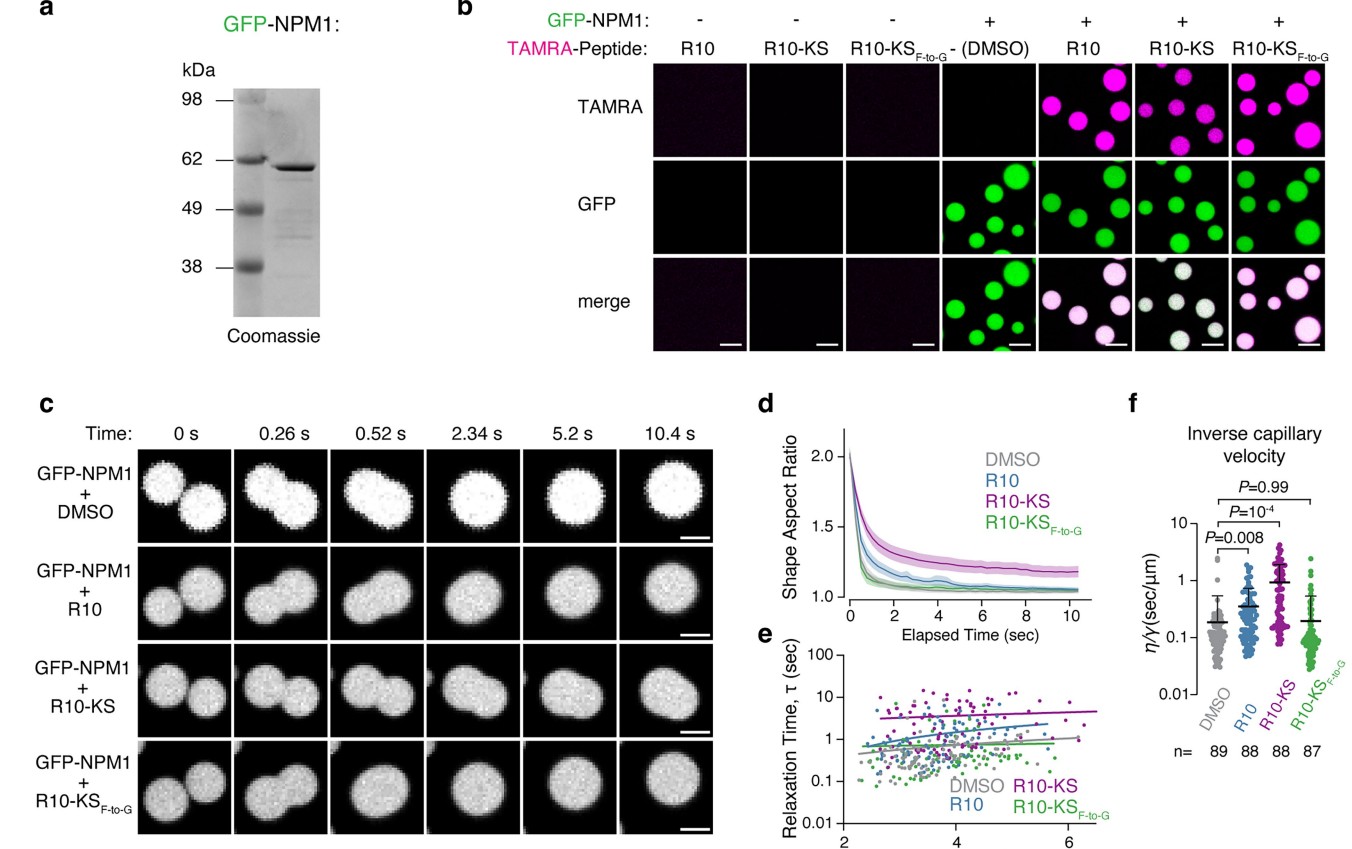

**Extended Data Fig. 7 | The killswitch increases the viscosity of NPM1 in vitro droplets. a**. Coomassie gel of purified recombinant msfGFP-NPM1. The experiment was performed once. **b**. Representative images of the in vitro mixing of msfGFP-NPM1 with DMSO, TAMRA-R10, TAMRA-R10-KS, or TAMRA-R10-KS_{F-to-G} peptides. This experiment was performed independently twice with similar results. Scale bar: 5 μm. **c**. Representative images of the in vitro condensate fusion events at different timepoints for msfGFP-NPM1 mixed with DMSO, TAMRA-R1, TAMRA-R10-KS, or TAMRA-R10-KS_{F-to-G} peptides. The experiment was performed independently three times with similar results. Scale bar: 3 μm. **d**. Plot of the Shape Aspect Ratio vs. Elapsed Time of in vitro condensate fusion events for msfGFP-NPM1 mixed with DMSO (n = 89), TAMRA-R10 (n = 88), TAMRA-R10-KS (n = 88), or TAMRA-R10-KS_{F-to-G} (n = 87) peptides. The curves show the mean value at each time point; the shade around the curves show the 95% CI. **e**. Plot of the Relaxation time ($\tau$) vs. Length Scale ($\ell$) of in vitro condensate fusion events for msfGFP-NPM1 mixed with DMSO (n = 89), TAMRA-R10 (n = 88), TAMRA-R10-KS (n = 88), or TAMRA-R10-KS_{F-to-G} (n = 87) peptides. The lines are simple linear regression fits for each condition. **f**. Plot for ratio of viscosity ($\eta$) to surface tension ($\gamma$) (inverse capillary velocity) of in vitro condensate fusion events for msfGFP-NPM1 mixed with DMSO (n = 89), TAMRA-R10 (n = 88), TAMRA-R10-KS (n = 88), or TAMRA-R10-KS_{F-to-G} (n = 87) peptides. "n" represents fusion events for each condition examined over three independent experiments. Data are mean ± s.d. P-values are from Dunnett's multiple T3 comparisons test versus DMSO condition after one-way ANOVA.

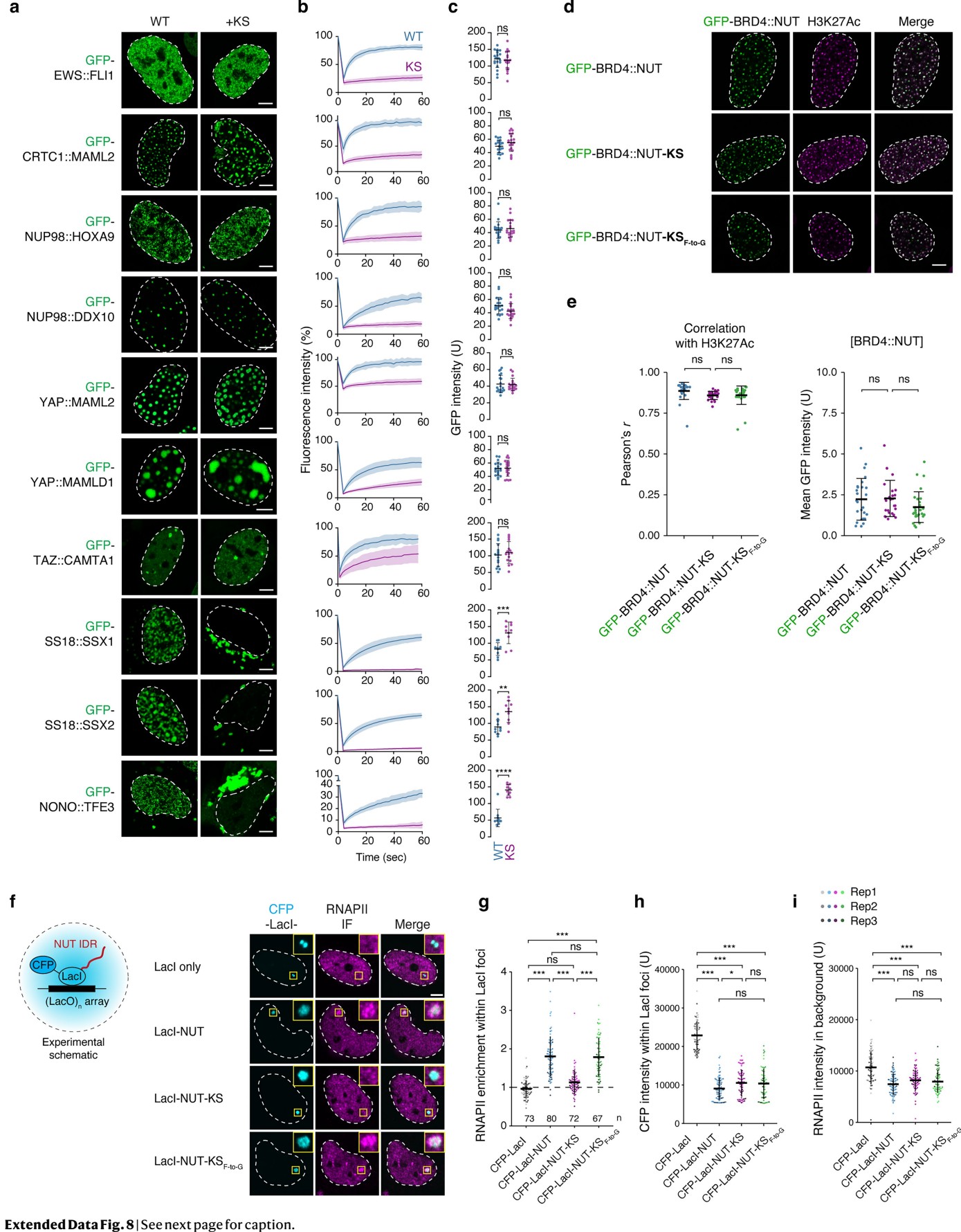

**Extended Data Fig. 8** | See next page for caption.

**Extended Data Fig. 8 | The killswitch arrests dynamics of fusion oncoprotein condensates. a**. Live cell fluorescence microscopy of U2OS expressing ectopic fusion onco-proteins with or without C-terminal KS. EWS::FLI1 was expressed in MCF7 cells, and YAP::MAMLD1 was expressed in C2C12 cells. The cell nucleus is highlighted with a dashed white line contour. For SS18::SSX1, SS18::SSX2 and NONO::TFE3, genetic fusion of the killswitch leads to cytoplasmic aggregation of the protein. Note that targeting the killswitch with the anti-GFP-nanobody to the already formed condensates has no such effect (see Fig. 3a). Scale bar: 5 μm. **b**. FRAP of GFP signal of GFP-fusion oncoproteins. Data are mean ± s.d. n = 20 for EWS::FLI1, CRTC1::MAML2, NUP98::HOXA9, NUP98::DDX10, YAP::MAML2 and YAP::MAMLD1; n = 16 for TAZ::CAMTA1; n = 10 for SS18::SSX1, SS18::SSX2 and NONO::TFE3. **c**. Quantification of mean GFP fluorescence intensity in the bleached area of GFP-fusion oncoproteins. Data are mean ± s.d. n = 20 for EWS::FLI1, CRTC1::MAML2, NUP98::HOXA9, NUP98::DDX10, YAP::MAML2-WT and YAP::MAMLD1; n = 19 YAP-MAML2-KS; n = 16 for TAZ::CAMTA1; n = 10 for SS18::SSX1, SS18::SSX2 and NONO::TFE3. n represents number of cells from two biologically independent experiments. *P*-values are from unpaired two-tailed t test. $P_{(EWS::FLI1)}$ = 0.60, $P_{(CTRC1::MAML2)}$ = 0.14, $P_{(NUP98::HOXA9)}$ = 0.60, $P_{(NUP98::DDX10)}$ = 0.03, $P_{(YAP::MAML2)}$ = 0.80, $P_{(YAP::MAMLD1)}$ = 0.83, $P_{(TAZ::CAMTA1)}$ = 0.54, $P_{(SS18::SSX1)}$ = 0.0007, $P_{(SS18::SSX2)}$ = 0.0015, $P_{(NONO::TFE3)}$ = < 0.0001. **d**. Fixed cell immunofluorescence of U2OS cells expressing ectopic EGFP-BRD4::NUT WT or -KS variants stained for H3K27Ac chromatin modification. Scale bar: 5 μm. **e**. (left) Correlation of GFP-BRD4::NUT and H3K27Ac staining intensities measured by Pearson's correlation coefficients. Each dot represents measurement from one cell nucleus. n = 26,

26 and 29 for -WT, -KS and -KS$_{F-to-G}$ variants, respectively. *P*-values are from Tukey's post-hoc test after one-way ANOVA. standard deviation. (right) Quantification of mean GFP fluorescence intensity in the nuclei of U2OS cells expressing ectopic GFP-BRD4::NUT variants analysed for colocalization in **d**. *P*-values are from Tukey's post-hoc test after one-way ANOVA. (Left): $P_{(GFP-BRD4::NUT\ vs\ GFP-BRD4::NUT-KS)}$ = 0.09, $P_{(GFP-BRD4::NUT-KS\ vs\ GFP-BRD4::NUT-KS\_F-to-G)}$ = 0.99; (right): $P_{(GFP-BRD4::NUT\ vs\ GFP-BRD4::NUT-KS)}$ = 0.99, $P_{(GFP-BRD4::NUT-KS\ vs\ GFP-BRD4::NUT-KS\_F-to-G)}$ = 0.18. Data are mean ± s.d. **f**. Fixed cell immunofluorescence images of U2OS 2-6-3 cells ectopically expressing CFP-LacI (control) or CFP-LacI-NUT with or without KS and stained for endogenous RNAPII. **g**. Enrichment of RNAPII intensity in the LacI foci. Data are mean ± s.d. *P*-values are from Tukey's post-hoc test after one-way ANOVA. $P_{(NUT\ vs\ control)}$ = <0.001, $P_{(NUT-KS\ vs\ control)}$ = 0.06, $P_{(NUT-KS\_F-to-G\ vs\ control)}$ = <0.001, $P_{(NUT-KS\ vs\ NUT)}$ = < 0.001, $P_{(NUT-KS\_F-to-G\ vs\ NUT)}$ = 0.99, $P_{(NUT-KS\_F-to-G\ vs\ NUT-KS)}$ = < 0.001. n represents number of cells examined from three biologically independent experiments. **h**. Intensity of CFP intensity at LacI foci. Data are mean ± s.d. *P*-values are from Tukey's post-hoc test after one-way ANOVA. $P_{(NUT\ vs\ control)}$ = <0.001, $P_{(NUT-KS\ vs\ control)}$ = <0.001, $P_{(NUT-KS\_F-to-G\ vs\ control)}$ = <0.001, $P_{(NUT-KS\ vs\ NUT)}$ = 0.04, $P_{(NUT-KS\_F-to-G\ vs\ NUT)}$ = 0.10, $P_{(NUT-KS\_F-to-G\ vs\ NUT-KS)}$ = 0.99. n represents number of cells examined from three biologically independent experiments. **i**. Background RNAPII intensity in LacO tethering experiment. For panels **g-i**, *P*-values are from Tukey's post-hoc test after one-way ANOVA. Data are mean ± s.d. $P_{(NUT\ vs\ control)}$ = <0.001, $P_{(NUT-KS\ vs\ control)}$ = <0.0001, $P_{(NUT-KS\_F-to-G\ vs\ control)}$ = <0.0001, $P_{(NUT-KS\ vs\ NUT)}$ = 0.20, $P_{(NUT-KS\_F-to-G\ vs\ NUT)}$ = 0.56, $P_{(NUT-KS\_F-to-G\ vs\ NUT-KS)}$ = 0.93. "n" represents number of cells examined from three biologically independent experiments.

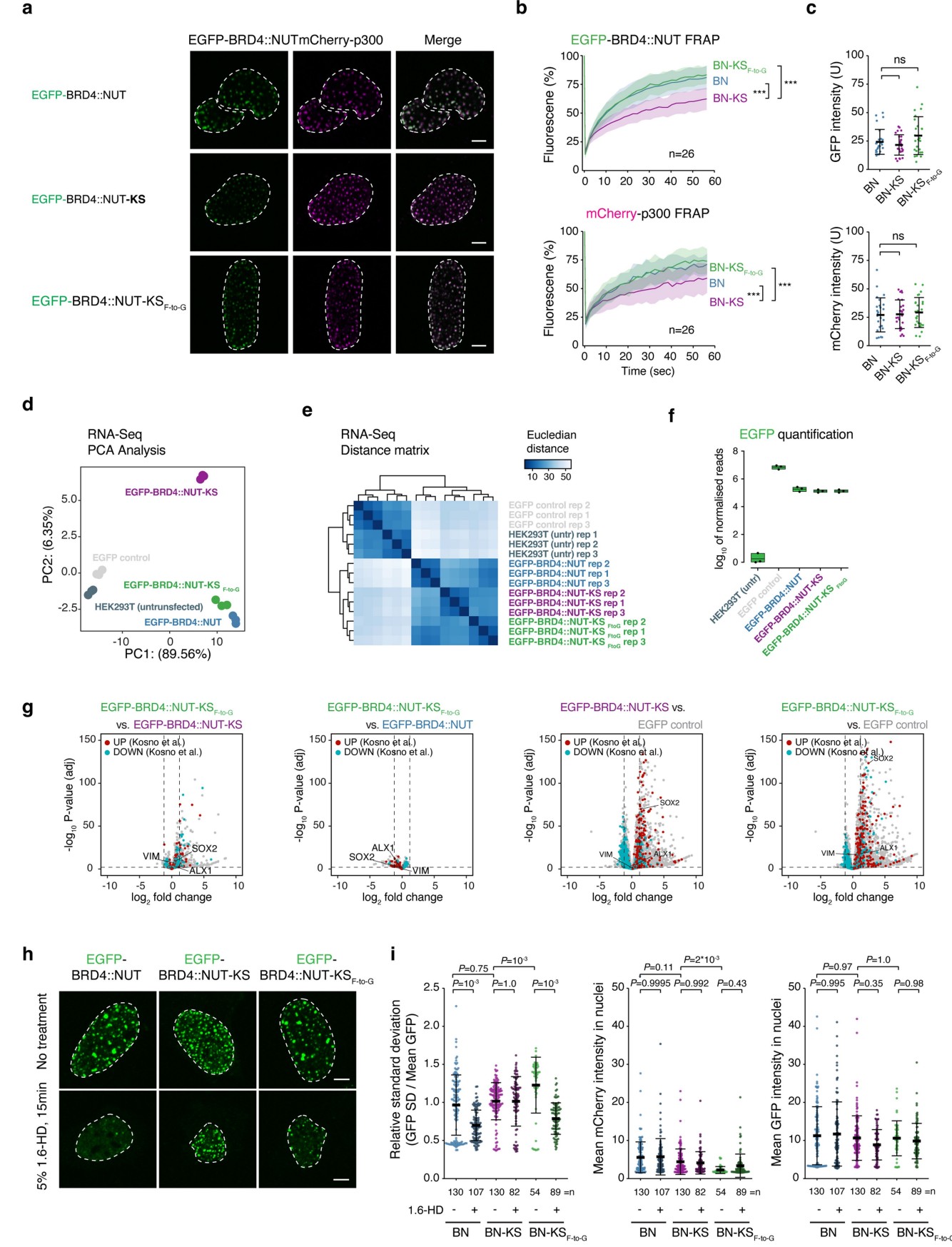

**Extended Data Fig. 9** | See next page for caption.

**Extended Data Fig. 9 | Additional BRD4::NUT data. a**. (left) Representative live cell fluorescence microscopy of U2OS expressing ectopic EGFP-BRD4::NUT with or without C-terminal KS together with mCherry-p300. Scale bar: 5 µm. **b**. FRAP of the EGFP-BRD4::NUT proteins, and FRAP of mCherry-p300. Data are mean ± s.d. $P$-values are from Tukey's multiple comparison test after one-way ANOVA. For the last timepoint, (top): $P_{(BN-KS\,vs\,BN)} = <10^{-5}$, $P_{(BN-KS\,vs\,BN-KS\_F-to-G)} = <10^{-5}$; (bottom): $P_{(BN-KS\,vs\,BN)} = 0.0005$, $P_{(BN-KS\,vs\,BN-KS\_F-to-G)} = <10^{-4}$. n = 26 cells for all samples from two biologically independent experiments. **c**. Quantification of GFP fluorescence intensity and mCherry fluorescence intensity in bleached areas. Data are mean ± s.d. P-values are from Tukey's multiple comparison test after one-way ANOVA. n = 26 cells for all samples from two biologically independent experiments. **d**. Principal component analysis of the RNA-Seq expression profiles of untransfected HEK293T cells, EGFP-control, EGFP-BRD4::NUT-WT, EGFP-BRD4::NUT-KS, and EGFP-BRD4::NUT-KS$_{F-to-G}$ (PC1 vs. PC2) **e**. Sample distance matrix calculated using Euclidean distance. **f**. Quantification of EGFP expression using normalized counts from the DEseq2. For the box plots, the center line shows the median, the bounds of the box correspond to interquartile (25th–75th) percentile, and whiskers extend to Q3 + 1.5× the interquartile range and Q1 − 1.5× the interquartile range. n = 3 biologically independent experiments. **g**. Differential expression analysis of EGFP-BRD4::NUT-KS$_{F-to-G}$ compared to EGFP-BRD4::NUT-KS (left), EGFP-BRD4::NUT-KS$_{F-to-G}$ compared to EGFP-BRD4::NUT-WT (middle left), EGFP-BRD4::NUT- KS compared to EGFP-control (middle-right), and EGFP-BRD4::NUT-KS$_{F-to-G}$ compared to EGFP-control (right). The differentially expressed proteins identified by Kosno et al.[87] was used to colour the genes. Referred upregulated genes are coloured in red and downregulated in blue. $P$-values were determined using the Benjamini–Hochberg method. **h**. Fixed cell imaging of U2OS cells ectopically expressing EGFP-BRD4::NUT, EGFP-BRD4::NUT-KS and EGFP-BRD4::NUT-KS$_{F-to-G}$ and cleavable mCherry reporter with or without treatment with 5% 1,6-hexanediol. **i**. Quantification of effect of 1,6-hexanediol to granularity of GFP signal for cells shown in panel h represented by relative standard deviation for nuclear GFP (left). Mean intensities of cleavable mCherry reporter (middle) and mean nuclear GFP intensity (right). P-values are from Tukey's post-hoc test after one-way ANOVA. Data are mean ± s.d. "n" represents number of cells examined from two biologically independent experiments.

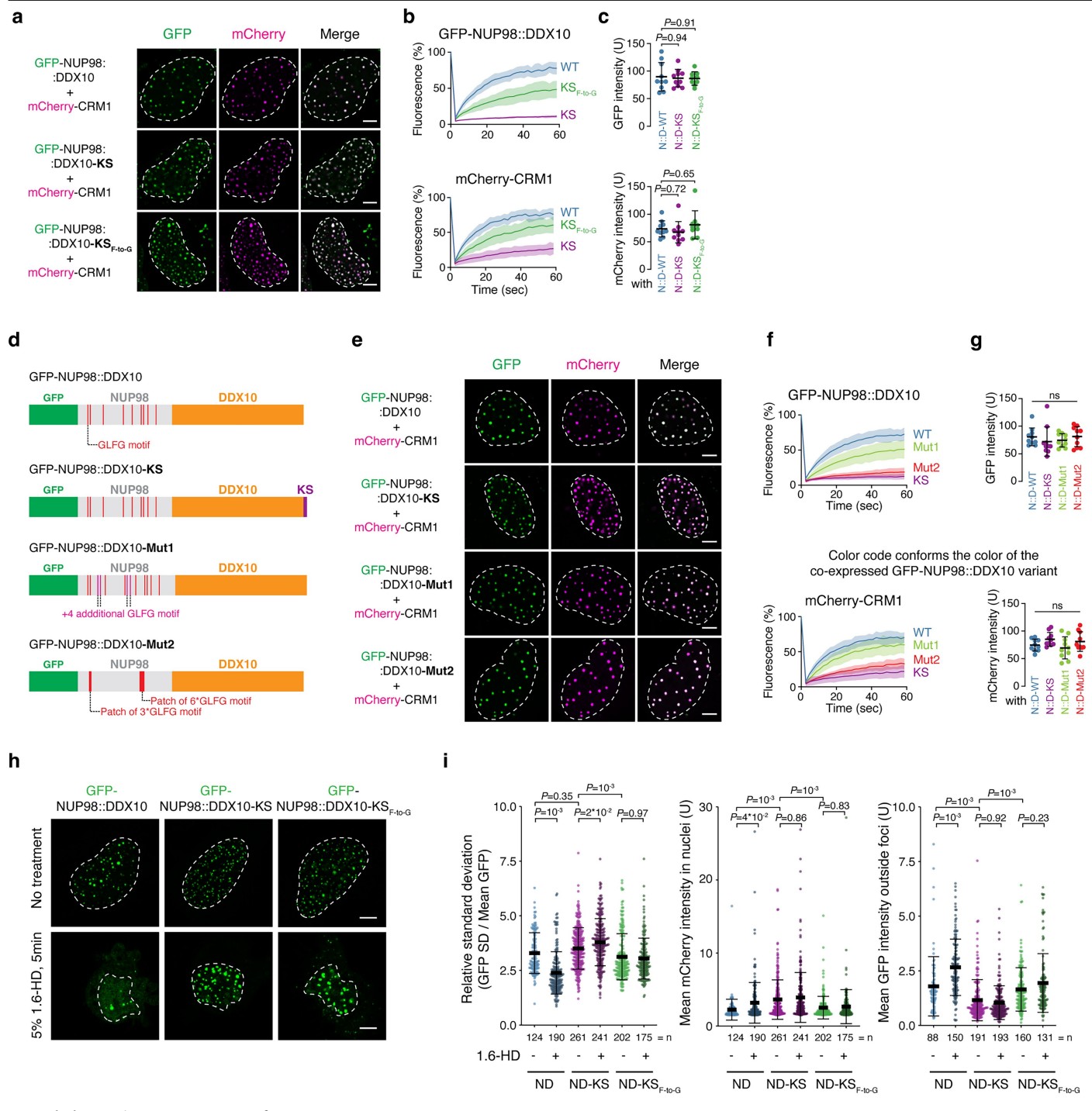

**Extended Data Fig. 10** | See next page for caption.

**Extended Data Fig. 10 | NUP98::DDX10 mutagenesis and 1,6-hexanediol treatment on cells expressing GFP-NUP98::DDX10 condensates targeted with the killswitch. a**. Live cell fluorescence microscopy images of U2OS cells expressing GFP-NUP98::DDX10 constructs each co-transfected with mCherry-CRM1. Scale bar: 5 µm. **b**. FRAP of GFP-NUP98::DDX10 (top), and of mCherry-CRM1 (bottom) in GFP-NUP98::DDX10 condensates in U2OS cells exemplified in **a**. Data are mean ± s.d. n = 10 (except 9 for $KS_{F-to-G}$) in all cases. **c**. (top) Quantification of GFP intensity in the bleached condensates. (bottom) Control quantification of mCherry intensity in the bleached condensates. Data are mean ± s.d. *P*-values are from Tukey's post-hoc test after one-way ANOVA. n = 10 (except 9 for $KS_{F-to-G}$) cells in all cases from two biologically independent experiments. **d**. Schematic representation of the NUP98::DDX10 control and mutagenesis constructs. The schematics are in scale to the actual amino acid positions. **e**. Live cell fluorescence microscopy images of U2OS cells expressing the GFP-NUP98::DDX10 constructs each co-transfected with mCherry-CRM1. Scale bar: 5 µm. **f**. FRAP of GFP-NUP98::DDX10 (top), and of mCherry-CRM1 (bottom) in GFP-NUP98::DDX10 condensates in U2OS cells exemplified in **b**. Data are mean ± s.d. n = 10 in all cases. **g**. (top) Quantification of GFP intensity in the bleached condensates. (bottom) Control quantification of mCherry intensity in the bleached condensates. Data are mean ± s.d. *P*-values are from Tukey's post-hoc test after one-way ANOVA. (Top): $P = 0.64$; (bottom): $P = 0.15$. n = 10 cells in all cases from two biologically independent experiments. **h**. Fixed cell imaging of U2OS cells ectopically expressing GFP-NUP98::DDX10, GFP-NUP98::DDX10-KS and GFP-NUP98::DDX10-$KS_{F-to-G}$ and cleavable mCherry reporter with or without treatment with 5% 1,6-hexanediol. **i**. Quantification of effect of 1,6-hexanediol to granularity of GFP signal represented by relative standard deviation* of GFP signal in the nuclei (left). Small regions of saturated GFP signal were allowed - see Methods. Mean intensities of cleavable mCherry reporter (middle) and intensities of mean GFP signal outside the detected NUP98::DDX10 foci (right). *P*-values are from Tukey's post-hoc test after one-way ANOVA. (Left): $P_{(ND_{+HD} vs ND_{-HD})} = {} < 0.001$, $P_{(ND-KS_{-HD} vs ND_{-HD})} = 0.35$, $P_{(ND-KS_{+HD} vs ND-KS_{-HD})} = 0.02$, $P_{(ND-KS\_F-to-G_{-HD} vs ND-KS_{-HD})} = {} < 0.001$, $P_{(ND-KS\_F-to-G_{+HD} vs ND-KS\_F-to-G_{-HD})} = 0.97$; (middle): $P_{(ND_{+HD} vs ND_{-HD})} = 0.04$, $P_{(ND-KS_{-HD} vs ND_{-HD})} = {} < 0.001$, $P_{(ND-KS_{+HD} vs ND-KS_{-HD})} = 0.86$, $P_{(ND-KS\_F-to-G_{-HD} vs ND-KS_{-HD})} = {} < 0.001$, $P_{(ND-KS\_F-to-G_{+HD} vs ND-KS\_F-to-G_{-HD})} = 0.83$; (right): $P_{(ND_{+HD} vs ND_{-HD})} = {} < 0.001$, $P_{(ND-KS_{-HD} vs ND_{-HD})} = {} < 0.001$, $P_{(ND-KS_{+HD} vs ND-KS_{-HD})} = 0.92$, $P_{(ND-KS\_F-to-G_{-HD} vs ND-KS_{-HD})} = {} < 0.001$, $P_{(ND-KS\_F-to-G_{+HD} vs ND-KS\_F-to-G_{-HD})} = 0.23$. Data are mean ± s.d. "n" represents the number of cells from two biologically independent experiments.

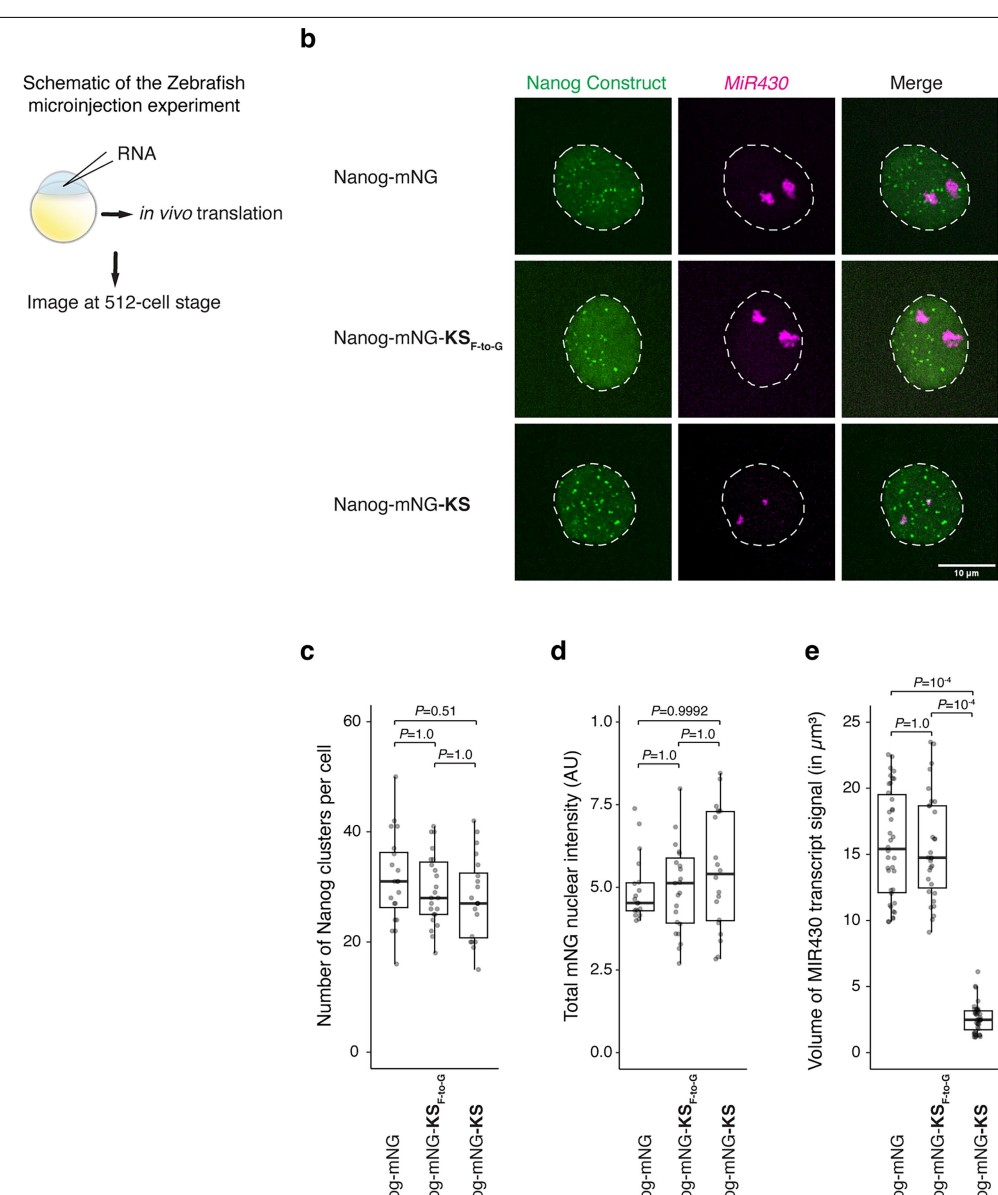

**Extended Data Fig. 11 | Targeting the killswitch to transcription bodies in zebrafish embryos using Nanog results in diminished transcriptional output. a**. Schematic of the embryo injection experiment system. **b**. RNA coding for Nanog-mNG, Nanog-mNG-KS_F-to-G and Nanog-mNG-KS was injected in 1-cell stage wild type embryos. Images were taken at 512-cell stage at the midpoint between two mitoses. Nanog constructs are labelled in green and the *MiR430* transcripts in magenta. Shown are maximum intensity projections (MIPs) of representative images of individual nuclei. mNG: mNeonGreen. **c**. Quantification of the number of total mNeonGreen-Nanog clusters per nucleus in each condition. n = 20 for Nanog-mNG, 20 for Nanog-mNG-KS, and 23 for Nanog-mNG-KS_F-to-G. n represent nuclei from three biologically independent experiments. For **c**, **d**, and **e**, the box and whisker plots are presented with individual data points as dots. The central line represents the median value of

the data. The box represents the middle 50% of the data and the boundaries of the box correspond to 25% and 75% (the first and third quartile). The lines (whiskers) extending from the plot show the range of data. For statistical analysis, we used the Kruskal-Wallis test with no matching or pairing between observations, and corrected for multiple comparisons using Dunn's test. Each p-value is adjusted to account for multiple comparisons. **d**. Quantification of the total Nanog-mNG constructs' nuclear intensity, in arbitrary units. n = 19 for Nanog-mNG, 20 for Nanog-mNG-KS, and 23 for Nanog-mNG-KS_F-to-G. n represent nuclei from three biologically independent experiments. **e**. Quantification of the total volume of the MiR430 transcript signal in transcription bodies in all three conditions. n = 38 for Nanog-mNG, 40 for Nanog-mNG-KS, and 31 for Nanog-mNG-KS_F-to-G. n represent nuclei from three biologically independent experiments.

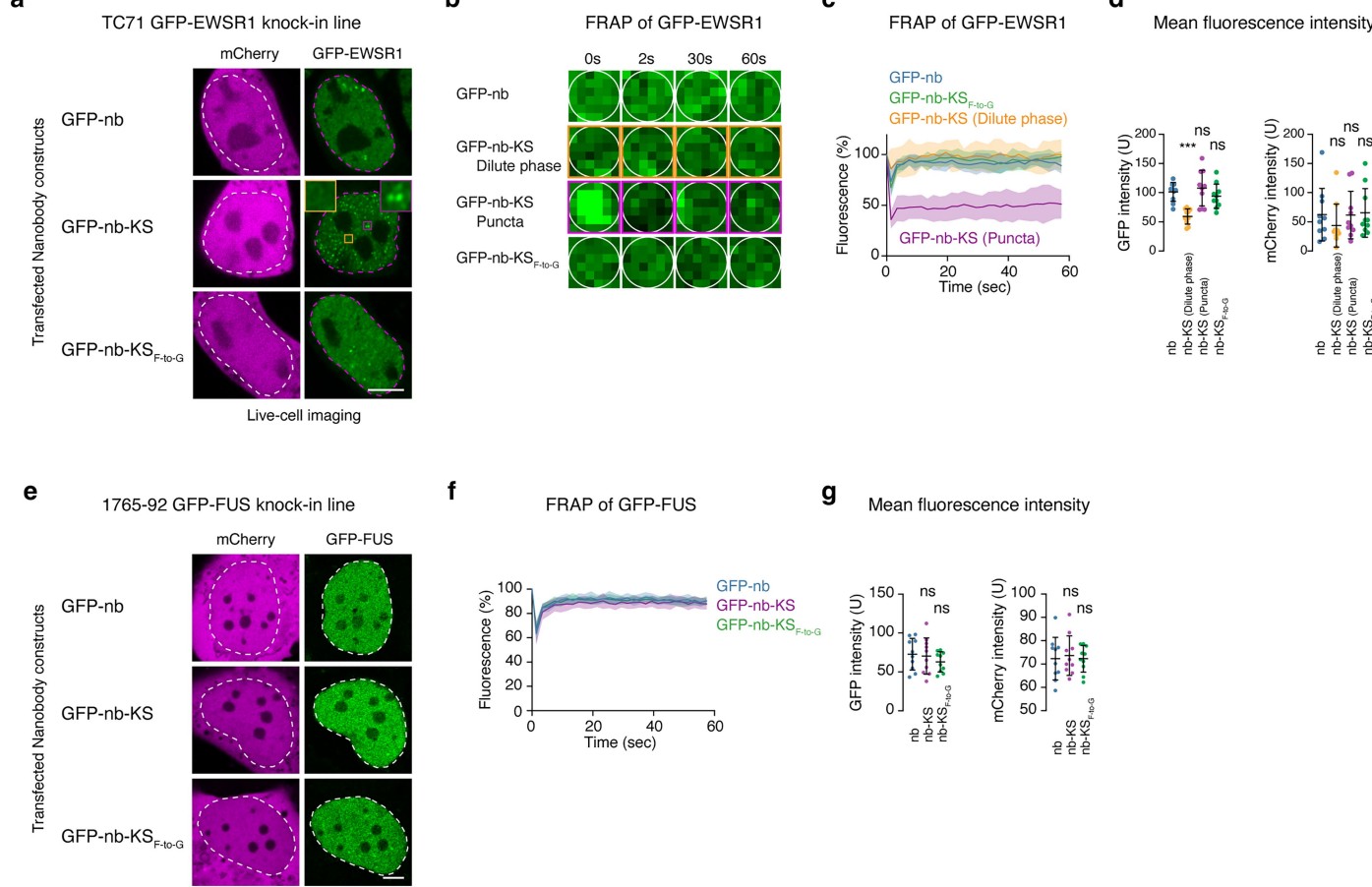

**Extended Data Fig. 12 | The killswitch arrests dynamics of endogenously GFP-tagged condensates, but not the dilute phase GFP signal. a.** Live cell fluorescence microscopy images of TC71 cells expressing GFP-tagged EWSR1 from endogenous locus and ectopic anti-GFP-nanobody and nanobody-KS variants. Orange and magenta boxes show zoom-ins into dilute phase and punctate GFP signal, respectively. The cell nucleus is highlighted with a dashed line contour. Scale bar: 5 μm. **b.** Representative images of the bleached regions at indicated timepoints during FRAP experiment in TC71 GFP-EWSR1 knock-in line. **c.** FRAP experiment for TC71 GFP-EWSR1 knock-in line and ectopic anti-GFP-nanobody and nanobody-KS variants. Data are mean ± s.d. P-values are from Dunnett's multiple comparisons test versus GFP-nb after one-way ANOVA. Scale bar: 5 μm. **d.** Quantification of GFP fluorescence intensity and mean mCherry fluorescence intensity within bleached area measured in **c.** Data are mean ± s.d. P-values are from Dunnett's multiple comparisons test versus GFP-nb after one-way ANOVA. (Left): $P$(nb-KS(Dilute phase)) = 0.0002,

$P$(nb-KS(Puncta)) = 0.84, $P$(nb-KS_F-to-G) = 0.78; (right): $P$(nb-KS(Dilute phase)) = 0.62, $P$(nb-KS(Puncta)) = >0.9999, $P$(nb-KS_F-to-G) = 0.997. n = 10 cells for all samples from two biologically independent experiments. **e.** Live cell fluorescence microscopy images of 1765-92 cells expressing GFP-tagged FUS from endogenous locus and ectopic anti-GFP-nanobody and nanobody-KS variants. The cell nucleus is highlighted with a dashed line contour. Scale bar: 5 μm. **f.** FRAP experiment for 1765-92 GFP-FUS knock-in line and ectopic anti-GFP-nanobody and nanobody-KS variants. Data are mean ± s.d. P-values are from Dunnett's multiple comparisons test versus GFP-nb after one-way ANOVA. Scale bar: 5 μm. **g.** Quantification of GFP fluorescence intensity and mean mCherry fluorescence intensity within bleached area measured in **f.** Data are mean ± s.d. P-values are from Dunnett's multiple comparisons test versus GFP-nb after one-way ANOVA. (Left): $P$(nb-KS) = 0.95, $P$(nb-KS_F-to-G) = 0.43; (right): $P$(nb-KS) = 0.91, $P$(nb-KS_F-to-G) = > 0.9999. n = 10 cells for all samples from two biologically independent experiments.

# Reporting Summary

## Statistics

For all statistical analyses, confirm that the following items are present in the figure legend, table legend, main text, or Methods section.

| n/a | Confirmed | |
|---|---|---|
| ☐ | ☒ | The exact sample size (*n*) for each experimental group/condition, given as a discrete number and unit of measurement |
| ☐ | ☒ | A statement on whether measurements were taken from distinct samples or whether the same sample was measured repeatedly |
| ☐ | ☒ | The statistical test(s) used AND whether they are one- or two-sided *Only common tests should be described solely by name; describe more complex techniques in the Methods section.* |
| ☒ | ☐ | A description of all covariates tested |
| ☐ | ☒ | A description of any assumptions or corrections, such as tests of normality and adjustment for multiple comparisons |
| ☒ | ☐ | A full description of the statistical parameters including central tendency (e.g. means) or other basic estimates (e.g. regression coefficient) AND variation (e.g. standard deviation) or associated estimates of uncertainty (e.g. confidence intervals) |
| ☐ | ☒ | For null hypothesis testing, the test statistic (e.g. *F*, *t*, *r*) with confidence intervals, effect sizes, degrees of freedom and *P* value noted *Give P values as exact values whenever suitable.* |
| ☒ | ☐ | For Bayesian analysis, information on the choice of priors and Markov chain Monte Carlo settings |
| ☒ | ☐ | For hierarchical and complex designs, identification of the appropriate level for tests and full reporting of outcomes |
| ☐ | ☒ | Estimates of effect sizes (e.g. Cohen's *d*, Pearson's *r*), indicating how they were calculated |

*Our web collection on statistics for biologists contains articles on many of the points above.*

## Software and code

Policy information about availability of computer code

| Data collection | Data from fluorescence microscopy was acquired with Zen Black 2.3 (Zeiss). |
|---|---|
| Data analysis | Microscopy data was analyzed using ImageJ 2.14.0/1.54f or ZenBlue 3.2, 3.4, or 3.9 (Zeiss) as indicated in methods. Graph generation and statistical analysis was performed using GraphPad Prism v9 and Rstudio 2024.04.2 with multicomp and ggplot 2 packages. Protein structure predictions were performed using AlphaFold v3. Structural data was visualized using ChimeraX v1.6. Proteomics raw peak data was processed with MaxQuant v2.6.6.0 and output processed with Alphastats v0.6.9. Proteomics correlation calculations were done using SciPy packagev1.10.1 in python v3.10, volcano plots with Alphastats, and plotted with seaborn v0.13.2. RNA-Seq data was trimmed with TrimGalore v0.6.10, mapped using STAR v 2.7.11b and differential gene analysis was done using DEseq2 v1.42.1. Data visualization with ggplot2, distance matrix using dist function, R v4.4. Flow Cytometry data was collected and analysed using BD FACSDiva™ Software vs 8.0.1 or ForeCyte Software. Growth curves were recorded using ForeCyte Software (Essen Bioscience, Ann Arbor, Michigan, USA) Standard Edition 10.0 (R1) Version 10.0.8272, Build Date 8/25/2022.<br><br>All custom code used for data analyses are publicly available at Zenodo under https://doi.org/10.5281/zenodo.15322636. |

For manuscripts utilizing custom algorithms or software that are central to the research but not yet described in published literature, software must be made available to editors and reviewers. We strongly encourage code deposition in a community repository (e.g. GitHub). See the Nature Portfolio guidelines for submitting code & software for further information.

## Data

Policy information about availability of data

All manuscripts must include a data availability statement. This statement should provide the following information, where applicable:

- Accession codes, unique identifiers, or web links for publicly available datasets
- A description of any restrictions on data availability
- For clinical datasets or third party data, please ensure that the statement adheres to our policy

Sequencing data were deposited at the Gene Expression Omnibus, under the accession ID: GSE284494. Mass spectrometry data were deposited to the ProteomeXchange Consortium (http://proteomecentral.proteomexchange.org) via the PRIDE partner repository, with the dataset identifier PXD058854. The NGS experiments of human samples used the human genome hg38 and annotation from GENCODE GRCh38.p13. Plasmids were deposited at Addgene (237619-237693 and 238231-238298). All raw and processed data were deposited at Zenodo, and are publicly available under https://doi.org/10.5281/zenodo.15322636. The source data behind all graphs in figures are provided with this paper as a Source Data workbook.

## Research involving human participants, their data, or biological material

Policy information about studies with human participants or human data. See also policy information about sex, gender (identity/presentation), and sexual orientation and race, ethnicity and racism.

| | |
|---|---|
| Reporting on sex and gender | No data on human participants is included. |
| Reporting on race, ethnicity, or other socially relevant groupings | No data on human participants is included. |
| Population characteristics | No data on human participants is included. |
| Recruitment | No data on human participants is included. |
| Ethics oversight | No data on human participants is included. |

Note that full information on the approval of the study protocol must also be provided in the manuscript.

# Field-specific reporting

Please select the one below that is the best fit for your research. If you are not sure, read the appropriate sections before making your selection.

☒ Life sciences      ☐ Behavioural & social sciences      ☐ Ecological, evolutionary & environmental sciences

For a reference copy of the document with all sections, see nature.com/documents/nr-reporting-summary-flat.pdf

# Life sciences study design

All studies must disclose on these points even when the disclosure is negative.

| | |
|---|---|
| Sample size | No statistical methods were used to predetermine sample sizes. Sample sizes for RNA, protein expression and imaging experiments are consistent with current standards and are indicated in the figures, legends or methods section. Examples of imaging & FRAP: PMID: 37683610; RNA-seq: PMID: 39373104, PMID: 39448850, proteomics: PMID: 38355802. |
| Data exclusions | Thresholds of mCherry expression were used to exclude non-transfected cells as described in the methods. We used cleavable mCherry as a reporter to focus analysis on cells that had been transfected, and omitted untransfected cells based on mCherry expression level. Thresholds of mCherry expression were not pre-determined, but determined empirically based on each experiment and are described in the methods section. Although threshold vary between different experimental and image acquisition conditions, thresholds were kept similar across samples that were compared. In addition, for BRD4::NUT hexanediol experiment, nuclei with saturated pixels on GFP channel were excluded to have reliable measurements of GFP signal variability. |
| Replication | All experiments were repeated at least two times with similar results unless stated otherwise in the figure legends or "Statistics and Reproducibility" in the methods section. |
| Randomization | Not relevant for this study. Samples were allocated into wt, KS and KS variant groups without randomization, but the order of samples on plates or in which they are imaged would not affect the results. |
| Blinding | Blinding was not relevant for these experiments and in some cases even impossible since solidifying effect of the KS would become apparent during imaging. However, data was collected and analyzed in similar manner regardless of sample groups, allowing unbiased analysis of data. |

# Reporting for specific materials, systems and methods

We require information from authors about some types of materials, experimental systems and methods used in many studies. Here, indicate whether each material, system or method listed is relevant to your study. If you are not sure if a list item applies to your research, read the appropriate section before selecting a response.

## Materials & experimental systems

| n/a | Involved in the study |
|-----|----------------------|
| ☐ | ☒ Antibodies |
| ☐ | ☒ Eukaryotic cell lines |
| ☒ | ☐ Palaeontology and archaeology |
| ☐ | ☒ Animals and other organisms |
| ☒ | ☐ Clinical data |
| ☒ | ☐ Dual use research of concern |
| ☒ | ☐ Plants |

## Methods

| n/a | Involved in the study |
|-----|----------------------|
| ☒ | ☐ ChIP-seq |
| ☐ | ☒ Flow cytometry |
| ☒ | ☐ MRI-based neuroimaging |

## Antibodies

**Antibodies used**

Alexa Fluor 647 donkey anti-mouse, Jackson Immuno Research, 715-605-150, 1:1000.
Alexa Fluor 647 anti-rabbit, Jackson Immuno Research, and 711-605-152, 1:1000.
5.8S rRNA (Novus, NB100-662SS, 1:500)
RNAPII (Abcam, ab26721, 1:500)
H3K27Ac (Abcam, ab4729, 1:1000)
Alexa Fluor goat anti-rabbit 488 antibody (Life Technologies, Cat#: A-11008), 1:1000.
goat anti-mouse 488 antibody (Life Technologies, Cat#: A-11001), 1:1000.
Antibody to 52K (gift from P. Hearing, Stony Brook University, NY; PMID: 15709002), species: rabbit, polyclonal, 1:500.
IIIA (gift from P. Hearing), species: rabbit, polyclonal, WB 1:10,000
DBP (gift from A. Levine; PMID: 6310869), species: mouse, clone: B6-8, 1:400.
HA-Tag (C29F4) Rabbit mAb #3724, Cell Signaling, 1:1000.
Hexon, Penton, Fiber (Abcam Cat#: ab6982), species: rabbit, polyclonal, WB 1:10,000.
GAPDH (GeneTex, Cat#: GTX100118, Lot: 43712), species: rabbit, polyclonal, WB 1:5,000
HRP-conjugated goat anti-rabbit (Jackson Laboratories, Cat#: 111-035-045), 1:10,000.
TCOF1 (Santa Cruz, sc-374536, 1:750),
GFP (Invitrogen, A11122, 1:2000),
NPM1 (Invitrogen, 32-5200, 1:2000),
HP1α (CST, #2616, 1:1000),
Histone H3 (Abcam, ab1719, 1:10000),
GAPDH (CST, #14C10, 1:4000),
HSP90 (BD, 610419, 1:2000),
anti-mouse Gr-1/Ly-6C BV421 (clone RB6-8C5, Biolegend), 1:200,
anti-mouse CD117/c-Kit APC (clone 2B8, Biolegend), 1:200.
NEPRO (Santa cruz, sc-376579) 1:100

**Validation**

Antibodies were not validated in-house, but all antibodies have been cited in numerous publications.

5.8S rRNA (Novus, NB100-662SS)
https://www.novusbio.com/PDFs/NB100-662.pdf

RNAPII (Abcam, ab26721)
https://www.abcam.com/en-us/products/primary-antibodies/rna-polymerase-ii-ctd-repeat-ysptsps-antibody-chip-grade-ab26721?srsltid=AfmBOooZuSRt47SrB9UwBQaor2WJ_evhZYeRoKK76hbWpNqppNHsoAlb

H3K27Ac (Abcam, ab4729)
https://www.abcam.com/en-us/products/primary-antibodies/histone-h3-acetyl-k27-antibody-chip-grade-ab4729?srsltid=AfmBOorW3BjfDiqA1eFCeG6ejgg5O1n-ZQZYPUXfT4LtLa_b0djgpVi8

HA-Tag (C29F4) (#3724, Cell Signaling)
https://www.cellsignal.com/products/3724/datasheet?images=1&protocol=0&size=A4

GAPDH (GeneTex, Cat#: GTX100118)
https://www.genetex.com/PDF/Download?catno=GTX100118&srsltid=AfmBOopuq1lsZv4hVsKqaGuu4U43wwnYWWmOX0Dt4_ktlK83gTnFnBwF

TCOF1 (Santa Cruz, sc-374536)

https://datasheets.scbt.com/sc-374536.pdf

GFP (Invitrogen, A11122)
https://www.thermofisher.com/order/genome-database/dataSheetPdf?
producttype=antibody&productsubtype=antibody_primary&productId=A-11122&version=Local

NPM1 (Invitrogen, 32-5200)
https://www.thermofisher.com/order/genome-database/dataSheetPdf?
producttype=antibody&productsubtype=antibody_primary&productId=32-5200&version=Local

HP1α (CST, #2616)
https://www.cellsignal.com/products/2616/datasheet?images=1&protocol=0&size=A4

Histone H3 (Abcam, ab1719)
https://www.abcam.com/en-us/products/primary-antibodies/histone-h3-antibody-nuclear-marker-and-chip-grade-ab1791?
srsltid=AfmBOop1I1ZVZPzH92dKeqZrPNGqTvF7LHjTFbUSfpDHbD-0iyCHFVgC

GAPDH (CST, #14C10)
https://www.cellsignal.com/products/2118/datasheet?images=1&protocol=0&size=A4

HSP90 (BD, 610419)
https://www.bdbiosciences.com/content/dam/bdb/products/global/reagents/microscopy-imaging-reagents/immunofluorescence-
reagents/610xxx/6104xx/610419_base/pdf/610419.pdf

anti-mouse Gr-1/Ly-6C BV421 (clone RB6-8C5, Biolegend)
https://d1spbj2x7qk4bg.cloudfront.net/Default.aspx?
ID=13406&pdf=true&displayInline=true&ProductID=460&leftRightMargin=15&topBottomMargin=15&filename=PE%20anti-mouse%
20Ly-6GLy-6C%20(Gr-1)%20Antibody.pdf&v=20250407123848

anti-mouse CD117/c-Kit APC (clone 2B8, Biolegend)
https://d1spbj2x7qk4bg.cloudfront.net/Default.aspx?
ID=9851&pdf=true&displayInline=true&ProductID=77&leftRightMargin=15&topBottomMargin=15&filename=Purified%20anti-
mouse%20CD117%20(c-Kit)%20Antibody.pdf&v=20250407123848

NEPRO (Santa cruz, sc-376579)
https://datasheets.scbt.com/sc-376579.pdf

Adenovirus late protein antibody (gift from J. Wilson): Recognizes Hexon (band of approx. 110 kDa), Penton (band of approx. 65 kDa),
and Fiber (band of approx. 61 kDa) in adenovirus infected whole cell lysates but not uninfected lysates (Kozarsky et al. 1996. DOI:
10.1038/ng0596-54; Herrmann et al. 2020. DOI: 10.1038/s41564-020-0750-9; Charman et al. 2023. DOI: 10.1038/
s41586-023-05887-y).

52K (gift from P. Hearing): Recognizes 52K (band of approx. 52kDa) on western blots of adenovirus infected whole cell lysates, but
not uninfected whole cell lysates or lystaes from cells infected with a Δ52K mutant adenovirus. Immunofluorescence staining shows
signal in adenovirus infected cells but not uninfected cells. (Ostapchuk et al. 2005. DOI: 10.1128/JVI.79.5.2831-2838.2005; Charman
et al. 2023. DOI: 10.1038/s41586-023-05887-y).

IIIa (gift from P. Hearing): Recognizes IIIa (band of approx. 65 kDa) on western blots of adenovirus infected whole cell lysates or
adenovirus particles, but not uninfected whole cell lysates (Ma & Hearing. 2011. DOI: 10.1128/JVI.00467-11; Charman et al. 2023.
DOI: 10.1038/s41586-023-05887-y).

DBP (gift from A. Levine): Recognizes DBP (band of approx. 72 kDa) on western blots of adenovirus infected whole cell lysates, but
not uninfected whole cell lysates. Immunofluorescence staining shows signal in adenovirus infected cells but not uninfected cells.
Immunofluorescence staining shows localization to viral replication compartments in H5ts107-infected cells grown at the permissive
temperature of 32°C but not in H5ts107-infected cells grown at the non-permissive temperature of 39.5° C (Reich et al 1983. DOI:
10.1016/0042-6822(83)90274-x; Charman et al. 2023. DOI: 10.1038/s41586-023-05887-y ).

# Eukaryotic cell lines

Policy information about cell lines and Sex and Gender in Research

| Cell line source(s) | U2-OS (ATCC #HTB-96),<br>HEK293T (ATCC #CRL-3216),<br>Lenti-X 293T (Takara Bio, Cat. No. 632180),<br>HCT-116 (ATCC #CCL-247),<br>HCT-116 GFP-NPM1 (this study),<br>U2OS GFP-NPM1 (this study),<br>HCT-116 GFP-TCOF1 (this study),<br>HCT-116 SRRM2-GFP (this study),<br>HAP1 SRRM2-GFP tr0 (Source: Tugce Aktas lab, PMID: 33095160),<br>A673 (Source: Dr. Heinrich Kovar. Originally from CLS #300454),<br>A673 Dox-inducible EGFP-NPM1 (this study),<br>MCF7 (ATCC # HTB-22),<br>C2C12 (ATCC # CRL-1772), |
| --- | --- |

V6.5 mESC (Source: Alexander Meissner lab),
V6.5 mESC GFP-HP1a (this study),
Murine fetal liver cells (Source: Florian Grebien lab),
H3122 (CLS, #300484),
TC71 (DMSZ, #ACC516),
TC71 GFP-EWSR1 (this study),
1765-92 (source: Dr. Pierre Åman),
1765-92 GFP-FUS (this study).
For experiments involving expression of 52K and KS variants: HEK-293 (ATCC #CRL-1573), HEK-293T (ATCC #CRL-3216).

| | |
|---|---|
| Authentication | Cell line identity were verified using morphological characteristics, but lines have not been authenticated. |
| Mycoplasma contamination | All cell lines tested negative for mycoplasma using LookOut Mycoplasma PCR Detection Kit (Sigma-Aldrich, MP0035) or PCR Mycoplasma Test Kit II (Applichem, A8994). Mycoplasma testing was carried out on 0.2–1 mL of cell culture media taken from tissue culture dishes containing confluent monolayers of cells on a routine basis at least twice a year. |
| Commonly misidentified lines (See ICLAC register) | No commonly misidentified cell lines were used in the study. |

## Animals and other research organisms

Policy information about studies involving animals; ARRIVE guidelines recommended for reporting animal research, and Sex and Gender in Research

| | |
|---|---|
| Laboratory animals | For establishment of murine AML cell lines, fetal liver cells were transduced with oncogene-expressing plasmids and transplanted into recipient mice. For this, male and female C57BL/6J.SJL mice at the age of 10-12 weeks were used. Mice were kept in specific opportunistic pathogen free quality (SOPF) under stringent controlled standard conditions, in individually ventilated cages, fed with Sniff Haltungsfutter CHOW standard 10mm pellets (Catalog-No. V1534-000), ad libitum.<br>For the Zebrafish experiments: the study used embryos of Wild Type (ABTL) zebrafish. No fish older than 5 days were used, which means that according to the regulations on animal experimentation, we did not perform animal experiments. Adult fish were maintained according to local husbandry regulation, which equally does not constitute animal experimentation. To obtain embryos, male and female fish were placed in a water tank with a separating net in the afternoon; the fish were placed together the following morning and embryos could be collected after few minutes of spontaneous mating. |
| Wild animals | No wild animals were used in the study. |
| Reporting on sex | Male and female recepient mice were used in this study for transplantation and establishment of murine AML models. This study does not include any experiments in which animals were subjected to different treatment cohorts, for which sex-based analysis would be relevant. |
| Field-collected samples | No field collected samples were used in the study. |
| Ethics oversight | All animal studies were performed according to ethical animal license protocols and were approved by the responsible authorities of the Austrian government (BMBWF-68.205/0199-V/3b/2018). |

Note that full information on the approval of the study protocol must also be provided in the manuscript.

## Plants

| | |
|---|---|
| Seed stocks | Not relevant for this study. |
| Novel plant genotypes | Not relevant for this study. |
| Authentication | Not relevant for this study. |

# Flow Cytometry

## Plots

Confirm that:

☒ The axis labels state the marker and fluorochrome used (e.g. CD4-FITC).

☒ The axis scales are clearly visible. Include numbers along axes only for bottom left plot of group (a 'group' is an analysis of identical markers).

☐ All plots are contour plots with outliers or pseudocolor plots.

☒ A numerical value for number of cells or percentage (with statistics) is provided.

## Methodology

| | |
|---|---|
| Sample preparation | For NUP98::KDM5A experiments: cells were washed with PBS and resuspended in PBS with 0.5% FCS, followed by staining for 30 min with dilutions (1:200) of the following antibodies (all from Biolegend, San Diego, CA, USA): anti-mouse Gr-1/Ly-6C BV421 (clone RB6-8C5) and anti-mouse CD117/c-Kit APC (clone 2B8).<br>For NuFANCI experiments: described comprehensively in NuFANCI method section. |
| Instrument | For NUP98::KDM5A: BD FACSCanto II; For NuFANCI: BD FACSAria™ Fusion |
| Software | FlowJo (FlowJo LLC, Ashland, OR, USA). |
| Cell population abundance | The final sorted populations of "Nucleolus" were 13.7 - 27.4% of the total events. Representative samples of n=4 independent experiments of each condition are shown in Supplementary Figure 4e. |
| Gating strategy | For NuFANCI experiments, sorting of "Nucleolus" three gates were used: 1) DAPI (uv-450/50-A) vs GFP (b-530/30-A) was used to identify the population containing "Nucleolus" (GFP+, DAPI-intermediate), determined by sorting different fractions outlined in Supplementary Figure 1c and subsequent imaging (Supplementary Figure 1d); 2) FSC-A vs SSC-A gate was used to exclude large events; 3) GFP (b-530/30) vs mCherry (yg-610/20) was used to sort for either mCherry– (for the samples untransfected and Actinomycin D) or mCherry+ (for the samples Nb, KS, KSFtoG and 2xKS). Gates were determined by comparing to mCherry– samples (untransfected). For NUP98::KDM5A experiments, Live cells were discriminated based on forward scatter height (FSC-H) and side scatter height (SSC-H). Single cells were gated based on forward scatter height (FSC-H) and forward scatter area (FSC-A). mCherry-positive cells were identified by their signal intensity in the ECD channel. Cellular staining for anti-mouse Gr-1/Ly-6C was assessed based on the signal intensity in the Pacific Blue channel (BV421), while anti-mouse CD117/c-Kit staining was evaluated based on signal intensity in the APC channel. |

☒ Tick this box to confirm that a figure exemplifying the gating strategy is provided in the Supplementary Information.

