## [Peer Review File · Nature]

Probing condensate microenvironments with a micropeptide killswitch

Corresponding Author: Dr Denes Hnisz

Version 0:

Reviewer comments:

Referee #1

(Remarks to the Author)

In this paper, Zhang et al., co-opt a sequence from a pathogenic HMGB-fusion protein containing a C-terminal frameshift to create a new tool (the "Killswitch") that binds and reduces the dynamics of nucleolar proteins. They show that this 17 a.a. peptide self-assembles and disruption of self-assembly reduced the efficacy of the Killswitch (F-A mutant). Thus, the Killswitch modifies condensate properties (solidifies them) by adding valence. By coupling the Killswitch to a GFP nanobody, they demonstrate that the Killswitch can alter the dynamic properties and activities of a wide range of naturally occurring and pathogenic condensates, however with the condition that these condensates contain a GFP-fusion protein. Throughout the paper, the authors include the dead Killswitch (F-A mutant) as a vital control to show that the effect is due to Killswitch self-interactions, not necessarily through heterotypic interactions between the Killswitch and proteins within the target condensate.

In my opinion, the results of this study are convincing and interesting. The exact mechanism of how the Killswitch works may not be fully understood, and it may well be context dependent, but that doesn't dim my enthusiasm. Reducing condensate dynamics through added valence is not a novel idea, but the Killswitch is a versatile tool, and many condensate researchers will be interested in using it. The main downside of this paper is that the Killswitch is currently limited to GFP-tagged proteins, so we don't know if it can be used as a therapeutic for diseases involving the formation of pathological condensates. I discuss these points in more detail below.

Major points:

Interpretation of the results may be difficult due to complexity of altering material properties. For example, the Killswitch may alter other features of condensates, such as growth, fusion, or susceptibility to proteasome-mediated degradation. To me, that seems irrelevant at this point. The Killswitch's effect is likely context-dependent, and it will be up to future studies to determine how the Killswitch specifically affects each condensate. Thus, it is hard to say that the effect is strictly due to a change in material properties. Again, the same thing could be said for any perturbation that solidifies the condensate. Adding some caveats on this point in the discussion would be useful.

One must consider this paper in the context of other studies inventing clever ways of modifying condensate properties. For example, Cliff Brangwynne's lab already devised a way to solidify condensates with the Opto-droplet tagging system. Deep quenching, or long exposures to blue light, drives condensates into an arrested state (PMC5562165). Other studies have used Cry2o or FKBP-FRB (PMC9669173) to add valence. Disruption of condensate properties with small molecules was done in Hernandez-Candia 2021. Gui et al (2023 Mol Cell) developed short peptides to perturb condensate formation and activity (YAP, TAZ, Hippo pathway). Thus, the killswitch is another way to perturb condensates by, presumably, making their connections stronger and decreasing dynamics. However, each of the perturbations in these previous studies are specific to those engineered systems; in other words, they do not present a broadly applicable technology. By adding the Killswitch to the GFP nanobody, this study makes a user-friendly tool to study all condensates containing GFP-tagged proteins. Thus, while the technological and conceptual leap is not that advanced, the product will be useful to the broader condensate community.

The paper makes several points about how we need tools to disrupt endogenous condensates, which would have profound implications as a therapeutic:

"A few studies have reported protein-tagging approaches to dissolve ectopic condensates 22,31,32, but none were tested on

endogenous condensates, that are known to significantly differ in their biophysical and functional properties from condensates formed by ectopically overexpressed proteins 5,33.”

“Condensate alterations have been linked to hundreds of human diseases, notably, cancers driven by fusion oncoproteins, and rare genetic diseases, and these conditions typically have no therapeutic remedy 34-36. We therefore tested the effect of recruiting the killswitch to multiple disease-specific condensates.”

Herein lies the main weakness of the study. Their current Killswitch can only recognize GFP-tagged proteins. Thus, it cannot be used on endogenous condensates in unmodified cells or as a therapeutic. If the authors could design a Killswitch that targets a naturally occurring, unlabeled protein, that would be a tremendous breakthrough. For example, make a nanobody against the Nup98::KDM5A fusion protein, then label that nanobody with the Killswitch peptide. Does it still stop proliferation of the murine AML cells? If so, they would have generated a drug to target the endogenous pathogenic protein. This experiment would also further demonstrate the applicability of this technique to non-GFP-tagged proteins. Considering the time and potential difficulty of generating such a nanobody, I leave this suggestion to the discretion of the editor.

Referee #2

(Remarks to the Author)

The work by Zhan et al addresses an important challenge in understanding the role of biomolecular condensates in cells. Despite their ubiquitous distribution, dissection of their roles is often very tricky since mutations of condensate-forming proteins are often associated with other functions of these proteins that are difficult to decouple of their propensity to undergo phase transitions. The development of a "killswitch" peptide to probe condensate microenvironments represents a significant advancement for the field. The authors employ a clever strategy by using a non-natural micropeptide combined with a nanobody-based recruitment system. This allows for precise manipulation and observation of condensates within living cells, and the choice of targets, such as nucleolar proteins and oncoprotein condensates, is well-justified and demonstrates the versatility of the approach. Together, these findings provide a new tool for studying the functional dynamics of cell condensates. The manuscript is well-organized and clearly written, and the findings are clearly presented, highlighting the power of their approach in revealing condensate's functions. One of the weaknesses is the lack of the molecular mechanism of the killswitch and how it specifically interacts with target proteins in condensates. Another weakness is that the authors only demonstrate the applicability of the KS with a nanobody targeting GFP. It would be useful to illustrate how other nanobodies targeting scaffold or client proteins directly can also recruit KS in natural systems without GFP, e.g., a killswitch fused to peptides/proteins that naturally partition to condensates rather than fused to nanobodies

Major considerations:

1. The authors refer to the material properties throughout the text, however, they only rely on FRAP data – I would rephrase this as without actual material property measurement techniques (and FRAP isn't one of the methods) one cannot comment on whether these condensates are solid/gel-like or liquid-like.

Further comments:

107: Partitioning into nucleolus is reported, but it is not clear which regions have affected mobility. FC, DFC, GC specifically? If all, how/why?

111: The removal of Phe residues from KS inactivates it – can other aromatic residues restore its activity? Can the authors comment on whether the removal of the aromatic residues affects the (predicted) helical conformation of KS?

175: There's no justification of why 2 killswitch-Nanobodies are necessary. Also, an earlier justification of Nanobodies over other delivery systems is missing. An additional/orthogonal delivery approach would strengthen this manuscript significantly. Does the location of the KS peptide (C- or N-terminal) affect its ability to interfere with condensates?

Can the authors discuss the underlying reasons why some condensates were less affected by the killswitch or needed double KS?

181: Why is the formation of spherical structures used as evidence of hardening? FRAP data in 2E shows little reduction for a single killswitch. Can the authors explain this?

188: GC is the assembly site of early/pre-ribosomes. rRNA processing occurs primarily in the dense fibrillar component (DFC) of the nucleolus.

195, 205: The authors report that only certain proteins' mobility is affected by killswitch rather than condensate-wide effects. If other proteins and RNA behave normally, the impact of the killswitch and usefulness as a tool in question. The authors don't address these concerns, instead using findings to argue against previous findings.

253: why does killswitch affect the partitioning of some constituents but not others?

A discussion on possible causes for why killswitch affects some but not all components of condensate, including whether this is a nanobody-induced effect is required.

Fig 1E: The presentation of FRAP data doesn't show the recovery rate of mobile fraction, e.g. if slow but still mobile.

Fig 2D: GFP HP1a shows very high background, aberrant effects of killswitch from nonspecific targeting. Also, the ordering of nanobody imaging conditions is mixed up between the first & last 2 conditions, which is confusing.

Fig 4C: A control with a marker localizing specifically to target condensates should be included. There appears to be a high background of killswitch in the cytosol, possibly confounding effects. The Phe to Ala control wouldn't aggregate/form condensates alone/with other molecules in the cytosol, therefore it isn't necessarily an adequate control/comparison when investigating effects beyond condensate level, e.g. cell viability.

Referee #3

(Remarks to the Author)

Zhang et al. report on repurposing a short peptide sequence, found as a naturally occurring pathogenic frameshift variant of HMGB1, as a modulator of condensate function. The authors show some interesting observations in vitro and cells. The peptide appears to self-associate in some way, and thought this self-association, perturbs condensate function. These are interesting observations, but the paper is slightly let down by a lack of characterisation/understanding of the mechanism of action. The effects are also slightly hard to understand/rationalise, for instance in the case of NPM1. It is therefore not clear to this reviewer whether the findings will well enough defined to be of value as a condensate modulation approach.

Referee #4

(Remarks to the Author)

Yaotian Zhang and colleagues report a genetically encodable peptide “killswitch” for probing how the material properties of biomolecular condensates contribute to cellular functions. The work originates from the authors’ previous observation of a frameshift mutation in HMGB1 that leads to nucleolar partitioning and disruption of nucleolar dynamics. The authors identify a 17-aa C-terminal peptide from the mutant HMGB1 protein as being responsible for altering the liquid-like behavior of the nucleolus. After some further characterization of the molecular mechanism, including mutational analyses and structural modeling, the authors set out to apply the killswitch peptide as a molecular tool to investigate the link between the material properties of various condensates and their function in cells. To do so, the authors append the killswitch peptide to an anti-GFP nanobody to enable recruitment to GFP-tagged proteins known to form nuclear condensates, including both endogenously labeled native proteins and overexpressed oncogenic fusions.

The need for tools to investigate endogenous biomolecular condensates in cells cannot be understated. However, it may be difficult to use the killswitch. The peptide gets incorporated into condensates and can lead to more gel-like properties. But different condensates have different characteristic dynamics and vary in their native “liquid” or “gel-like” behaviors – the material properties are different for different systems and potentially for individual condensates with the cell. Thus, the killswitch may not be effective in condensates that are naturally more gel-like. As shown by the authors’ data, incorporation of the killswitch can have different effects, including minimal changes, partial changes in the makeup of the condensates, or triggering the activation of degradation machinery. Whether these effects can all be linked directly to changes in the material properties of the corresponding condensates remains an open question. The FRAP analyses performed in the study provide an incomplete picture. Both the native material properties of the condensates and the killswitch-induced changes in these properties need to be more systematically and quantitatively characterized. As it currently stands, the data obtained using the killswitch cannot be reliably interpreted, casting doubt on the potential for broader application of this tool.

Version 1:

Reviewer comments:

Referee #1

(Remarks to the Author)

The reviewers have addressed my previous major concerns. I think the manuscript is suitable for publication provided that they address a few minor points detailed below.

Overview:

The mechanism of killswitch activity is well documented and convincing. The authors demonstrate that homotypic interactions driven largely by three phenylalanines underlie the changes in condensate dynamics. An F-to-G mutant kills this activity, and both size exclusion chromatography and AF3 modeling indicate that this is because these mutations prevent multimerization of the killswitch. Thus, I agree with the authors that the Killswitch most likely works through added valence, which should increase the connectivity of the tagged protein (via homotypic and not heterotypic interactions) and therefore decrease the dynamics of the tagged molecule.

The new experiments in Figure 3 showing that the Killswitch can change the composition of condensates is astounding. The authors thus strengthen their story by adding another layer to the mechanism by which the Killswitch affects condensate function.

My main concern of the previous version was that the utility of the Killswitch was limited by its fusion to a GFP-nanobody. Thus, only non-native, GFP-labeled proteins could be affected. The authors present many disease-relevant models where perturbing condensate function could be a cure. They speculated that the Killswitch could be a therapeutic, but this would depend upon developing a Killswitch that recognizes endogenous proteins.

In their revised version, the authors present several additional methods of targeting the Killswitch to condensates. They fuse the Killswitch to three additional nanobodies that recognize widely used epitopes (e.g., HA tag, ALFA tag, VH005 tag). They also fuse the Killswitch to an R10 peptide that is membrane-permeable and targets the nucleolus. In extended Fig. 23 they demonstrate the Killswitch can be introduced through injecting mRNA encoding a fusion protein (Nanog-KS). The authors claim they have yet to make a Killswitch that targets endogenous, disease-causing condensates. This could be due to the difficulty in making a highly-specific nanobody.

While the authors could not completely address my main concern, the additional methods of introducing the Killswitch strengthen the paper and broaden the applicability of this system. I, for one, would love to use this approach, and I am waiting for this paper to be published so I can do so. I'm sure future studies will identify the tools to turn the Killswitch into a potential therapeutic.

Minor Comments:

1. The authors frequently jump back and forth between "Dynamics" and "Material Properties", and so, they are implicitly stating that one equals the other. There are no standard measurements of material properties in this paper (e.g., rheology to determine viscoelastic moduli, surface tension, creep, etc.). Rather, there is much inferred from FRAP and shape characteristics (e.g., whether nucleoli look more or less circular). The authors should tone down their statements about the Killswitch affecting material properties.

2. Susceptibility to hexanediol is not informative about material properties. We have no sense whether viscoelastic moduli change in response to hexanediol for the condensates tested. The authors need to restate this conclusion to be more precise in their language. I am referring to this statement:

"Treatment of the cells with 1-6 hexanediol dissolved BRD4::NUT but not BRD4::NUT-KS condensates, further confirming that material properties of the condensates were affected by the killswitch."

3. Extended Data Fig. 23 seems out of place. It is mentioned in passing at the beginning of the discussion without proper justification. Why was this experiment done? How was it done? Does it belong here? Is the point that the Killswitch can be used in another species, or that the Killswitch can be used to affect function of another condensate with an easily detected readout? The authors should consider explaining this experiment, the rationale, and results better. Otherwise, leave it out. I personally, think it's too much.

Referee #2

(Remarks to the Author)

I am satisfied with the revised version of the manuscript, and the authors have done their best to address the points I and some of my colleagues also raised while reading the original version of the manuscript.

Referee #4

(Remarks to the Author)

The revised manuscript from Zhang and colleagues incorporates substantial new data from multiple experiments, yet key deficiencies remain unresolved. In particular, the killswitch is presented as "a widely applicable tool to alter the material properties of endogenous condensates". However, concerns regarding the impact of the killswitch peptide on condensate material properties are not sufficiently addressed. The FRAP analyses and hexanediol experiment are inadequate to properly assess condensate material properties. Both the native material properties of the targeted condensates and the killswitch-induced changes in these properties need to be systematically and quantitatively characterized, for example by performing in vitro measurement using purified or reconstituted condensates. Meanwhile, the newly added FACS studies seem to only complicate things by revealing that killswitch treatment alters the molecular composition of targeted condensates, which suggests an entirely separate mechanism of action. As written by the author – "the killswitch altered condensate composition, component dynamics and function, and even had an effect on the transcriptional output of transcription bodies in Zebrafish embryos. The changes appeared selective to certain condensate components, suggesting that various proteins likely partition into condensates through different underlying molecular features. The killswitch had more dramatic effects on some condensates (e.g., NUP98-fusion condensates) and marginal effect on others (e.g. chromocenters)". Are these different effects linked to changes in the material properties of the corresponding condensates? Without a sufficient understanding of the mechanism, how would users of this tool try to interpret the data obtained? These concerns cast doubt on the potential for broader application of this tool.

Version 2:

Reviewer comments:

Referee #1

(Remarks to the Author)

The authors have addressed all my concerns. In my opinion, the paper is ready for publication.

Referee #4

(Remarks to the Author)

The added in vitro data has appropriately addressed the concerns about the actual material properties being affected by the killswitch. I am now satisfied with the revised manuscript.

General Response to Reviewers

We thank the reviewers for the useful comments which guided us to a substantially improved manuscript. Our study reports the development of a universally applicable micropeptide tool (i.e., the “killswitch”) to control the material properties of endogenous and disease-specific condensates in human cells. Using this tool, we discovered changes in composition, component-dynamics and functions of nucleoli, oncoprotein condensates and viral condensates upon targeting them with the killswitch. The results provide direct evidence for specialized microenvironments in condensates that enable non-stoichiometric control of cellular activity.

The reviewers noted that “*The need for tools to investigate endogenous biomolecular condensates in cells cannot be understated*”, and recognized that the results of our study are “*convincing and interesting*”, and “*represent a significant advancement*”. The reviewers also noted important concerns, which we believe we have addressed in the revision. The two major areas of concern are summarized as follows:

1) Alternative targeting approaches

In the original manuscript we recruited the killswitch with an anti-GFP-nanobody to GFP-labelled condensates. In the revised manuscript we include several additional methods to recruit the killswitch to condensates, all of which led to measurable cellular effects:

- i) we include data on recruiting the killswitch with three additional nanobodies;
- ii) we conjugated the killswitch to a short cell-permeable peptide and show that it works;
- iii) we targeted the killswitch using a trans-recruitment system in embryos.

Two out of the three new approaches target unlabeled endogenous condensate-forming proteins that do not have a protein tag, suggesting a more universal applicability of the killswitch beyond what was implied in the first submission.

2) More insights into the mechanism of the killswitch on condensates

In the original manuscript we described various, but seemingly disparate effects when targeting four different condensates, and we used only the killswitch to alter the material properties of condensates. We now include multiple new experiments that provide mechanistic insights into the effect of the killswitch, reveal consistency in the findings, and further substantiate that the killswitch alters material properties of condensates:

- i) we provide new data showing that for four different condensates tested, the killswitch affects composition, component dynamics and function;
- ii) we developed a new FACS-based method to sort condensates from cells, and describe the global compositional changes in nucleoli upon targeting with the killswitch;
- iii) we demonstrate that effects of the killswitch are recapitulated by genetic mutations in the condensate forming protein that alter condensate material properties;
- iv) we provide additional orthogonal evidence that the killswitch affects material properties of two condensates using a common condensate dissolution agent.

The newly added data provide additional support for the model that the killswitch alters condensate composition, component dynamics and function through immobilizing the targeted condensate-forming protein within condensates. The results collectively provide **direct evidence for specialized microenvironments within condensates in cells, which enable non-stoichiometric control of cellular activity**. The results create an experimental and conceptual framework to experimentally dissect the molecular basis of non-stoichiometric partitioning of proteins into condensates, which is a major outstanding frontier. Therefore, we believe the revised manuscript contains improved and important technological and conceptual advances.

The changes in the revised manuscript are described in detail, point-by-point addressing the Reviewer's comments. Our responses are in blue font. Based on the Reviewers' comments we added 9 new Main Figure panels, 43 new Extended Data Figure panels and two new Supplementary Tables. The changes in the manuscript files and the reference to the new figure panels are highlighted in blue font.

Referee #1 (Remarks to the Author):

In this paper, Zhang et al., co-opt a sequence from a pathogenic HMGB-fusion protein containing a C-terminal frameshift to create a new tool (the “Killswitch”) that binds and reduces the dynamics of nucleolar proteins. They show that this 17 a.a. peptide self-assembles and disruption of self-assembly reduced the efficacy of the Killswitch (F-A mutant). Thus, the Killswitch modifies condensate properties (solidifies them) by adding valence. By coupling the Killswitch to a GFP nanobody, they demonstrate that the Killswitch can alter the dynamic properties and activities of a wide range of naturally occurring and pathogenic condensates, however with the condition that these condensates contain a GFP-fusion protein. Throughout the paper, the authors include the dead Killswitch (F-A mutant) as a vital control to show that the effect is due to Killswitch self-interactions, not necessarily through heterotypic interactions between the Killswitch and proteins within the target condensate.

In my opinion, the results of this study are convincing and interesting. The exact mechanism of how the Killswitch works may not be fully understood, and it may well be context dependent, but that doesn’t dim my enthusiasm. Reducing condensate dynamics through added valence is not a novel idea, but the Killswitch is a versatile tool, and many condensate researchers will be interested in using it. The main downside of this paper is that the Killswitch is currently limited to GFP-tagged proteins, so we don’t know if it can be used as a therapeutic for diseases involving the formation of pathological condensates. I discuss these points in more detail below.

We thank the reviewer for the insightful comments and the guidance on improvements.

In the revised manuscript we include **five new approaches** to recruit the killswitch to condensates. These approaches include three different nanobodies, a cell-permeable peptide-based delivery system, and a trans-recruitment system using ectopic protein. Recruiting the killswitch with all five additional approaches led to **measurable cellular effects**. The new results demonstrate that the killswitch can be targeted to non-GFP-tagged proteins, and indicate substantially broader utility of the killswitch than in the original submission. The new data are described below in detail.

As a proof-of-concept we expressed GFP-NPM1 fused to three different epitope tags (V5, VHH05 and ALFA-tag), and co-expressed the corresponding nanobodies, the nanobodies fused to the killswitch or the F-to-G killswitch rescue variant. In all three systems, the Nanobody fused to the killswitch immobilized GFP-NPM1 in nucleoli. The data are included as new **Extended Data Fig. 7a-d**. These results demonstrate that targeting the killswitch works with several nanobodies beyond the anti-GFP nanobody described in the first submission.

Second, we generated an oligo-arginine peptide (R10), and the same peptide fused to the killswitch or the F-to-G killswitch rescue variant. These peptides are cell-permeable, and localize to the nucleolus, as expected from previous studies¹⁻⁴. We found that the R10-Killswitch peptide almost instantaneously killed the cells that took up the peptide. In the dying cells, the R10-Killswitch peptide localized to the nucleolus, and NPM1 was immobilized in the nucleolus of the dying cells. Under identical conditions, the control peptides did not kill the cells, and did not immobilize NPM1. The data are included as new **Extended Data Fig. 12a-h**. These results demonstrate that targeting the killswitch to nucleoli works even without a nanobody.

Third, we delivered Nanog-killswitch RNA into Zebrafish embryos. Nanog contributes to two large transcription bodies in early Zebrafish embryos which transcribe a microRNA-cluster

mir430⁵. The in vivo translated Nanog-killswitch protein was sufficient to repress the activity of the transcriptional body on the microRNA-cluster. The data are included as new **Extended Data Fig. 23a-d**. The results demonstrate that targeting the killswitch works on untagged protein and using an RNA-delivery approach.

We realize that the five additional alternative targeting systems are still rather lab tools than therapeutic agents, therefore we also toned down sentences referring to therapeutic implications in the text. We nevertheless strongly believe that the five additional targeting and delivery systems we describe in the revised manuscript provide a strong foundation for further development of the killswitch as proof-of-concept therapeutics.

We also include substantial new data that provide new insights into the mechanism of the killswitch. In brief, for four different condensates, we found that the killswitch selectively alters condensate composition, component dynamics and function. We provide additional experimental data that the changes are caused by altered material properties of the condensates. The data are described in detail at the response to Reviewers 2 and 3 below.

Major points:

Interpretation of the results may be difficult due to complexity of altering material properties. For example, the Killswitch may alter other features of condensates, such as growth, fusion, or susceptibility to proteasome-mediated degradation. To me, that seems irrelevant at this point. The Killswitch's effect is likely context-dependent, and it will be up to future studies to determine how the Killswitch specifically affects each condensate. Thus, it is hard to say that the effect is strictly due to a change in material properties. Again, the same thing could be said for any perturbation that solidifies the condensate. Adding some caveats on this point in the discussion would be useful.

The point of better linking changes of condensate material properties and function is important and appreciated. In the revised manuscript we further substantiate that condensate compositional and component dynamic changes elicited by the killswitch are caused by the change in material properties of the targeted proteins. First, we mutagenized the NUP98 IDR in the NUP98::DDX10 fusion protein by clustering short motifs that contain aromatic residues. Previous work suggested that clustering aromatic residues in condensate forming proteins arrests the dynamics the mutant protein in condensates^{6,7}. Indeed, we found that the clustered aromatic mutant NUP98::DDX10 formed condensates in which the dynamics of both NUP98::DDX10 and its client CRM1 was reduced (**new Figure 5d-e**). The effect of the mutation phenocopies the effect targeting the killswitch to NUP98::DDX10 (**Figure 5a-c**). Moreover, we treated cells expressing NUP98::DDX10 and BRD4-NUT condensates or the fusions fused to the killswitch with 1,6-hexanediol, an aliphatic alcohol that ubiquitously dissolves condensates⁸. We found that the killswitch-tagged NUP98::DDX10 and BRD4-NUT condensates were resistant to hexanediol treatment, whereas the wild types condensates were dissolved (**new Extended Data Fig. 14h-i, 15e-f**). Collectively these data provide further evidence that the killswitch affects the material properties of condensates.

One must consider this paper in the context of other studies inventing clever ways of modifying condensate properties. For example, Cliff Brangwynne's lab already devised a way to solidify condensates with the Opto-droplet tagging system. Deep quenching, or long exposures to blue light, drives condensates into an arrested state (PMC5562165). Other studies have used Cry2o or FKBP-FRB (PMC9669173) to add valence. Disruption of condensate properties with small

molecules was done in Hernandez-Candia 2021. Gui et al (2023 Mol Cell) developed short peptides to perturb condensate formation and activity (YAP, TAZ, Hippo pathway). Thus, the killswitch is another way to perturb condensates by, presumably, making their connections stronger and decreasing dynamics. However, each of the perturbations in these previous studies are specific to those engineered systems; in other words, they do not present a broadly applicable technology. By adding the Killswitch to the GFP nanobody, this study makes a user-friendly tool to study all condensates containing GFP-tagged proteins. Thus, while the technological and conceptual leap is not that advanced, the product will be useful to the broader condensate community.

We thank the reviewer for the comments. We cite all these studies in the manuscript. We of course agree with the Reviewer that other investigators have developed useful approaches to probe condensates, and that the killswitch provides unique advantages.

We strongly believe that targeting the killswitch to various condensates revealed insights that do constitute an important conceptual leap. The new data in the revised manuscript are particularly important in this regard. In the revised manuscript, we developed **a FACS-based method to isolate nucleoli** and killswitch-targeted nucleoli directly from cell lysates, and performed proteomics analysis using Mass Spectrometry. The results revealed that a set of ~20 RNA-binding proteins are selectively depleted from killswitch-targeted nucleoli (**new Figure 3a-c, Extended Data Figure 9a-h**). The killswitch system and the sorting approach in itself represent a technological leap. Moreover, the data provide empirical evidence that the microenvironment within a cellular condensate impacts non-stoichiometric recruitment of client proteins. It is key to note that all the protein-protein interaction interfaces on the proteins are present, yet there is a selective depletion of a set of ~20 RNA-binding proteins in nucleoli targeted with the killswitch. *These results create a foundation for future work on dissecting the molecular basis of the selective depletion and partitioning, which is an important outstanding problem.*

The paper makes several points about how we need tools to disrupt endogenous condensates, which would have profound implications as a therapeutic:

“A few studies have reported protein-tagging approaches to dissolve ectopic condensates 22,31,32, but none were tested on endogenous condensates, that are known to significantly differ in their biophysical and functional properties from condensates formed by ectopically overexpressed proteins 5,33.”

“Condensate alterations have been linked to hundreds of human diseases, notably, cancers driven by fusion oncoproteins, and rare genetic diseases, and these conditions typically have no therapeutic remedy 34-36. We therefore tested the effect of recruiting the killswitch to multiple disease-specific condensates.”

Herein lies the main weakness of the study. Their current Killswitch can only recognize GFP-tagged proteins. Thus, it cannot be used on endogenous condensates in unmodified cells or as a therapeutic. If the authors could design a Killswitch that targets a naturally occurring, unlabeled protein, that would be a tremendous breakthrough. For example, make a nanobody against the Nup98::KDM5A fusion protein, then label that nanobody with the Killswitch peptide. Does it still stop proliferation of the murine AML cells? If so, they would have generated a drug to target the endogenous pathogenic protein. This experiment would also further demonstrate the applicability of this technique to non-GFP-tagged proteins. Considering the time and potential difficulty of generating such a nanobody, I leave this suggestion to the discretion of the editor.

In the revised manuscript we include **five new approaches** to recruit the killswitch to condensates. These approaches include three different nanobodies, a cell-permeable peptide-based delivery system, and a trans-recruitment system using ectopic protein. The latter two are independent of nanobodies, and target naturally occurring endogenous proteins. Recruiting the killswitch with all five additional approaches led to **measurable cellular effects**. The new results therefore demonstrate that the killswitch can be targeted to non-GFP-tagged proteins, and indicate substantially broader utility of the killswitch than in the original submission. The new data are described above at the general response to Reviewer 1.

We realize that the five additional alternative targeting systems are still rather lab tools than therapeutic agents, therefore we also toned down sentences referring to therapeutic implications in the text. We nevertheless strongly believe that the five additional targeting and delivery systems we describe in the revised manuscript provide a strong foundation for further development of the killswitch as proof-of-concept therapeutics. We are in the process of generating nanobodies against endogenous proteins (e.g., a fusion-specific NUP98 nanobody), but this is less than trivial because the endogenous condensate forming proteins are highly enriched in intrinsic disorder.

Referee #2 (Remarks to the Author):

The work by Zhang et al addresses an important challenge in understanding the role of biomolecular condensates in cells. Despite their ubiquitous distribution, dissection of their roles is often very tricky since mutations of condensate-forming proteins are often associated with other functions of these proteins that are difficult to decouple of their propensity to undergo phase transitions. The development of a "killswitch" peptide to probe condensate microenvironments represents a significant advancement for the field. The authors employ a clever strategy by using a non-natural micropeptide combined with a nanobody-based recruitment system. This allows for precise manipulation and observation of condensates within living cells, and the choice of targets, such as nucleolar proteins and oncoprotein condensates, is well-justified and demonstrates the versatility of the approach. Together, these findings provide a new tool for studying the functional dynamics of cell condensates. The manuscript is well-organized and clearly written, and the findings are clearly presented, highlighting the power of their approach in revealing condensate's functions. One of the weaknesses is the lack of the molecular mechanism of the killswitch and how it specifically interacts with target proteins in condensates. Another weakness is that the authors only demonstrate the applicability of the KS with a nanobody targeting GFP. It would be useful to illustrate how other nanobodies targeting scaffold or client proteins directly can also recruit KS in natural systems without GFP, e.g., a killswitch fused to peptides/proteins that naturally partition to condensates rather than fused to nanobodies.

We thank the reviewer for the insightful comments and the guidance on improvements. We include substantial new data on the mechanism of the killswitch and on alternative targeting approaches in the revised manuscript.

Mechanism

In the revised manuscript we include new data that provide several new experimental approaches and results that provide further insights into the mechanism of the killswitch. In brief, for four different condensates, we found that the killswitch selectively alters condensate composition, component dynamics and function, through immobilizing the targeted condensate-forming protein within condensates.

In the revised manuscript, we developed a FACS-based method to isolate nucleoli and killswitch-targeted nucleoli directly from cell lysates, and performed proteomics analysis using Mass Spectrometry. The results revealed that a set of ~20 RNA-binding proteins are selectively depleted from killswitch-targeted nucleoli (**new Figure 3a-c, Extended Data Figure 9a-h**). The killswitch system and the sorting approach in itself represent a technological leap. Moreover, the data provide empirical evidence that the microenvironment within a cellular condensate impacts non-stoichiometric recruitment of client proteins. It is key to note that all the protein-protein interaction interfaces on the proteins are present, yet there is a selective depletion of a set of ~20 RNA-binding proteins in nucleoli targeted with the killswitch. The results of the nucleoli-proteomics data are consistent with what we observed for other condensates. For example, we found that the killswitch depleted RNAPII from BRD4-NUT condensates (**Fig. 4d-e**), and IIIa protein from adenoviral 52K condensates (**Fig. 6g-h**). *These results create a foundation for future work on dissecting the molecular basis of the selective depletion and partitioning, which is an important outstanding problem.*

Furthermore, we include data from FRAP experiments consistently showing that immobilizing a condensate-forming protein with the killswitch inhibits the dynamics of client proteins in the

condensates (RPL18 in nucleoli: **Fig. 3g**, p300 in BRD4-NUT condensates **new Fig. 4f**; CRM1 in NUP98::DDX10 condensates: **Fig 5b**.)

Moreover, we further substantiate that condensate compositional and component dynamic changes elicited by the killswitch is caused by the change in material properties of the targeted proteins with mutagenesis experiments and 1,6-hexanediol treatment described in detail at the response of Major Point 1 of Reviewer 1 (**new Fig 5d-e, Extended Data Fig. 14h-i, 15e-f**).

In summary, for four different condensates, we found that the killswitch selectively alters condensate composition, component dynamics and function through immobilizing the targeted condensate-forming protein.

Alternative targeting approaches

(The text below is included in the general Response to Reviewer 1, who had the same concern.)

In the revised manuscript we include **five new approaches** to recruit the killswitch to condensates. These approaches include three different nanobodies, a cell-permeable peptide-based delivery system, and a trans-recruitment system using ectopic protein. Recruiting the killswitch with all five additional approaches led to **measurable cellular effects**. The new results therefore demonstrate that the killswitch can be targeted to non-GFP-tagged proteins, and indicate substantially broader utility of the killswitch than in the original submission. The new data are described below in detail.

As a proof-of-concept we expressed GFP-NPM1 fused to three different epitope tags (V5, VHH05 and ALFA-tag), and co-expressed the corresponding nanobodies, the nanobodies fused to the killswitch or the F-to-G killswitch rescue variant. In all three systems, the Nanobody fused to the killswitch immobilized GFP-NPM1 in nucleoli. The data are included as new **Extended Data Fig. 7a-d**. These results demonstrate that targeting the killswitch works with several nanobodies beyond the anti-GFP nanobody described in the first submission.

Second, we generated an oligo-arginine peptide (R10), and the same peptide fused to the killswitch or the F-to-G killswitch rescue variant. These peptides (as expected from previous studies) are cell-permeable, and localize to the nucleolus. We found that the R10-Killswitch peptide almost instantaneously killed the cells that took up the peptide. In the dying cells, the R10-Killswitch peptide localized to the nucleolus, and NPM1 was immobilized in the nucleolus of the dying cells. Under identical conditions, the control peptides did not kill the cells, and did not immobilize NPM1. The data are included as new **Extended Data Fig. 12a-h**. These results demonstrate that targeting the killswitch to nucleoli works even without a nanobody.

Third, we delivered Nanog-killswitch RNA into Zebrafish embryos. Nanog contributes to two transcribe two large transcription bodies in early Zebrafish embryos which transcribe a microRNA-cluster⁵. The translated Nanog-killswitch protein was sufficient to repress the activity of the transcriptional body on the microRNA-cluster. The data are included as new **Extended Data Fig. 23a-d**. These results demonstrate that targeting the killswitch works on untagged protein and using an RNA-delivery approach.

Major considerations:

1. The authors refer to the material properties throughout the text, however, they only rely on FRAP data – I would rephrase this as without actual material property measurement techniques

(and FRAP isn't one of the methods) one cannot comment on whether these condensates are solid/gel-like or liquid-like.

The point of better linking changes of condensate material properties and function is important and appreciated. In the revised manuscript we further substantiate that condensate compositional and component dynamic changes elicited by the killswitch is caused by the change in material properties of the targeted proteins. First, we mutagenized the NUP98 IDR in the NUP98::DDX10 fusion protein by clustering short motifs that contain aromatic residues. Previous work suggested that clustering aromatic residues in condensate forming proteins arrests the dynamics the mutant protein in condensates ^{6,7}. Indeed, we found that the clustered aromatic mutant NUP98::DDX10 formed condensates in which the dynamics of both NUP98::DDX10 and its client CRM1 was reduced (**new Figure 5d-e**). The effect of the mutation phenocopies the effect targeting the killswitch to NUP98::DDX10 (**Figure 5a-c**). Moreover, we treated cells expressing NUP98::DDX10 and BRD4-NUT condensates or the fusions fused to the killswitch with 1,6-hexanediol, an aliphatic alcohol that ubiquitously dissolves condensates ⁸. We found that the killswitch-tagged NUP98::DDX10 and BRD4-NUT condensates were resistant to hexanediol treatment, whereas the wild types condensates were dissolved (**new Extended Data Fig. 14h-i, 15e-f**). Collectively these data provide further evidence that the killswitch affects the material properties of condensates. At the same time, taking the guidance of the reviewer, we refer to the effect of the killswitch as “immobilizes the targeted protein in condensates” wherever appropriate.

Further comments:

107: Partitioning into nucleolus is reported, but it is not clear which regions have affected mobility. FC, DFC, GC specifically? If all, how/why?

The partitioning is reported for the entire nucleolus. In these experiments, an RFP-FIB1 vector is co-transfected with the GFP-HMGB1 vector to serve as a guidance on the location of the FC/DFC. The GFP-HMGB1 protein is mildly enriched in the GC but is also found in the FC/DFC. Some mutants, e.g. the delta killswitch mutant has similar partitioning in the entire nucleolus, and the RFP-FIB1 overlaps with the GFP signal (i.e. the RFP does not precisely delineate where the GC/DFC boundary is (see e.g. Extended Data Fig. 1b). In the FRAP experiments, we bleach the GFP-HMGB1 in what is predicted to be the GC, but cannot rule out that the bleaching affects the DFC/FC.

111: The removal of Phe residues from KS inactivates it – can other aromatic residues restore its activity? Can the authors comment on whether the removal of the aromatic residues affects the (predicted) helical conformation of KS?

In the revised manuscript we mutagenized the phenylalanines into tyrosines or tryptophans. In both cases the killswitch still immobilized the HMGB1 protein it was fused to, but the effect was less pronounced than with phenylalanines (**new Extended Data Fig. 1c-d**). The removal of the aromatic residues is predicted to reduce the helical propensity of the killswitch by AlphaFold (**Extended Data Fig. 4d-e**). In our previous work we showed with circular dichroism experiments that the killswitch has a mild helical propensity but it is clearly not a stable helix ⁹.

175: There's no justification of why 2 killswitch-Nanobodies are necessary. Also, an earlier justification of Nanobodies over other delivery systems is missing. An additional/orthogonal

delivery approach would strengthen this manuscript significantly. Does the location of the KS peptide (C- or N-terminal) affect its ability to interfere with condensates?

We believe that the killswitch is unique as a tool in its ability to experimentally inform on the nature of the interactions between the condensate-forming proteins in the condensate. Significant theoretical work indicates that the valence and affinity of interactions determine parameters of condensate formation and material properties¹⁰⁻¹². As such, the valence added by the killswitch, and the hydrophobic contacts through which the killswitch adds valence can have different effects on condensates. We think the effect is therefore informative on the nature of the interactions within the condensates. We added text to lines 372-377 accordingly.

The justification of using the GFP-nanobody system was that there are many available cell lines with integrated GFP-tags on endogenous proteins (see e.g., OpenCell¹³), and such cell lines can be immediately useful for targeting endogenously GFP-labelled condensates with the killswitch. The justification is included in the text in lines 149-153.

In the revised manuscript we include **five new delivery approaches**. These data and results are described in detail above at the general response to the Reviewer.

When genetically fusing the killswitch to a condensate-forming protein, the location of the killswitch matters. On the N-terminus, the killswitch tends to affect protein sorting and/or folding. Throughout the study we use the killswitch on the C-terminus as a genetic fusion.

Can the authors discuss the underlying reasons why some condensates were less affected by the killswitch or needed double KS?

We added text to lines 374-377 accordingly (as detailed at the comment above).

181: Why is the formation of spherical structures used as evidence of hardening? FRAP data in 2E shows little reduction for a single killswitch. Can the authors explain this?

The spherical structures are not used as evidence for hardening. We simply documented that we observed both. Nuclear speckles for example over time collapse into spherical structures, and the material in the spherical structures has reduced recovery after photobleaching.

The way we interpret these results is that there is more valence contributed to the condensates by tandem two copies of the killswitch than one. Also, there is likely a critical threshold of how many protein molecules need to be bound by the killswitch to affect the condensates. In the current system we cannot quantify the amount of Nanobody-killswitch in the condensates (because the large size of adding a fluorophore causes artefacts, see e.g. Extended Data Fig. 6a).

188: GC is the assembly site of early/pre-ribosomes. rRNA processing occurs primarily in the dense fibrillar component (DFC) of the nucleolus.

Thank you! We fixed the sentence accordingly.

195, 205: The authors report that only certain proteins' mobility is affected by killswitch rather than condensate-wide effects. If other proteins and RNA behave normally, the impact of the killswitch and usefulness as a tool in question. The authors don't address these concerns, instead using findings to argue against previous findings.

We favor a different model, as described above. We believe that the killswitch is unique as a tool in its ability to experimentally inform on the nature of the interactions between the condensate-forming proteins in the condensate. We added text to lines 374-377 accordingly. Also, there is an effect on RNA (Fig 3h-i). To minimize confusion, we removed the problematic cited sentences from the text.

253: why does killswitch affect the partitioning of some constituents but not others? A discussion on possible causes for why killswitch affects some but not all components of condensate, including whether this is a nanobody-induced effect is required.

We added text to lines 374-377 accordingly (as detailed at the comments above).

Fig 1E: The presentation of FRAP data doesn't show the recovery rate of mobile fraction, e.g. if slow but still mobile.

We added the individual FRAP curves as **new Extended Data Fig. 2c**.

Fig 2D: GFP HP1a shows very high background, aberrant effects of killswitch from nonspecific targeting. Also, the ordering of nanobody imaging conditions is mixed up between the first & last 2 conditions, which is confusing.

Indeed, HP1a has a lower partition ratio in cells than e.g., NPM, and GFP-HP1a has a substantial soluble fraction. To rule out that the killswitch affects soluble proteins we performed FRAP experiments, and found no effect of the killswitch on soluble proteins (Extended Data Fig. 22). These experiments were not done on HP1a. We acknowledge specifically in the text that it remains a possibility that the killswitch affects soluble proteins (even though we find no evidence for it in our simple tests), in lines 367-369.

Fig 4C: A control with a marker localizing specifically to target condensates should be included. There appears to be a high background of killswitch in the cytosol, possibly confounding effects. The Phe to Ala control wouldn't aggregate/form condensates alone/with other molecules in the cytosol, therefore it isn't necessarily an adequate control/comparison when investigating effects beyond condensate level, e.g. cell viability.

We assume the reviewer refers to the 52K-IIIa-GFP co-localization experiment that was moved to Fig. 6g in the revised manuscript. It is well documented that much of the capsid protein IIIa is not nuclear, but it is present in the nucleus within the 52K condensates linked to viral progeny production (see e.g. Figure 2h in Charman et al., 2023¹⁴). It is important to note that in our data, the amount of nuclear IIIa-GFP is similar in cells that express the 52K wild type protein and the 52K-KS protein (**Fig. 6h**). Therefore, the reduced partitioning of IIIa-GFP into the 52K-KS condensates versus into control condensates cannot be explained by differences in the protein levels.

Referee #3 (Remarks to the Author):

Zhang et al. report on repurposing a short peptide sequence, found as a naturally occurring pathogenic frameshift variant of HMGB1, as a modulator of condensate function. The authors show some interesting observations in vitro and cells. The peptide appears to self-associate in some way, and thought this self-association, perturbs condensate function. These are interesting observations, but the paper is slightly let down by a lack of characterisation/understanding of the mechanism of action. The effects are also slightly hard to understand/rationalise, for instance in the case of NPM1. It is therefore not clear to this reviewer whether the findings will well enough defined to be of value as a condensate modulation approach.

We thank the reviewer for the straightforward and incisive assessment. Indeed, some of the interesting observations were preliminary, and we added substantial new data on mechanism of action of the killswitch in the revised manuscript. *Below is a brief summary, which is the same text as parts of the general response to Reviewer 2 who had a similar comment.* More detail on specific experiments are further included at the responses to the specific comments of Reviewer 1 and 2.

In the revised manuscript, we developed a FACS-based method to isolate nucleoli and killswitch-targeted nucleoli directly from cell lysates, and performed proteomics analysis using Mass Spectrometry. The results revealed that a set of ~20 RNA-binding proteins are selectively depleted from killswitch-targeted nucleoli (**new Figure 3a-c, Extended Data Figure 9a-g**). The killswitch system and the sorting approach in itself represent a technological leap. Moreover, the data provide empirical evidence that the microenvironment within a cellular condensate impacts non-stoichiometric recruitment of client proteins. It is key to note that all the protein-protein interaction interfaces on the proteins are present, yet there is a selective depletion of a set of ~20 RNA-binding proteins in nucleoli targeted with the killswitch. The results of the nucleoli-proteomics data are consistent with what we observed for other condensates. For example, we found that the killswitch depleted RNAPII from BRD4-NUT condensates (**Fig. 4d-e**), and IIIa protein from adenoviral 52K condensates (**Fig. 6g-h**). *These results create a foundation for future work on dissecting the molecular basis of the selective depletion and partitioning, which is an important outstanding problem.*

Furthermore, we include data from FRAP experiments consistently showing that immobilizing a condensate-forming protein with the killswitch inhibits the dynamics of client proteins in the condensates (RPL18 in nucleoli: **Fig. 3g**, p300 in BRD4-NUT condensates **new Fig. 4f**; CRM1 in NUP98::DDX10 condensates: **Fig 5b**.)

Moreover, we further substantiate that condensate compositional and component dynamic changes elicited by the killswitch is caused by the change in material properties of the targeted proteins with mutagenesis experiments and 1,6-hexanediol treatment described in detail at the response of Major Point 1 of Reviewer 1 (**new Fig 5d-e, Extended Data Fig. 14h-i, 15e-f**).

In summary we include new data that provide several new experimental approaches and results that provide insights into the mechanism of the killswitch. For four different condensates, we found that the killswitch selectively alters condensate composition, component dynamics and function, through immobilizing the targeted condensate-forming protein within condensates. The approach also revealed surprises, for instance a condensate-dependent degradation pathway in leukemia cells.

Referee #4 (Remarks to the Author):

Yaotian Zhang and colleagues report a genetically encodable peptide “killswitch” for probing how the material properties of biomolecular condensates contribute to cellular functions. The work originates from the authors’ previous observation of a frameshift mutation in HMGB1 that leads to nucleolar partitioning and disruption of nucleolar dynamics. The authors identify a 17-aa C-terminal peptide from the mutant HMGB1 protein as being responsible for altering the liquid-like behavior of the nucleolus. After some further characterization of the molecular mechanism, including mutational analyses and structural modeling, the authors set out to apply the killswitch peptide as a molecular tool to investigate the link between the material properties of various condensates and their function in cells. To do so, the authors append the killswitch peptide to an anti-GFP nanobody to enable recruitment to GFP-tagged proteins known to form nuclear condensates, including both endogenously labeled native proteins and overexpressed oncogenic fusions.

The need for tools to investigate endogenous biomolecular condensates in cells cannot be understated. However, it may be difficult to use the killswitch. The peptide gets incorporated into condensates and can lead to more gel-like properties. But different condensates have different characteristic dynamics and vary in their native “liquid” or “gel-like” behaviors – the material properties are different for different systems and potentially for individual condensates with the cell. Thus, the killswitch may not be effective in condensates that are naturally more gel-like. As shown by the authors’ data, incorporation of the killswitch can have different effects, including minimal changes, partial changes in the makeup of the condensates, or triggering the activation of degradation machinery. Whether these effects can all be linked directly to changes in the material properties of the corresponding condensates remains an open question. The FRAP analyses performed in the study provide an incomplete picture. Both the native material properties of the condensates and the killswitch-induced changes in these properties need to be more systematically and quantitatively characterized. As it currently stands, the data obtained using the killswitch cannot be reliably interpreted, casting doubt on the potential for broader application of this tool.

We thank the reviewer for the comments. We include substantial new data on the mechanism of the killswitch and on alternative targeting approaches in the revised manuscript. *(The text below is included in the general Response to Reviewer 1, who had the similar comments.)*

Mechanism

In the revised manuscript we include new data that provide several new experimental approaches and results that provide further insights into the mechanism of the killswitch. In brief, for four different condensates, we found that the killswitch selectively alters condensate composition, component dynamics and function, through immobilizing the targeted condensate-forming protein within condensates.

In the revised manuscript, we developed a FACS-based method to isolate nucleoli and killswitch-targeted nucleoli directly from cell lysates, and performed proteomics analysis using Mass Spectrometry. The results revealed that a set of ~20 RNA-binding proteins are selectively depleted from killswitch-targeted nucleoli (**new Figure 3a-c, Extended Data Figure 9a-g**). The killswitch system and the sorting approach in itself represent a technological leap. Moreover, the data provide empirical evidence that the microenvironment within a cellular condensate impacts non-stoichiometric recruitment of client proteins. It is key to note that all the protein-protein interaction interfaces on the proteins are present, yet there is a selective depletion of a set of ~20 RNA-binding proteins in nucleoli targeted with the killswitch. The results of the nucleoli-

proteomics data are consistent with what we observed for other condensates. For example, we found that the killswitch depleted RNAPII from BRD4-NUT condensates (**Fig. 4d-e**), and IIIa protein from adenoviral 52K condensates (**Fig. 6g-h**). *These results create a foundation for future work on dissecting the molecular basis of the selective depletion and partitioning, which is an important outstanding problem.*

Furthermore, we include data from FRAP experiments consistently showing that immobilizing a condensate-forming protein with the killswitch inhibits the dynamics of client proteins in the condensates (RPL18 in nucleoli: **Fig. 3g**, p300 in BRD4-NUT condensates **new Fig. 4f**; CRM1 in NUP98::DDX10 condensates: **Fig 5b**.)

Moreover, we further substantiate that condensate compositional and component dynamic changes elicited by the killswitch is caused by the change in material properties of the targeted proteins with mutagenesis experiments and 1,6-hexanediol treatment, described in detail at the response of Major Point 1 of Reviewer 1 (**new Fig 5d-e, Extended Data Fig. 14h-i, 15e-f**).

In summary, for four different condensates, we found that the killswitch selectively alters condensate composition, component dynamics and function through immobilizing the targeted condensate-forming protein.

Alternative targeting approaches

In the revised manuscript we include **five new approaches** to recruit the killswitch to condensates. These approaches include three different nanobodies, a cell-permeable peptide-based delivery system, and a trans-recruitment system using ectopic protein. Recruiting the killswitch with all five additional approaches led to **measurable cellular effects**. The new results therefore demonstrate that the killswitch can be targeted to non-GFP-tagged proteins, and indicate substantially broader utility of the killswitch than in the original submission. The new data are described below in detail.

As a proof-of-concept we expressed GFP-NPM1 fused to three different epitope tags (V5, VHH05 and ALFA-tag), and co-expressed the corresponding nanobodies, the nanobodies fused to the killswitch or the F-to-G killswitch rescue variant. In all three systems, the Nanobody fused to the killswitch immobilized GFP-NPM1 in nucleoli. The data are included as new **Extended Data Fig. 7a-d**. These results demonstrate that targeting the killswitch works with several nanobodies beyond the anti-GFP nanobody described in the first submission.

Second, we generated an oligo-arginine peptide (R10), and the same peptide fused to the killswitch or the F-to-G killswitch rescue variant. These peptides (as expected from previous studies) are cell-permeable, and localize to the nucleolus. We found that the R10-Killswitch peptide almost instantaneously killed the cells that took up the peptide. In the dying cells, the R10-Killswitch peptide localized to the nucleolus, and NPM1 was immobilized in the nucleolus of the dying cells. Under identical conditions, the control peptides did not kill the cells, and did not immobilize NPM1. The data are included as new **Extended Data Fig. 12a-h**. These results demonstrate that targeting the killswitch to nucleoli works even without a nanobody.

Third, we delivered Nanog-killswitch RNA into Zebrafish embryos. Nanog contributes to two transcribe two large transcription bodies in early Zebrafish embryos which transcribe a microRNA-cluster⁵. The translated Nanog-killswitch protein was sufficient to repress the activity of the transcriptional body on the microRNA-cluster. The data are included as new

Extended Data Fig. 23a-d. These results demonstrate that targeting the killswitch works on untagged protein and using an RNA-delivery approach.

References

- 1 Schneider, A. F. L., Kithil, M., Cardoso, M. C., Lehmann, M. & Hackenberger, C. P. R. Cellular uptake of large biomolecules enabled by cell-surface-reactive cell-penetrating peptide additives. *Nat Chem* **13**, 530-539, doi:10.1038/s41557-021-00661-x (2021).
- 2 Brunner, J. & Barton, J. K. Targeting DNA mismatches with rhodium intercalators functionalized with a cell-penetrating peptide. *Biochemistry* **45**, 12295-12302, doi:10.1021/bi061198o (2006).
- 3 Schneider, A. F. L., Wallabregue, A. L. D., Franz, L. & Hackenberger, C. P. R. Targeted Subcellular Protein Delivery Using Cleavable Cyclic Cell-Penetrating Peptides. *Bioconjug Chem* **30**, 400-404, doi:10.1021/acs.bioconjchem.8b00855 (2019).
- 4 Kwon, I. *et al.* Poly-dipeptides encoded by the C9orf72 repeats bind nucleoli, impede RNA biogenesis, and kill cells. *Science* **345**, 1139-1145, doi:10.1126/science.1254917 (2014).
- 5 Kuznetsova, K. *et al.* Nanog organizes transcription bodies. *Current biology : CB* **33**, 164-173 e165, doi:10.1016/j.cub.2022.11.015 (2023).
- 6 Holehouse, A. S., Ginell, G. M., Griffith, D. & Boke, E. Clustering of Aromatic Residues in Prion-like Domains Can Tune the Formation, State, and Organization of Biomolecular Condensates. *Biochemistry* **60**, 3566-3581, doi:10.1021/acs.biochem.1c00465 (2021).
- 7 Martin, E. W. *et al.* Valence and patterning of aromatic residues determine the phase behavior of prion-like domains. *Science* **367**, 694-699, doi:10.1126/science.aaw8653 (2020).
- 8 Alberti, S., Gladfelter, A. & Mittag, T. Considerations and Challenges in Studying Liquid-Liquid Phase Separation and Biomolecular Condensates. *Cell* **176**, 419-434, doi:10.1016/j.cell.2018.12.035 (2019).
- 9 Mensah, M. A. *et al.* Aberrant phase separation and nucleolar dysfunction in rare genetic diseases. *Nature*, doi:10.1038/s41586-022-05682-1 (2023).
- 10 Choi, J. M., Holehouse, A. S. & Pappu, R. V. Physical Principles Underlying the Complex Biology of Intracellular Phase Transitions. *Annu Rev Biophys* **49**, 107-133, doi:10.1146/annurev-biophys-121219-081629 (2020).
- 11 Harmon, T. S., Holehouse, A. S., Rosen, M. K. & Pappu, R. V. Intrinsically disordered linkers determine the interplay between phase separation and gelation in multivalent proteins. *eLife* **6**, doi:10.7554/eLife.30294 (2017).
- 12 Pappu, R. V., Cohen, S. R., Dar, F., Farag, M. & Kar, M. Phase Transitions of Associative Biomacromolecules. *Chem Rev* **123**, 8945-8987, doi:10.1021/acs.chemrev.2c00814 (2023).
- 13 Cho, N. H. *et al.* OpenCell: Endogenous tagging for the cartography of human cellular organization. *Science* **375**, eabi6983, doi:10.1126/science.abi6983 (2022).
- 14 Charman, M. *et al.* A viral biomolecular condensate coordinates assembly of progeny particles. *Nature* **616**, 332-338, doi:10.1038/s41586-023-05887-y (2023).

Referees' comments: (Responses in blue font)

Referee #1 (Remarks to the Author):

The reviewers have addressed my previous major concerns. I think the manuscript is suitable for publication provided that they address a few minor points detailed below.

We thank the reviewer for the guidance and help throughout the revision process! We addressed the remaining few minor points as described below.

Overview:

The mechanism of killswitch activity is well documented and convincing. The authors demonstrate that homotypic interactions driven largely by three phenylalanines underlie the changes in condensate dynamics. An F-to-G mutant kills this activity, and both size exclusion chromatography and AF3 modeling indicate that this is because these mutations prevent multimerization of the killswitch. Thus, I agree with the authors that the Killswitch most likely works through added valence, which should increase the connectivity of the tagged protein (via homotypic and not heterotypic interactions) and therefore decrease the dynamics of the tagged molecule.

The new experiments in Figure 3 showing that the Killswitch can change the composition of condensates is astounding. The authors thus strengthen their story by adding another layer to the mechanism by which the Killswitch affects condensate function.

My main concern of the previous version was that the utility of the Killswitch was limited by its fusion to a GFP-nanobody. Thus, only non-native, GFP-labeled proteins could be affected. The authors present many disease-relevant models where perturbing condensate function could be a cure. They speculated that the Killswitch could be a therapeutic, but this would depend upon developing a Killswitch that recognizes endogenous proteins.

In their revised version, the authors present several additional methods of targeting the Killswitch to condensates. They fuse the Killswitch to three additional nanobodies that recognize widely used epitopes (e.g., HA tag, ALFA tag, VH005 tag). They also fuse the Killswitch to an R10 peptide that is membrane-permeable and targets the nucleolus. In extended Fig. 23 they demonstrate the Killswitch can be introduced through injecting mRNA encoding a fusion protein (Nanog-KS). The authors claim they have yet to make a Killswitch that targets endogenous, disease-causing condensates. This could be due to the difficulty in making a highly-specific nanobody.

While the authors could not completely address my main concern, the additional methods of introducing the Killswitch strengthen the paper and broaden the applicability of this system. I, for one, would love to use this approach, and I am waiting for this paper to be published so I can do so. I'm sure future studies will identify the tools to turn the Killswitch into a potential therapeutic.

We are happy to share openly the killswitch tools with everybody in the scientific community!

Minor Comments:

1. The authors frequently jump back and forth between “Dynamics” and “Material Properties”, and so, they are implicitly stating that one equals the other. There are no standard measurements of material properties in this paper (e.g., rheology to determine viscoelastic moduli, surface tension, creep, etc.). Rather, there is much inferred from FRAP and shape characteristics (e.g., whether nucleoli look more or less circular). The authors should tone down their statements about the Killswitch affecting material properties.

We now include new data on in vitro assembled nucleoli, which demonstrate that the killswitch affects the material properties of condensates in vitro. In brief, we purified recombinant GFP-tagged NPM1, and assembled it into droplets in vitro, which is a commonly used approach (Feric et al., 2016; Mitrea et al., 2016; Mitrea et al., 2018). We then targeted the killswitch to the NPM1 droplets with the oligo-arginine (R10) system, imaged fusion events between droplets, and measured the inverse capillary velocity of the fusion events. These measurements are standard measurements for material properties (Brangwynne et al., 2011; Feric et al., 2016; Lafontaine et al., 2021). We found that the killswitch significantly reduced the speed of droplet fusion events, compared to the control conditions, including the key F-to-G killswitch mutant. The data confirms in a purified, isolated system, with standard measurements, that the killswitch changes condensate material properties. The data are included as Extended Data Figure 13, and are described in the text in lines 217-222. For easier inspection for the Reviewer, the data are included below at our response to Reviewer 4.

2. Susceptibility to hexanediol is not informative about material properties. We have no sense whether viscoelastic moduli change in response to hexanediol for the condensates tested. The authors need to restate this conclusion to be more precise in their language. I am referring to this statement:

“Treatment of the cells with 1-6 hexanediol dissolved BRD4::NUT but not BRD4::NUT-KS condensates, further confirming that material properties of the condensates were affected by the killswitch.”

We respectfully disagree on the notion that hexanediol is not informative about material properties. Rather, hexanediol is a very crude tool that can be used to infer some, minimal information on material properties. Hexanediol for example has been used to infer whether FUS granules have liquid-like to solid-like features (Kroschwald et al., 2017; Muzzopappa et al., 2022). We modified the sentence as follows:

“Treatment of the cells with 1-6 hexanediol dissolved BRD4::NUT but not BRD4::NUT-KS condensates, suggesting that material properties of the condensates may be affected by the killswitch (Extended Data Fig. 15h-i).”

3. Extended Data Fig. 23 seems out of place. It is mentioned in passing at the beginning of the discussion without proper justification. Why was this experiment done? How was it done? Does it belong here? Is the point that the Killswitch can be used in another species, or that the Killswitch can be used to affect function of another condensate with an easily detected readout?

The authors should consider explaining this experiment, the rationale, and results better. Otherwise, leave it out. I personally, think it's too much.

In the revised manuscript we now explain the experiment, the rationale, and results better. In brief, the experiment was done to test whether the killswitch can be used to probe condensate functions in the context of a multicellular embryo, using the well-described transcriptional bodies in Zebrafish embryos (Kuznetsova et al., 2023; Ugolini et al., 2024). Furthermore, the experiment also shows that the killswitch can be administered using an RNA-injection -based delivery system. The new text is in lines 370-375.

Referee #2 (Remarks to the Author):

I am satisfied with the revised version of the manuscript, and the authors have done their best to address the points I and some of my colleagues also raised while reading the original version of the manuscript.

We thank the reviewer for the guidance and help throughout the revision process!

Referee #4 (Remarks to the Author):

The revised manuscript from Zhang and colleagues incorporates substantial new data from multiple experiments, yet key deficiencies remain unresolved. In particular, the killswitch is presented as “a widely applicable tool to alter the material properties of endogenous condensates”. However, concerns regarding the impact of the killswitch peptide on condensate material properties are not sufficiently addressed. The FRAP analyses and hexanediol experiment are inadequate to properly assess condensate material properties. Both the native material properties of the targeted condensates and the killswitch-induced changes in these properties need to be systematically and quantitatively characterized, for example by performing in vitro measurement using purified or reconstituted condensates. Meanwhile, the newly added FACS studies seem to only complicate things by revealing that killswitch treatment alters the molecular composition of targeted condensates, which suggests an entirely separate mechanism of action. As written by the author – “the killswitch altered condensate composition, component dynamics and function, and even had an effect on the transcriptional output of transcription bodies in Zebrafish embryos. The changes appeared selective to certain condensate components, suggesting that various proteins likely partition into condensates through different underlying molecular features. The killswitch had more dramatic effects on some condensates (e.g., NUP98-fusion condensates) and marginal effect on others (e.g. chromocenters)”. Are these different effects linked to changes in the material properties of the corresponding condensates? Without a sufficient understanding of the mechanism, how would users of this tool try to interpret the data obtained? These concerns cast doubt on the potential for broader application of this tool.

We thank the reviewer for pressing on the important question on material properties, and suggesting us the reconstitutions experiments. We systematically and quantitatively characterized the effect of the killswitch on droplets *in vitro*, which provide now critical evidence that the killswitch indeed affects material properties of condensates.

We purified recombinant GFP-tagged NPM1, and assembled it into droplets *in vitro*, which is a commonly used approach (Feric et al., 2016; Mitrea et al., 2016; Mitrea et al., 2018). We then targeted the killswitch to the NPM1 droplets with the oligo-arginine (R10) system, imaged fusion events between droplets, and measured the inverse capillary velocity of the fusion events. These measurements are standard measurements for material properties (Brangwynne et al., 2011; Feric et al., 2016; Lafontaine et al., 2021). We found that the killswitch significantly reduced the speed of droplet fusion events, compared to the control conditions, including the key F-to-G killswitch mutant. The data confirms in a purified, isolated system, with standard measurements, that the killswitch changes condensate material properties. The data are included as Extended Data Figure 13, and are described in the text in lines 217-222.

Extended Data Figure 13

Extended Data Figure 13. The killswitch increases the viscosity of NPM1 *in vitro* droplets. **a.** Coomassie gel of purified recombinant msfGFP-NPM1. **b.** Representative images of the *in vitro* mixing of msfGFP-NPM1 with DMSO, TAMRA-R10, TAMRA-R10-KS, or TAMRA-R10-KS_{F10-G} peptides. Scale bar: 5 μ m. **c.** Representative images of the *in vitro* condensate fusion events at different timepoints for msfGFP-NPM1 mixed with DMSO, TAMRA-R10, TAMRA-R10-KS, or TAMRA-R10-KS_{F10-G} peptides. Scale bar: 3 μ m. **d.** Plot of the Shape Aspect Ratio vs. Elapsed Time of *in vitro* condensate fusion events. The curves show the mean value at each time point; the shade around the curves show the 95% CI. **e.** Plot of the Relaxation time (τ) vs. Length Scale (ℓ) of *in vitro* condensate fusion events. The lines are simple linear regression fits for each condition. **f.** Plot for ratio of viscosity (η) to surface tension (γ) (inverse capillary velocity) of *in vitro* condensate fusion events for msfGFP-NPM1 mixed with DMSO (n=80), TAMRA-R10 (n=88), TAMRA-R10-KS (n=88), or TAMRA-R10-KS_{F10-G} (n=87) peptides. *P*-values are from Dunnett's multiple T3 comparisons test versus DMSO condition after ANOVA.

The Reviewer raises further points on the mechanism of the killswitch, that we are happy to clarify.

The killswitch peptide has an ability to self-associate, and when targeted to a condensate-forming protein, it essentially glues the condensate-forming proteins together. We show that gluing together different condensate forming proteins consistently changes the composition of the condensates, the dynamics of their client proteins, and their cellular functions. In the case of the nucleolus, the FACS-based data commented by the reviewer revealed that there is a set of proteins whose abundance is selective reduced by targeting the killswitch to NPM1. This in itself is an important insight, which suggests that there are at least two ways to enrich proteins in the nucleolus and one is killswitch-sensitive, and the other one is not. We specifically stress that previous studies that attempt to describe the molecular basis of condensate partitioning work with a common assumption that there is one underlying so-called molecular grammar of each condensate (see e.g. (Kilgore et al., 2024)). The killswitch combined with the FACS-based sorting technology provides an experimental way to dissect the dependence of the abundance of each protein in each condensate, which we consider to be of great importance for the entire field, which lacks any such approach. In summary, the mechanism of the killswitch appears to be the same in different condensates (i.e. immobilizing the condensate-forming protein), but the consequence of immobilizing a condensate-forming proteins is different for various condensates, which makes the killswitch a unique discovery-tool.

References

- Brangwynne, C.P., Mitchison, T.J., and Hyman, A.A. (2011). Active liquid-like behavior of nucleoli determines their size and shape in *Xenopus laevis* oocytes. *Proceedings of the National Academy of Sciences of the United States of America* *108*, 4334-4339.
- Feric, M., Vaidya, N., Harmon, T.S., Mitrea, D.M., Zhu, L., Richardson, T.M., Kriwacki, R.W., Pappu, R.V., and Brangwynne, C.P. (2016). Coexisting Liquid Phases Underlie Nucleolar Subcompartments. *Cell* *165*, 1686-1697.
- Kilgore, H.R., Chinn, I., Mikhael, P.G., Mitnikov, I., Van Dongen, C., Zylberberg, G., Afeyan, L., Banani, S., Wilson-Hawken, S., Lee, T.I., *et al.* (2024). Protein codes promote selective subcellular compartmentalization. *bioRxiv*.
- Kroschwald, S., Maharana, S., and Alberti, S. (2017). Hexanediol: a chemical probe to investigate the material properties of membrane-less compartments. *Matters*.
- Kuznetsova, K., Chabot, N.M., Ugolini, M., Wu, E., Lalit, M., Oda, H., Sato, Y., Kimura, H., Jug, F., and Vastenhouw, N.L. (2023). Nanog organizes transcription bodies. *Current biology : CB* *33*, 164-173 e165.
- Lafontaine, D.L.J., Riback, J.A., Bascetin, R., and Brangwynne, C.P. (2021). The nucleolus as a multiphase liquid condensate. *Nature reviews Molecular cell biology* *22*, 165-182.
- Mitrea, D.M., Cika, J.A., Guy, C.S., Ban, D., Banerjee, P.R., Stanley, C.B., Nourse, A., Deniz, A.A., and Kriwacki, R.W. (2016). Nucleophosmin integrates within the nucleolus via multi-modal interactions with proteins displaying R-rich linear motifs and rRNA. *eLife* *5*.

Mitrea, D.M., Cika, J.A., Stanley, C.B., Nourse, A., Onuchic, P.L., Banerjee, P.R., Phillips, A.H., Park, C.G., Deniz, A.A., and Kriwacki, R.W. (2018). Self-interaction of NPM1 modulates multiple mechanisms of liquid-liquid phase separation. *Nature communications* 9, 842.

Muzzopappa, F., Hummert, J., Anfossi, M., Tashev, S.A., Herten, D.P., and Erdel, F. (2022). Detecting and quantifying liquid-liquid phase separation in living cells by model-free calibrated half-bleaching. *Nature communications* 13, 7787.

Ugolini, M., Kerlin, M.A., Kuznetsova, K., Oda, H., Kimura, H., and Vastenhouw, N.L. (2024). Transcription bodies regulate gene expression by sequestering CDK9. *Nature cell biology* 26, 604-612.